# Chitosan: A Potential Biopolymer in Drug Delivery and Biomedical Applications

**DOI:** 10.3390/pharmaceutics15041313

**Published:** 2023-04-21

**Authors:** Nimeet Desai, Dhwani Rana, Sagar Salave, Raghav Gupta, Pranav Patel, Bharathi Karunakaran, Amit Sharma, Jyotsnendu Giri, Derajram Benival, Nagavendra Kommineni

**Affiliations:** 1Department of Biomedical Engineering, Indian Institute of Technology Hyderabad, Kandi 502285, India; bm21resch11003@iith.ac.in (N.D.); jgiri@bme.iith.ac.in (J.G.); 2National Institute of Pharmaceutical Education and Research (NIPER), Ahmedabad 382355, India; dhwanirana73@gmail.com (D.R.); sagarsalave1994@gmail.com (S.S.); raghav.gupta@niperahm.res.in (R.G.); pranavpatel7111@gmai.com (P.P.); bharathi.k@niperahm.res.in (B.K.); amit.sharma@niperahm.res.in (A.S.); derajram@niperahm.res.in (D.B.); 3Center for Biomedical Research, Population Council, New York, NY 10065, USA

**Keywords:** chitosan, chitin, drug delivery, tissue regeneration, wound healing

## Abstract

Chitosan, a biocompatible and biodegradable polysaccharide derived from chitin, has surfaced as a material of promise for drug delivery and biomedical applications. Different chitin and chitosan extraction techniques can produce materials with unique properties, which can be further modified to enhance their bioactivities. Chitosan-based drug delivery systems have been developed for various routes of administration, including oral, ophthalmic, transdermal, nasal, and vaginal, allowing for targeted and sustained release of drugs. Additionally, chitosan has been used in numerous biomedical applications, such as bone regeneration, cartilage tissue regeneration, cardiac tissue regeneration, corneal regeneration, periodontal tissue regeneration, and wound healing. Moreover, chitosan has also been utilized in gene delivery, bioimaging, vaccination, and cosmeceutical applications. Modified chitosan derivatives have been developed to improve their biocompatibility and enhance their properties, resulting in innovative materials with promising potentials in various biomedical applications. This article summarizes the recent findings on chitosan and its application in drug delivery and biomedical science.

## 1. Introduction

Chitosan, a cationic polysaccharide derived from the deacetylation of chitin, is one of the biomaterials used abundantly in drug delivery applications. It is composed of repeating units of β-(1–4) N-acetyl glucosamine and D-glucosamine with native amine groups that undergo protonation at the physiological pH [1]. Chitosan is soluble in aqueous acids such as acetic acid and lactic acid but exhibits poor solubility in neutral and basic media. The solubility of chitosan is determined by the degree of deacetylation, molecular weight, pH, temperature, and polymer crystallinity; a high degree of deacetylation and a low molecular weight improve solubility [2,3]. The positive charge of the ionizable amino group promotes electrostatic interaction with the negatively charged mucosal surfaces, conferring mucoadhesive properties to chitosan-based drug delivery carriers. However, the low water solubility of chitosan and its poor mechanical properties necessitates the need for several modifications at the hydroxyl and free amino groups, that would improve its solubility for suitability in tissue engineering and drug delivery applications [4]. Though the modifications do not alter the basic skeleton of chitosan, it has been found that the derivatives exhibit enhanced properties for effective drug delivery. The incorporation of therapeutically active agents in the polymeric matrices made of chitosan derivatives protects the biologically active agent from degradation. In addition, it also confers controlled-release characteristics, improves absorption, and subsequently results in the reduction of frequency of administration, thereby improving patient compliance. Quaternary chitosan derivatives, thiolated chitosan, mono-N-carboxymethyl chitosan, and chemical grafting of chitosan using radiation, enzymatic, free-radical, or cationic polymerization are some examples of modifications that have been explored [5].

Chitosan’s relatively stable chemical structure, along with polycationic, innocuous, non-toxic, and biodegradable qualities, makes it biocompatible with a wide range of organs, tissues, and cells. Chitosan molecules are both physically and physiologically active, and they may be chemically or enzymatically changed for a variety of applications [6]. Chitosan of varying molecular weights, ranging between 2800 and 30,000 Daltons, that are categorized by different degrees of deacetylation are commonly used in the design of drug delivery carriers. The pharmaceutical and biotechnological applications of chitosan-based nanomaterials are manifold owing to their muco-adhesiveness, biocompatibility, biodegradability, and the ability to attain tailorable characteristics through numerous chemical and mechanical modifications [7]. The wide range of therapeutic activities exhibited by chitosan-based nanocarriers, including their antifungal, anti-tumor, antiviral, anti-inflammatory, and anti-bacterial activities, has triggered extensive research on exploring the various applications of chitosan as drug delivery carriers [8]. Various drug carriers including micro granules, microspheres, hydrogels, nanoparticles, and nanofibers have been explored for oral, ophthalmic, transdermal, nasal, and vaginal delivery. Further, chitosan has shown immense potential as a functional biomaterial in the design of scaffolds intended for tissue engineering applications, thus enabling the regeneration of various tissues such as the cornea, cartilage, bone, and cardiac tissues [9]. The molecules of chitosan encourage three-dimensional cell development and proliferation, as well as coordinate collagen deposition, enabling rapid healing [10]. The polycationic nature of chitosan derivatives facilitates complex formation with nucleic acids that carry a negative charge. Hence, chitosan derivatives are now also being explored as non-viral gene delivery vectors, enabling efficient transfection of genetic material into the cells for modification of function or behavior in the treatment of various diseases. The low toxicity (LD50: 16 g/Kg) and low immunogenicity of chitosan-based gene delivery vectors aid in overcoming the substantial toxicity issues associated with commonly used gene delivery vectors such as polyethyleneimine [11].

This review discusses in detail the various biological activities exhibited by the cationic polysaccharide chitosan, which are responsible for its promising potential in the use as a drug delivery carrier. The modifications of chitosan for obtaining the desired characteristics have been explained. Further, a complete overview of the applications of chitosan-based nanomaterials in the management of various diseases, with the most recent literature review, has been discussed.

## 2. Source of Chitosan

Naturally, chitosan is not present in the environment. It is obtained by the deacetylation process of chitin. Chitin has been extracted from a variety of natural resources, such as marine sources, terrestrial sources, and microbial sources. Chitin composition depends on age, season, gender, and other environmental factors [12,13].

### 2.1. Marine Source

Chitin biopolymers could be found in many marine microorganisms, such as crustaceans (for example, lobster, barnacles, crayfish, shrimp, krill, etc.), mollusks (cuttlefish, oysters, octopus, clams, and snails squids), and algae (brown algae, diatoms, and green algae) [13]. Crustacean shells are one of the most abundant sources of chitosan and chitin owing to their well-established industrial extraction techniques. As per the FAO, approximately 11.2 million metric tons of crustacean shells were produced in 2022 [14]. Squid pen is the primary source of beta chitin in crustaceans. It consists of around 31–49 percent chitin. Chitosan is also extracted from squid pans. It has huge market potential due to being free of copolymers and minerals [15]. Fish waste is a viable source of chitin and chitosan. Globally, nearly 50 million tons of fish waste are generated, with a significant portion of this amount being effectively used for a range of applications [16]. Crab exoskeletons and krill both contain 15–30% and 20–30% chitin, respectively. Framed shrimp also contain 18% chitin. The enormous amount of waste that is readily generated from the food processing industry makes crustaceans the most commonly used chitin source [17].

### 2.2. Terrestrial Source

The terrestrial source of chitin includes cockroaches, silkworms, honeybees, mosquitoes, *Extatosoma tiaratum*, *Sipyloidea sipylus*, and *Drosophila melanogaster*. Its cuticle and wings contain chitin. Chitin obtained from the insect is mainly used in the pharmaceutical industry owing to its low quantity of mineral compounds compared to the crustacean shell [18]. Terrestrial sources are important for viable production and research. Some species could be artificially reared, such as honey bees, silkworms, and other synanthropic species, in order to produce chitin for industrial use. The greatest amount of chitin is obtained through beekeeping, which can provide the ability for bees to produce more chitin. Chitin is also found in beetle larvae, beetles, grasshoppers, and crickets [12].

### 2.3. Microbial Source

Fungus is primarily used in the production of chitin from a microbial source. Chitin, beta-glucan, and chitosan are located in the fungal cell wall. The structural components of fungi have 22–44% of chitin. Due to their highly uniform and pure properties, chitosan and chitin derived from microbial origins became popular during the modern era [19]. It has the potential to solve the problem of the impact of environmental and seasonal variations on marine chitin. It is a simple process to cultivate and extract, as well as an environmentally friendly process that supports the usage of fungi as the primary source of chitin and chitosan. Chitin is present in chrysophyte algae, molds, yeast, fungi, prosthecate bacteria, ciliates, and some spore-forming bacteria such as spores Streptomyces sps. However, fungi-based fermentation is a commercially viable technique for chitosan and chitin production. The fungal waste biomass was produced by some bioprocess industries. which can also be used to produce chitin and is thus environmentally friendly. In recent years, a new source of chitin and chitosan has emerged. For example, in the biotech industry, citric acid production on a commercial scale generates huge quantities of mycelial waste, which is used for chitin and chitosan production [12,20].

## 3. Chitin and Chitosan Extraction Techniques

Chitosan and chitin are extracted from insects, crustaceans, and fungi with the help of biological and chemical methods. Figure 1 demonstrates the extraction of chitin and chitosan from various sources. The extraction method and steps used depend on the source of the raw material, such as fungi, crustaceans, and insects. Chitin is found abundantly in the exoskeleton of crustaceans and insects; in contrast, chitin and certain amounts of chitosan are also found in the fungal cell wall [12]. Further, the exoskeleton of crustaceans is made up of minerals such as inorganic carbonate salt, chitin-protein complex carotenoids (astaxanthin), and lipids. Although insects have the same exoskeleton as crustaceans, they have fewer minerals. The cell wall of fungi is made up of the chitin-glucan complex, glycoprotein, and a minor proportion of lipids, pigments, and inorganic salt. In general, the crustacean contains 20–30% chitin. Insects contain 5–25% chitin, and fungi contain 2–44% chitin, which is chemically linked to glucans (80–90%). This proportion varies according to the species’ growing environment [13]. Therefore, individual steps for extraction can change or even not be required depending on the biomass or raw material. For instance, demineralization, deproteinization, bleaching, and finally the deacetylation process all contribute to the extraction of chitosan. Crustaceans have a higher mineral content than fungi and insects; therefore, demineralization is an important step for chitosan extraction from crustaceans. In contrast, the cell wall of fungi does not contain minerals; therefore, the demineralization process does not involve the extraction of chitosan. Bleaching is an additional step for the extraction of chitosan, which is either involved or not in the extraction depending on the raw material used for extraction [13,21].

### 3.1. Preparation of Chitin

Before the chemical and biological extraction, some pretreatment methods are used for the preparation of the biomass for the extraction of chitin. This may include soft tissue removal by scraping, boiling, and pressing. After the biomass has been boiled, it must be mechanically dried and crushed. Mechanical crushers convert biomass into small particles. This fine-particle raw material is used in subsequent processing [21].

#### 3.1.1. Chemical Method

The chemical method includes four subsequent processes, such as demineralization, deproteinization, discoloration, and deacetylation [22]. Some authors used interchangeable processing sequences for demineralization and deprotonation processes, such as Kandile et al., for extraction from the shrimp shell beginning with the deproteinization step in contrast to Ali et al. for extraction from the crab shell beginning with the demineralization step. Some researchers conclude that this interchangeable process did not affect yield or quality [23,24].

Demineralization Process

Demineralization is preferred when the biomass contains a high concentration of minerals, such as 50% of CaCO_3_ present in the exoskeleton of crustaceans [25]. An acidic solvent reacts with the shell to eliminate minerals like calcium phosphate and calcium carbonate during the chemical demineralization process. The most widely used acids are HCl, HNO_3_, H_2_SO_4_, CH_3_COOH, and HCOOH. HCl is the most commonly used acid for the removal of minerals [26]. For demineralization, the acid reacts with the shell, and after this process, the demineralized shell is separated by a vacuum filter and washed with distilled water until pH becomes neutral. The demineralized shell is then dried in an oven at approximately 60 °C for 24 h [27]. Temperature, shell size, duration of extraction, solute/solvent ratio, and the concentration of acid depend on the types of shells or biomass characterization. Because solvent diffusion into the chitin matrix is temperature and particle-size-dependent, higher temperatures and smaller particles efficiently perform demineralization. Despite the fact that higher temperatures and a high concentration of acid negatively impact the properties of chitin. Different researchers developed different demineralization processes for the extraction of chitin, but all used an acidic solvent for the demineralization process [28].

Deproteinization of chitin

The demineralized chitin chain embeds in the protein matrix. Because this protein may be allergenic to the human body, it should be avoided whenever possible. As a result, when chitin is used in medicine or the food industry, it is best to remove all of the protein [29]. In deproteination, the stringent process parameter is required because the chitin-free amino group is bound to protein by covalent and multiple hydrogen bonds [30]. In the deproteination process, biomass is treated with an alkaline reagent. NaOH, Na_2_CO_3_, Ca (OH)_2_, NaHSO_3_, CaHSO_3_, Na_2_SO_3_, NaHCO_3_, KOH, Na_3_PO4, K_2_CO_3_, and Na_2_S are commonly used reagents. The protein is removed in this process when alkali reacts with the biomass with constant stirring for 2 h at approximately 90 °C. The insoluble chitin is then separated using a vacuum filter, and distilled water is passed through it until the pH is neutral and then dried with the help of an oven at around 60 °C [27]. The process temperature, the alkali concentration, and the biomass/alkali ratio directly impact the alkali deproteination process [31]. Based on the biomass characteristics, a decision is made for the process parameter because the breakdown of chemical bonds occurs between the chitin and protein. In general, KOH and NaOH are widely used for the deproteination process. NaOH performs fast deprotonation; therefore, it is often used in the industrial method, but it has a high Na concentration by-product, which can adversely impact the environmental soil and water systems. In contrast to NaOH, the literature suggests that the use of KOH has a lower environmental impact because KOH generates K rather than high concentrations of Na. This is beneficial because K is an important element for plant growth and in fertilizer [22].

Decolorization/bleaching and post-treatment

The final step in the process of chitin extraction is decolorization. This step helps remove the pigments, such as the pink color of crustaceans. Organic solvents like ethanol and chloroform have been used for decolorization, and hydrogen peroxide and potassium permanganate have been used for the bleaching of the chitin [32]. For instance, crustacean decolorization has been performed by bleaching agents such as sodium hypochlorite or potassium permanganate, hydrogen peroxide, or oxalic acid, while insect chitin decolorization has been accomplished using a mixture of chloroform and methanol or alcohol and chloroform [33]. To complete the chitin production process, some final post-treatment may be required, such as neutralization, drying, and milling [21].

#### 3.1.2. Biological Extraction of Chitin

In the chemical method of extraction, harsh chemicals react with the biomass at elevated temperatures for a prolonged period of time, resulting in changes to the physicochemical characteristics and functionality of chitosan. This chemical is also extremely hazardous to the environment. Purification of chitin obtained by chemical methods is tedious, energy- and time-consuming, and hazardous to the environment due to chitins containing high concentrations of caustic soda and minerals. Therefore, the biological extraction method is widely used to overcome these issues. The addition of greater reproducibility in chitin production is accomplished by biological extraction. The biological extraction method is performed by two methods: the enzymatic method and the fermentation method [27].

Enzymatic Method

How harsh an acid treatment is will determine whether it removes minerals such as calcium carbonate in chemical demineralization. Similarly, in the case of enzyme-mediated demineralization, acid has been used for demineralization, but lactic acid-producing bacteria are used. When this acid reacts with the calcium carbonate, the calcium salt precipitates in the solvent. The extra attention required to remove this precipitate after washing it serves as a preservative and anti-itching agent [34]. After demineralization, enzymatic deproteination was performed by the bacterially secreted proteolytic enzymes such as proteases (pepsin, papain, and trypsin) or isolated proteases [35]. For the production of chitin, protein hydrolysate, and astaxanthin recovery, the alclase enzyme is widely used because it controls the hydrolysis as well as the production of non-bitter hydrolysate. *Bacillus licheniformis* is used for the extraction of alclase [36]. In enzymatic processes, the process parameter is less harsh compared to the chemical deproteination; therefore, the residue of protein remains attached to the chitin chain after enzymatic deproteination, and the degree of deacetylation is less [16]. The chemical reaction resolved the inefficiency of the enzymatic reaction; therefore, enzymatic extraction is a combination of enzymatic and chemical reactions [37]. Even with the less harsh reaction parameters of the enzyme method, it has some limitations, such as an industrial-scale production cost because the enzyme used in this process is highly costly compared to the chemical reagent. As previously stated, this method could not remove the final residue of 10% of proteins, and the degree of deacetylation was not efficient [29,38].

Fermentation Method

Fermentation solves the issue associated with the enzymatic method, which uses proteolytic enzymes and acid-secreting microbes. In a bioreactor with favorable conditions, this microbe multiplies rapidly, reducing the high enzyme cost. As a result, most publications used the fermentation method for chitin/chitosan preparation instead of the enzymatic method [39]. The two types of fermentation methods are lactic acid fermentation and non-lactic acid fermentation, depending on whether the microbial strains secrete lactic acid [40]. The generally used lactic acid-producing bacteria belong to the Lactobacillus sps. (*L. paracasei*, *L. helveticus*, and *L. plantarum*). Fungi are also used for fermentation, such as Aspergillus sp., Pseudomonas sp., and Bacillus sp. [12,37]. Sometimes demineralization and deproteination reactions can occur partially simultaneously because pH is reduced by the acid, which results in the activation of the proteases [29]. The fermentation method has a lower negative impact on the chitin’s physicochemical characteristics; therefore, chitin has better mechanical properties [41]. The fermentation method has some drawbacks, such as the need for bacteria for fermentation, which takes a few days, making the process time-consuming and sometimes requiring a special microbial strain, making this process very complex because it is costly, is time-consuming, and needs a specialized facility. As previously stated, it also has less efficiency for the deproteination reaction, the same as the enzymatic method [22].

### 3.2. Preparation of Chitosan

Chitin is not used in the pharmaceutical industry because it is crystalline and does not dissolve in most solvents. However, it does dissolve in some solvents that are toxic or corrosive; therefore, it has not yet been explored in pharmaceutical scale-up applications [42]. Therefore, it is essential for chitin to convert its insoluble form derivatives into its soluble form derivatives. This intramolecular and intermolecular bond is responsible for crystallinity cleavage in the various reactions. Among these, N-deacetylation is the most common reaction for chitin to chitosan conversion [43]. In deacetylation, first, hydroxide ions attach to the carboxy group, and second, acetic acid splits off and forms amine [44]. This molecule is called chitosan. Chitosan is easily soluble in diluted acids that possess a pH of below 6 but are insoluble in water and organic acids. The number of acetyl functional groups that are replaced by the amine group is known as the degree of deacetylation (DDA). The extent of deacetylation greatly impacts the physicochemical properties of chitosan such, as when DDA is 70–85%, it is partly soluble in the diluted acid, an 85–95% DDA, making it highly soluble in diluted acids. It also affects acid–base behavior, biodegradability, electrostatic properties, self-aggregation, and sorption properties. Chitosan is one of the most important chitin derivatives in terms of applications [45]. Chitosan is produced in two ways: first, chitin is converted into chitosan via a deacetylation step, and second, chitosan is directly extracted from fungi biomass. The first option is commonly used for the production of chitosan [21].

#### 3.2.1. Chemical and Biological Deacetylation of Chitin

Although both acids and alkalis can deacetylate chitin, alkalis are more commonly used because the glycosidic bond is very susceptible to acid. As a result, depolymerization occurred in the acid [29]. There are two types of chemical deacetylation: homogeneous and heterogeneous [21]. The homogenous deacetylation method is performed in two steps. The first step is to make insoluble chitin soluble in water by reacting it with NaOH 10% (*w*/*v*) at room temperature for around 70 h. The second step is the deacetylation reaction, which is carried out under moderately alkaline conditions with the soluble chitin, for example, with an alkali 13% (*w*/*v*) for 12–24 h at around 25–40 °C [46]. It is possible to perform heterogeneous deacetylation on solid chitin in a single step. In heterogenous reactions, deacetylation reactions are only observed at amorphous zones, so a more block-like deacetylation pattern is shown. Higher temperatures of around 100–160 °C and higher aqueous alkali concentrations of around 40–50% (*w*/*v*) are required for the heterogenous deacetylation reaction [35]. The heterogeneous method is generally preferred by the industry for the deacetylation reaction because it makes it easy to separate the insoluble chitosan from the liquid NaOH residue and product [38,47]. Furthermore, in enzymatic deacetylation, biological sources such as insects and fungi are used to obtain chitin deacetylase. *Mucor rouxii*, *Colletotrichum lindemuthianum*, *Absidia coerulea*, and *Aspergillus nidulans* are all important fungal chitin deacetylases. This enzyme has a binding affinity towards β-(1, 4)-linked N-acetyl-D-glucosamine polymers and is also thermally stable. Batch and continuous cultures are both used for the enzyme process [13,48].

#### 3.2.2. Chitosan Extraction from Fungal Cell Wall

Both chitosan and chitin can be biosynthesized in the fungal cell wall. This chitosan can be isolated from the cell wall of fungi with the help of fermentation technology on an industrial scale [13]. Chitosan can be extracted from the Zygomycetes fungi, including Gongronella, Absidia, Rhizopus, Mucor, and *G. butleri*. The studies discovered that *G. butleri* yielded the highest amount of chitosan [49]. The main fungal skeleton polysaccharides are chitin/chitosan and glucan. Treatment with a strong alkali is required for the extraction of chitosan because the chitosan/chitin-glucan chain complex forms a cross-linked, rigid structure, resulting in issues associated with the extraction of intact chitosan and glucan [50]. Traditionally, chitosan is extracted in two steps, such as through base and acid treatment of the fungal cell wall. Following the fermentation of fungi, it is treated with alkaline conditions (for example, 4% to 2% NaOH at an elevated temperature of 90–121 °C for 15–120 min), resulting in an alkaline-soluble substance soluble in the alkali solution and a remaining alkali-insoluble substance containing protein and chitosan. To treat this alkali-insoluble substance, acids such as acetic acid, lactic acid, and hydrochloric acid (such as 2–10% acetic acid at 25–95 °C for 1–24 h) are used, resulting in the insoluble material fungal chitosan becoming soluble in the solution. This soluble chitin was separated from the solution with the help of precipitation that occurred when the pH was raised to 9–10. This is accomplished by using an alkali solution such as 2 N NaOH. This precipitate was separated with the help of centrifuges and washed with ethanol and acetone [51]. The enzymatic method is also used for the extraction of chitosan from the cell wall of fungi with the help of glucanase, chitinase, and amylase enzymes. This method does not require the demineralization and deproteination processes for the extraction of chitosan. This fungal chitosan contains no protein residue, which is responsible for the allergic reaction [52].

## 4. Bioactivities of Chitosan

The biological activity of chitosan and its derivatives are quite diverse. Chitosan is effective against a wide variety of bacteria, filamentous fungi, and yeasts [53]. The below section provides a brief overview of chitosan’s bioactivities.

### 4.1. Antibacterial Activity

Chitosan has a broad spectrum of action against several pathogens of bacterial origin and a considerable killing rate against both gram-negative and gram-positive species [29,54]. However, scientific investigations have revealed that chitosan has varying effects on various bacterial species. The precise mechanism by which chitosan and its derivatives exhibit its antibacterial action is not fully understood; however, several investigations have been undertaken to speculate on the underlying antimicrobial processes [3,54,55,56,57]. The mechanism of antibacterial action of chitosan has been the subject of several hypotheses. Chitosan’s antibacterial activity can be attributed to its similarity to cell surface characteristics as bacterial surfaces are chemically heterogeneous and structurally complex [54]. It has been proposed that the death caused by chitosan is not the product of one specific molecular target but rather a series of “untargeted” molecular events that may occur concurrently or sequentially. The largest sensitivity to chitosan was seen in the dltA mutant, suggesting that teichoic acids located on the cell surface play a significant role in the first contact between the negatively charged cell wall and polycationic chitosan macromolecule polymers by the virtue of electrostatic interactions. This leads to disruption of the equilibrium of cell wall dynamics. It is hypothesized that chitosan upon binding to the cell wall results in a cascade of secondary cellular effects, including the destabilization and consequent disruption of bacterial membrane function (through unknown mechanisms), which would result in the leakage of cellular components across the membrane. Additionally, the energy generation pathways that are membrane-bound are impacted, most likely as a result of disruption of the functional structure of the electron transport chain, which impedes the reduction of oxygen and forces the cells to switch to the anaerobic pathway for energy production. As a result, the entire cellular machinery may become dysfunctional. The polymer accumulation near the membrane might potentially set off a cascade of stress reactions owing to a local low pH or due to other causes that are yet to be found. However, the precise mechanisms by which these activities are connected or interrelated need to be completely elucidated [58].

Chitosan’s antibacterial activity is inhibited by its insolubility in water. In one work, Kong et al. developed an asymmetric bilayered nanofibrous membrane with the hydrophobic polycaprolactone (PCL) nanofibrous membrane as the top layer and the hydrophilic CS/chitosan oligosaccharide (COS) nanofibrous membrane as the bottom layer. The results revealed that adding COS enhanced the wettability of the CS membrane and that doing so increased the width of the inhibitory zones of E. coli and Staphylococcus aureus by 23% and 26%, respectively. The PCL layer may also prevent the adherence of bacteria and water. The PCL-CS/COS 0.5% membrane displayed acceptable cytocompatibility, outstanding water absorption (460%), and comparatively high mechanical characteristics. The potential for this asymmetric wettable membrane to function as a novel antibacterial dressing for wound healing is enormous [59]. The antimicrobial activity of chitosan of varying molecular weights was investigated by Zheng et al. using the gram-negative bacteria *E. coli* and the gram-positive bacteria *S. aureus* as test organisms. According to this research, the following two pathways are among the potential causes of antimicrobial action. On the cell surface, chitosan (higher molecular weight) can create a polymer barrier that inhibits the entry of nutrients into the cell. The lower molecular weight of chitosan can infiltrate the cell via permeation. The adsorption of electronegative material by chitosan and its subsequent entry into the cell results in its flocculation, causing disruption of the physiological activities, thereby killing the bacteria. For *S. aureus*, the former is the predominant mechanism, but for *E. coli*, the latter looks more plausible. It was noticed that the rise in the concentration of chitosan led to an intensification of the antibacterial action. When the concentration reached 1%, the inhibition rate for both *E. coli* and *S. aureus* reached 100% [60]. Moreover, it has also been reported that chitosan binds to the cell surface at low concentrations, causing its disruption through the leakage of intracellular components; however, protonated chitosan, at high concentrations, coats the cell surface, thereby preventing the leakage of intracellular components [57].

As previously observed, apart from chitosan’s molecular weight and its concentration, several other factors such as the physical state of chitosan, its chelating capacity with various metal ions, positive charge density, degree of substitution, and hydrophilic and hydrophobic characteristics and environmental factors such as pH, ionic strength, temperature, and time affect the antibacterial potency of chitosan [57,61]. Numerous studies have summarized the correlation between chitosan’s antibacterial activity, its degree of deacetylation, and pH. For instance, chitosan’s antibacterial action improves as its degree of deacetylation rises. It has been hypothesized that chitosan, being a cationic polymer, exhibits antibacterial action at pH levels below 6.5. Chitosan’s antibacterial action is highly conditional on both its characteristics and the particular strains of bacteria under investigation [62].

The antibacterial activities possessed by chitosan and one of its derivatives named chitosan oligosaccharide lactate (COL) were investigated by Yildirim-Aksoy et al. against three pathogens, namely, *Edwardsiella ictaluri*, *Aeromonas hydrophila*, and *Flavobacterium columnare*. Both COL and chitosan exhibited dose-dependent antibacterial activity against all three microorganisms tested; however, COL was found to be more effective in causing the inhibition of bacterial growth. The COL molecules and chitosan possess the ability to adsorb onto bacterial surfaces. The conductivity of bacterial cell solutions treated with chitosan or COL initially decreased, proving that these compounds inhibited the growth of all bacteria examined. There was a significant rise in conductivity again between 18 and 48 h, perhaps because of cellular ions leaking into the solution as a result of damaged bacterial cell membranes. Following a sequential process that began with adsorption to bacterial surfaces and ended with the leaking of internal components and cell death, this study revealed that both chitosan and COL displayed antibacterial action against all three bacterial species [53]. Vishu Kumar et al. developed chitooligosaccharides and evaluated their inhibitory action against *E. coli*. and *Bacillus cereus*. In comparison to native chitosan, the mixture of chito-oligomeric-monomers had a superior bactericidal effect against *B. cereus* and *E. coli*. Chito-oligomers with a higher degree of polymerization and a lower degree of acetylation showed the best growth inhibition. The latter resulted in pore formation and enhanced cell wall permeability of B. cereus, whereas nutrient flow blockade caused by aggregation of chito-oligomers-monomers were responsible for the inhibition and lysis of *E. coli* [63].

Moreover, antibiotic-free antimicrobial strategies have also attracted the attention of researchers. Common antibiotic-free antimicrobial agents include inorganic metal-based antibacterial substrates, and metal oxides have been used to prevent bacterial biofilms [64]. Liu et al. used a polydopamine in situ reduction technique to incorporate silver nanoparticles (Ag NPs) into the flower-like hierarchical chitosan/calcium pyrophosphate hybrid micro-flowers (CS-CaP). The interaction of Ag NPs with bacteria may be facilitated by the hierarchical structure of Ag@CS/CaP, enhancing the antibacterial properties. The developed Ag@CS-CaP had significant antimicrobial activity against both gram-negative and gram-positive bacteria, according to the in vitro antibacterial results. The findings establish that Ag@CS-CaP has the potential to be used in biomedical applications [65]. Across the board, chitosan’s antibacterial capabilities have been highlighted.

### 4.2. Antifungal Activity

Chitosan has potential therapeutic activity against fungal infections owing to its biodegradability, biocompatibility, and low toxicity [66]. When used to treat disorders brought on by human pathogenic fungus, chitosan shows significant promise as an antifungal agent. The cell wall of fungi serves as the interface between cells and its surroundings, thereby serving as a physical barrier to protect fungal cells from harmful environmental factors [67]. The compact network of fungi’s cell wall is not only necessary for the maintenance of its morphology and viability, but it also mediates the adhesion between cells and environmental material surfaces through the activity of several adhesin proteins, which is an early and important step in biofilm development [68,69].

*Candida albicans*, a polymorphic fungus, can cause infections ranging from superficial infections to life-threatening systemic infections [70]. Chitosan has been shown to have both physical extracellular interaction capabilities and the potential for intracellular activity for *C. albicans*. Research shows that chitosan inhibits ADA2 and Ada2-regulated genes present in the cell wall’s structure. Additionally, chitosan represses the GCN5 HAT, the GCN5 mutant strains, and cell wall-disrupting (calcofluor white and caspofungin) agents. The data imply that, upon treatment of *C. albicans*, the inhibition of SAGA complex component expression decreases the levels of β-glucan and chitin or causes alteration of the ultrastructure of the cell wall and cell membrane [62]. Some studies have shown that chitosan causes energy-dependent plasma membrane permeabilization in sensitive fungi [71].

The opportunistic fungal pathogen *Cryptococcus neoformans*, which causes cryptococcal meningoencephalitis, especially in immunocompromised people, requires chitosan to preserve cell wall integrity throughout the vegetative stage of its life cycle [72]. It has been well-documented that chitosan and its nanoparticle systems have antifungal potential against a range of fungi, including *C. albicans*, *Fusarium solani*, and *Aspergillus niger* [73]. The anticandidal activity of four types of fungal chitosan derived from *Mucor rouxii* DSM-1191 has also been determined against three *C. albicans* strains. Chitosan with a high degree of deacetylation of around 94% and a molecular weight of 32 kDa was found to be the most efficacious in causing the inhibition of *C. albicans* growth. Antifungal activity was higher in the water-soluble forms than in the 1% acetic acid solution-soluble forms. The primary interaction between the yeast cell wall and chitosan is its significant swelling and asymmetric rough forms, followed by cell wall lyses when exposure duration is increased. As a safe and effective alternative to chemical and synthetic fungicides, fungal chitosan may be advised for the control of *C. albicans* [74]. Moreover, Seyfarth et al. evaluated the antifungal effect of low- and high-molecular-weight chitosan hydrochloride, N-acetyl-d-glucosamine, chitosan oligosaccharide, and carboxymethyl chitosan against *C. krusei*, *C. albicans*, and *C. glabrata*. Results showed that *C. krusei* and *C. albicans* were the most vulnerable species tested. There was also a decrease in *C. glabrata* development; however, 1% of tested substances could not inhibit its growth completely. High correlation was found between the test and inhibition concentration, indicating that both chitosan hydrochlorides had an antifungal action. Chitosan oligosaccharide, carboxymethyl chitosan, and N-acetyl-d-glucosamine, on the other hand, had minimal to no antifungal action. According to the findings, the antifungal activity declines with decreasing molecular mass (such as N-acetyl-d-glucosamine and chitosan oligosaccharide) and rising masking of the protonated amino groups with functional groups such as carboxymethyl chitosan [75].

### 4.3. Antiviral Activity

Chitosan and chitosan-derived compounds have been demonstrated to have bioactivities including antiviral. Chitosan derivatives are thought to have antiviral effects owing to its ability to bind cell receptor molecules as a result of their protonated amino groups, varying degrees of acetylation, and the modification of polycationic moieties. Chitosan and its oligosaccharides can exhibit affinity for a variety of chemical substances, including fatty acids, other sugars, peptides, and phytochemicals like flavonoids and tannins, which is attributable to their ability to be open to modification at these positions [76]. The mucoadhesive qualities of chitosan also allow it to stay on the mucus surface of the host for extended periods, where it may exert its antiviral actions. It has been observed that bacteriophage, tobacco mosaic virus, murine norovirus, and feline calicivirus replication is inhibited by chitosan itself at high molecular weights (50–1000 kDa) with a degree of acetylation between 10% to 30%. Additionally, lowering the polymerization degree of low molecular weight chitosan increases their antiviral activity. Chitosan with a molecular weight below 10 kDa has been reported to display antiviral effects against several influenza virus subtypes, suggesting that molecular weight plays a role in antiviral activity [77,78].

Upon hydrolysis, chitosan yields polysaccharides known as chitosan oligomers (COS). The tripeptides consisting of tryptophan (W), methionine (M), and glutamine (Q) in conjugation with the chitosan oligomers QMW-COS and WMQ-COS have been demonstrated to be efficient HIV-1 fusion inhibitors and shown inhibition in HIV-1 replication possibly via blocking the interaction of HIV-1 glycoprotein gp41 with the cell membrane. However, the inhibitory potential of pep-COSs may depend on parameters such as the specific amino acid attachment, the sequence of amino acids, and the number of amino acids connected to the COS skeleton [79]. Similarly, asparagine-conjugated (COS-N) and glutamine-conjugated (COS-Q) COS have demonstrated HIV-1 inhibitory effects possibly by blocking the connection between HIV-1 (or infected cells) and uninfected cells. To put it another way, COS-N and COS-Q were able to protect cells against the lysis caused by HIV-1. The synthesis of the viral protein p24 was also reduced in COS conjugate-treated cells relative to both the COS-treated and untreated groups [78].

### 4.4. Anti-Tumor Activity

Drug delivery using chitosan, which contains a positive charge that facilitates non-covalent interactions with biological tissues, has been explored as a potential strategy for addressing the limitations of current chemotherapeutic approaches. Therapeutic agents conjugated with chitin or chitosan derivatives have demonstrated remarkable anticancer activity with fewer side effects than the original drugs [80]. Both acidic pH and a reductive environment are common in cancer tissue and have been extensively researched as an internal trigger for drug release in smart drug delivery systems. Chitosan may also accumulate at the tumor site, initiating M1 macrophage polarization and altering the immunosuppressive tumor microenvironment to an immune-supportive one, so boosting the efficacy of cancer immunotherapy and having an anticancer impact [81]. Studies have shown that chitosan formulations can upregulate pro-inflammatory markers, induce inflammatory mediators, and expand tumor-targeting cell types, all of which contribute to an immunostimulatory profile of antigen-presenting cells and adaptive immune cells and can reverse the generalized inactivation of the immune cells [82]. Furthermore, chitosan has shown promise as a therapeutic agent and a drug carrier for its anticancer effects. It has been hypothesized that their anticancer effect as a therapeutic agent is connected to their capacity to stimulate T-cell proliferation and hence trigger the generation of cytokines. It has been observed that it inhibits MMP-9 and has potent pro-apoptotic effects on tumor cells [66].

It has been demonstrated that chitosan derived from fungi inhibited the progression of colon precancerous lesions in vivo and that the underlying molecular mechanisms involved in the chitosan-mediated inhibition of cancer cell growth may involve the suppression of DNA synthesis through modulation of cell cycle regulatory molecules [83]. Low molecular weight chitosan is known to exert anti-tumor activities due to an enhancement in the natural killer activity of intestinal intraepithelial lymphocytes in sarcoma 180-bearing mice [84].

### 4.5. Anti-Oxidant and Anti-Inflammatory Activities

The oxygen radicals like hydroxyl, superoxide, and alkyl, as well as the stable DPPH radicals, are neutralized by chitosan and its derivatives, making them effective antioxidants. Chitosan and its derivatives are also known to block the oxidative chain reaction by serving as hydrogen donors. Additionally, it has been noted that the radical scavenging characteristics of chitosan vary with their degree of acetylation and molecular weight. It has been discovered that low molecular weight chitosan is more active than the ones with greater molecular weight. Low molecular weight (MW) chitosan samples (1–3 kDa) were shown to have a greater capacity to scavenge various radicals. Other tests have shown that at a concentration of 0.5 mg/mL, low molecular chitosan can scavenge more than 80% of superoxide radicals [29].

The formation of free radicals is closely linked to the inflammatory process, which is an inherent physiological reaction of the body to tissue injury. Although inflammation is essential for survival, it can harm an organism if it is exceedingly severe, is unable to eliminate the cause of the inflammation, or is aimed toward the host [3]. By stimulating the production of a wide range of pro- and anti-inflammatory cytokines, chemokines, growth factors, and bioactive lipids by innate immune cells, chitosan possesses potent immunostimulatory action [85]. This action again appears to be more apparent when the molecular weight of the chitosan is decreased and when the chito-oligosaccharides demonstrate higher activity [3]. Moreover, chitosan and its derivatives have been studied a lot for their anti-inflammatory and pro-inflammatory properties. Studies have demonstrated that chitosan therapy significantly reduces IL-10 and TNF- levels. Chitosan also inhibits NF-κB activation, preventing inflammatory reactions in the intestinal mucosa [86].

## 5. Modified Chitosan Preparation/Derivatives

Alkaline hydrolytic derivatives of chitin such as chitosan exhibit less crystallinity and better solubility profile in comparison to chitin. The presence of one amino group and two hydroxyl groups in chitosan opens up wide possibilities for chemical modification. The various chemical modifications such as acylation, alkylation, quaternization, oligomerization, carboxyalkylation, hydroxyalkylation, thiolation, phosphorylation, sulphation, graft copolymerization, and enzymatic modifications along with many such other modifications have been performed. Assorted modifications such as chitosan hybrids with cyclodextrins, dendrimers, sugars, and crown ethers have also been explored as novel multifunctional macromolecules [87]. These modifications do not alter the basic chitosan skeleton and also retain the original biochemical and physicochemical properties in addition to the new or improved properties (Figure 2). These derivatives with modified properties are further widely investigated for pharmaceutical, biomedical, and biotechnological applications [87].

The following are the important chitosan derivatives that have proven to be of significant importance in pharmaceutical applications and advanced research [89].

### 5.1. N-Carboxymethyl Chitosan (N-CM-Chitosan)

It is a widely used derivative of chitosan. This derivative is synthesized by the introduction of a carboxymethyl group onto chitosan’s parent structure. The reaction involves carboxymethylation of the amine and hydroxyl moieties of chitosan. A reductive alkylation reaction in the presence of glyoxylic acid leads to the formation of N-CM-chitosan. This reaction is carried out by treating chitosan with monochloroacetic acid and sodium hydroxide in isopropanol at 50 °C. sN-CM-Chitosan has enhanced solubility in neutral and alkaline conditions with no significant alteration in important characteristics such as biodegradability, biocompatibility, and antibacterial and antioxidant activities [90,91].

### 5.2. Hydrophobic Chitosan

By performing modifications using anhydrides and long-chain acyl chlorides, hydrophobic derivatives of chitosan can be prepared [92]. Hydrophobically modified chitosan derivatives tend to self-assemble in the aqueous medium to form nanoparticles or micelles which can be utilized as a promising drug delivery carrier [93].

### 5.3. Chitosan with Methoxyphenyl Functions

Methoxyphenyl aldehydes such as o-vanillin, syringaldehyde, vanillin, and veratraldehyde were found to undergo reaction with chitosan under normal as well as under reduced conditions to impart insolubility and other characteristics to chitosan [94].

### 5.4. Tyrosine Glucan

Tyrosine glucan is a modified chitosan synthesized with 4-hydroxyphenylpyruvic acid in the presence of the tyrosinase enzyme. This chitosan derivative is used for the preparation of stable and self-sustaining gels [95].

### 5.5. Highly Cationic Chitosan

Several derivatives of chitosan that are highly cationic have been prepared by using various reagents. N-acylation of chitosan with fatty acid (C6–C16) chlorides increases its hydrophobicity [96]. Tien and his colleagues demonstrated that N-acylation of chitosan with fatty acyl chlorides introduced hydrophobicity making it a suitable matrix for drug delivery [97].

### 5.6. Polyurethane Based Chitosan

Polyurethane-based chitosan has demonstrated a wide range of biomedical applications including its utility as tissue engineering scaffolds for wound healing and as drug delivery systems. The presence of NH_2_, NHCOCH-, NHCOO groups in chitosan, chitin, and polyurethane imparts bioactive properties which owe to their biomedical applications [98]. Barikani and his colleagues synthesized chitosan-based polyurethane elastomer dispersion by the reaction of isophorone diisocyanate and poly(ε-caprolactone), extended with various mass ratios of dimethylol propionic acid and chitosan. The combination of these polymers provides new materials with good mechanical and physical properties combined with degradability and bioactivity [99]. A study conducted by Mahanta and his co-workers demonstrated the applicability of polyurethane-grafted chitosan copolymer for sustained and controlled drug delivery. Biocompatibility including hemocompatibility of this newly developed material was demonstrated through various studies, and this novel copolymer is appropriate for tissue engineering and drug delivery [100].

### 5.7. Hydroxyalkyl Chitosan

Chitosan reacts with epoxides (such as butylene oxide, ethylene oxide, and propylene oxide) and glycidol leading to the formation of hydroxyalkyl chitosan. Hydroxyalkylation occurs either at -OH or -NH_2_ group, giving rise to O-hydroxyalkyl or N-hydroxyalkyl chitosan or a combination of both, thereby enhancing the biopolymer’s solubility [101]. Li et al. synthesized hydroxy butyl chitosan by modifying the hydroxy butyl group on the chitosan. It exhibited significant hygroscopicity and moisture retention ability [102].

### 5.8. Sugar Modified Chitosan

Sugar-modified chitosan is prepared with the intention to enhance the aqueous solubility of chitosan. Hall and Yalpani reported the different modifications of chitosan with sugars in 1980. They attached lactose to the 2-amino functional group of chitosan. The specific attachment of carbohydrates to the 2-amino functional group of chitosan transforms this linear, water-insoluble polymer into water-soluble derivatives with branched chains [103,104]. Park and his co-workers prepared chitosan derivatives by covalently attaching gluconic acid, a hydrophilic sugar moiety, via the formation of an amide bond with chitosan. N-Acetylation on the prepared sugar-modified chitosan derivative was further carried out to improve the water solubility, as it has been reported that chitosan’s crystallinity is responsible for its poor water solubility, which is in turn dependent on the degree of N-acetylation [105].

### 5.9. Dendrimer Chitosan Hybrid

Dendrimers are high molecular weight, monodisperse macromolecules with a well-defined chemical structure. They offer a wide range of possibilities in molecular design due to their multifunctional properties such as biomedical applications, etc. Various researchers have reported chemical modification of chitosan with dendrimers [106]. Sashiwa et al. developed chitosan-dendrimer hybrids with varying functional groups such as ester, poly (ethylene glycol) and carboxyl groups using dendrimer acetal by reductive N-alkylation [107]. In a study, they synthesized poly (amido amine) (PAMAM) dendrimers with tetramethylene glycol spacer and attached it to chitosan by reductive N-alkylation [106]. Similarly, Tsubokawa et al. reported another modification of chitosan’s surface by grafting hyperbranched dendritic polyamidoamine. The modification was carried out in two steps: (1) Michael addition of methyl acrylate to amino groups on the surface and (2) amidation of the resulting esters with ethylenediamine to yield polyamidoamine dendrimer-grafted chitosan powder [108].

### 5.10. Cyclodextrin-Linked Chitosan

Cyclodextrin (CD)-linked chitosan derivatives are developed to combine the desirable characteristics of chitosan with the ability of CD to form non-covalent inclusion complexes using several guest molecules that in turn would alter their physicochemical properties [87]. They have gained a wide variety of interests in cosmetics, drug delivery, and analytical chemistry [109]. CD can be linked to chitosan by different means. Auze’ly-Velty and his colleagues prepared a mono-substituted â-CD derivative that possesses a reducing sugar on the primary face and subsequently subjected it to reductive amination. They further characterized the prepared derivatives in terms of purity and chemical integrity using light scattering and high-resolution NMR [110]. Similarly, Cheng and Wang synthesized β-CD grafted with chitosan upon reaction with p-toluene sulfonyl chloride. Further, the inclusion complex was studied for its inclusion [111]. Venter et al. explored the mucoadhesive property of β-CD grafted chitosan chain polymer. The synthesized chitosan-CD-polymer demonstrated characteristics of a potential mucoadhesive drug delivery system with some inclusion properties obtained from β-CD [112] Gaetano et al. developed levofloxacin loaded sulfobutyl-ether-β-CD based chitosan nanoparticle (CH/SBE-β-CD NPs) for the treatment of ocular infections. The developed formulation was evaluated for encapsulation efficiency, morphology, size, and zeta potential. The study results demonstrated that the developed NPs were homogenous in size and have positive zeta potential values. In vitro study results displayed significant antibacterial activity against both gram-positive and gram-negative bacteria thus demonstrating it as a useful system for treating ocular infections [113]. In another study, CD-based chitosan nanospheres (CS NPs) were developed and evaluated as a potential carrier for nose-to-brain targeting of idebenone [114].

## 6. Applications of Chitosan

Chitosan and chitosan-based systems have been shown potential applications in drug delivery as well as biomedical field (Figure 3).

### 6.1. Drug Delivery Applications

Drug delivery is a crucial aspect of modern medicine as it determines the efficacy and safety of a therapeutic substance. Targeted drug delivery within the body is a complex process that requires overcoming several barriers, such as the removal of drugs by the liver/kidneys, rapid degradation in the bloodstream, and limited permeation across biological membranes. A suitable carrier or vehicle is necessary to protect the drug from degradation, increase its circulation time, and enhance its localization at the target site. The ideal drug delivery vehicle should possess several key attributes, such as biocompatibility, biodegradability, and controlled release characteristics [115]. Biocompatibility ensures that the delivery vehicle does not produce adverse effects on the body, while biodegradability ensures that the delivery vehicle can be safely metabolized and eliminated from the body after its function is complete. Controlled release properties ensure that the drug is released in a controlled manner over a defined period of time, allowing for sustained therapeutic levels of the drug and reducing the frequency of dosing. Chitosan has shown great promise for drug delivery applications due to its unique properties. On account of its outstanding biocompatible and biodegradability, this polymer has been classified by the FDA as “Generally Recognized as Safe” (GRAS) and is approved for use in tissue engineering and drug delivery applications [116]. The LD50 of chitosan in oral administration to mice is more than 16 g/kg, and results from multiple acute toxicity studies have established chitosan to be safe for systemic use [117]. The repeating glycosidic residue in chitosan has a primary amino group that acts as a site for chemical bonding and gives the molecule a positive charge. This positive charge gives chitosan pH sensitivity, allowing it to form ionic complexes with drugs and control drug release. One of the key benefits of chitosan is its versatility, as it can be modified into various dosage forms, each with its own unique properties and applications. Some widely used chitosan-based delivery systems have been discussed in the following sections (Table 1).

#### 6.1.1. Oral Delivery

Oral drug delivery involves the administration of medication through the mouth, usually in the form of tablets, capsules, or liquids, to be absorbed by the digestive system and eventually into the bloodstream. It is the most frequent and convenient routes of drug administration, as it is non-invasive and easily accessible. It is often the most practical and preferred route for patients, as it can be self-administered and does not require healthcare professionals. Additionally, oral delivery permits for the sustained release of medication over prolonged time, which can increase patient compliance and improve therapeutic outcomes. The extensive mucosal surface area of the gastrointestinal tract facilitates drug absorption, making it an exceedingly favored route for administering drugs. Many drugs cannot be administered orally owing to their susceptibility to degradation by gastrointestinal secretions, including enzymes, and the variable pH conditions. Physiological factors like transit time, blood perfusion, the presence of food, and limited absorption windows for several drugs can further hinder their absorption when administered orally [125]. Chitosan possesses favorable properties for use in oral drug delivery systems. Its cationic properties allow it to interact easily with negatively charged cell membranes, enhancing cellular uptake. Unlike many water-insoluble polymers, chitosan can partially dissolve in water, forming a hydrogel in aqueous environments. This unique quality enables it to adhere to mucosal surfaces, making it an excellent choice for oral drug delivery. Additionally, some chitosan derivatives, such as glycol chitosan solution or glycol chitosan-based nanoparticles, can inhibit the P-glycoprotein efflux pump (increasing the likelihood of absorption into the bloodstream) [126]. Some interesting applications of chitosan-based systems for oral delivery are discussed below.

N-cinnamyl substituted O-amine functionalized chitosan (cinnamyl-chitosan) was examined as a novel excipient by Ren et al. [127]. Directly compressed tablets were prepared using microcrystalline cellulose (as binder/filler), magnesium stearate (as a lubricant), acetaminophen (as drug substance), and either chitosan or the modified cinnamyl-chitosan. As the contents of other ingredients were kept the same for all formations, any difference in terms of flowability, compression behavior, and disintegration characteristics of the tableting blends was due to the chitosan type. While the plastic behavior of the 20% cinnamyl-chitosan blend was similar to the control group, their larger particle size provided them with higher elasticity, as evident by the result of the force-displacement profile analysis. This resulted in easy deformation of cinnamyl-chitosan, facilitating high compaction of the blend at relatively low levels of compressive pressures. With bigger particle sizes, cinnamyl-chitosan mixtures display good flowability and compaction characteristics, devoid of tableting defects. In comparison to MCC, they also exhibit good acetaminophen release properties as excipients. In addition to superior mechanical strength and drug delivery performance, the chemically modified cinnamyl-chitosan had enhanced antibacterial and antioxidative effects, presenting it as a possible alternative excipient for direct compression. In a similar study, Yasufuku et al. utilized modified low molecular weight chitosan’s antioxidant properties to create chitosan tablets for oral consumption with extended-release capabilities. The chitosan exhibited scavenging activity in response to hydroxyl and superoxide radicals’ oxidative stress, which allowed it to endure breakdown and gradually release theophylline (a model drug) over a duration exceeding 12 h in a controlled manner [128].

Patta et al. reported a novel polyionic particulate system based on chitosan-N-arginine having pH-responsive and mucoadhesive particles. Colloidal particles formed when ionized alginate and protonated chitosan containing chemically bound arginine interacted. At a chitosan-arginine monomer to alginate monomer ratio of 1.6, the biopolymer was highly charged, providing electrostatic interaction with Gibbs energy compensation of around 14 kcal/mol. The presence of hydrophobic drugs praziquantel or ivermectin did not hinder particle formation. When the pH changes, the bioparticle’s size and surface charge both change in response (as a result of protonation or deprotonation state of both polyelectrolytes), resulting in structures with nano- to micro-hydrodynamic diameters. As a result, a pH-responsive structure is demonstrated, creating ideal circumstances for interaction with the digestive tract. After freeze-drying, the bioparticles had an irregular shape and surface morphology but a consistent dry structure overall. Three species of teleost fish exhibit great compatibility with the particles in the in vivo experiments, and the macromolecule content of the particles has mucoadhesive properties that extend the retention of the structures in the intestinal mucosa (Figure 4A) [129].

Pellá et al. reported hybrid microgels based on modified chitosan and SiO_2_ nanoparticles (added to act as “spacers” and provide reinforcements to the structure of the microgels), synthesized by emulsion polymerization. The addition of SiO_2_ nanoparticles resulted in microgels with a smaller hydrodynamic diameter of 11.3 ± 8.07 μm compared to those without nanoparticles (which had a diameter of 18.7 ± 12.3 μm). The microgels were optimized using factorial design and showed pH responsiveness. The entrapped vitamin B12 displayed an initial burst release at pH 1.2 and then a slower sustained release, which was due to the gradual weakening of the microgel’s chemical bonds. This platform is promising for oral drug delivery systems, especially for treating gastric wounds, as it allows for the fast release of high drug concentrations for immediate pain relief, followed by a controlled release that facilitates the gradual recovery of the damaged area [130]. Ryu et al. reported a mussel-inspired chitosan–catechol adhesive patch (Chitoral), which can serve as an oral mucosal drug delivery system. The platform is based on a chitosan backbone modified with hydro-caffeic acid (a carboxyl-terminated catechol). Freeze-drying of this modified chitosan solution creates a porous, sponge-like adhesive material in which therapeutic drugs could be loaded within the polymeric network. When the platform comes into contact with saliva, it instantaneously dissolves and forms inter-molecular complexes with oral mucins, quickly turning into an insoluble, adhesive hydrogel-like substance by the combined activities of covalent crosslinking and physical entanglement. Lap-shear adhesion experiments on the porcine tongue and labial mucosa revealed a detachment stress of 10.3 ± 6.1 kPa. Triamcinolone acetonide-loaded Chitoral accelerated the healing of severe oral ulcers by facilitating a sustained release of encapsulated drugs [131].

Electrospinning is a useful technique for creating quick-dissolving oral films, but producing nanofibers from pure chitosan can be difficult owing to its polycationic properties in solution, rigid structure, and intra-molecular interactions. Qin et al. inspected the possibility of improving the electro spinnability of chitosan by blending it with pullulan, an extracellular polysaccharide derived from *Aureobasidium pullulans*, which can increase solution viscosity, reduce conductivity, and lower surface tension. To prepare the chitosan/pullulan blend for electrospinning, separate solutions of pullulan and chitosan were made by dissolving their powders in 50% (*v*/*v*) aqueous acetic acid at various concentrations. The blend was then created by mixing these solutions in varying ratios while keeping the total polymer content fixed at 10% *w*/*v*. After stirring for 12 h at room temperature and degassing for an additional hour, the solutions were electrospun. The ratio of chitosan to pullulan had an impact on both the properties of the solution and the morphology of the resulting nanofibers. Specifically, as the amount of chitosan increased, the viscosity and conductivity of the solutions also increased. Scanning electron microscopy (SEM) revealed that the diameter of the nanofibers reduced initially followed by an increase. Hydrogen bond interactions between chitosan and pullulan molecules were indicated by Fourier transform infrared spectra, whereas X-ray diffraction analysis confirmed that the electrospinning process reduced the overall crystallinity of the materials. Finally, wetting studies demonstrated that the material completely dissolved within 1 min, and aspirin encapsulation tests confirmed the potential use of the material in the oral mucosal release. (Figure 4B) [132].

**Figure 4 pharmaceutics-15-01313-f004:**
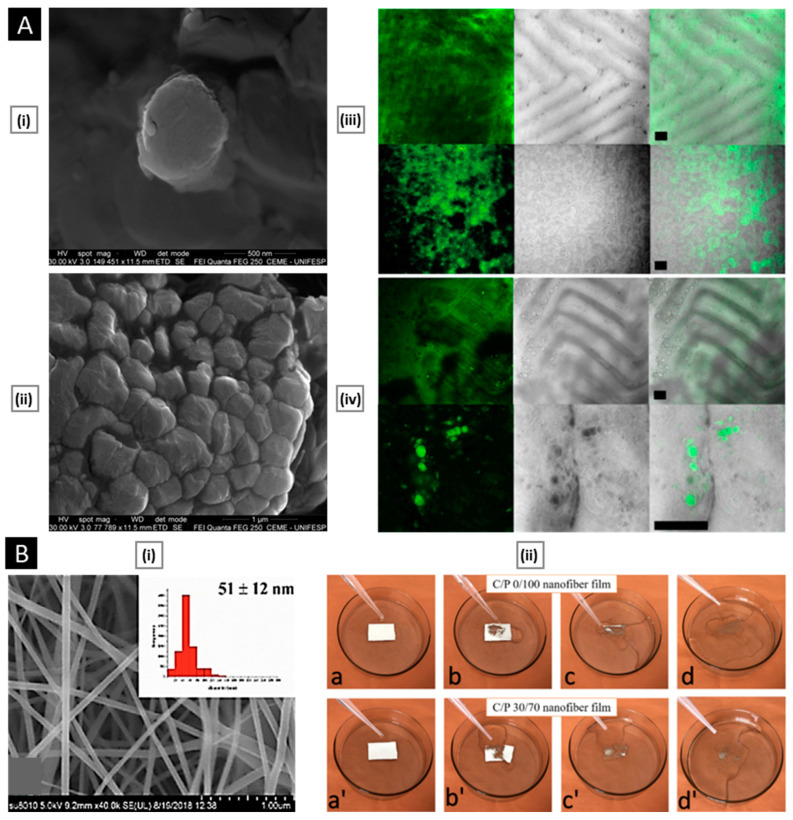
((**A**), i) SEM micrographs of isolated drug-free chitosan–arginine–alginate bioparticles after lyophilization, showing globular morphology and smooth to a slightly irregular surface. ((**A**), ii) SEM micrographs of particle cluster containing 38 μM praziquantel unveiling irregular polygonal-like structures and wrinkled surface. Confocal laser scanning micrographs (epifluorescence of labeled fluorescein isothiocyanate, phase contrast and merge) of anterior (upper panels) and posterior (lower panels) intestine of Carassius auratus after 30 min ((**A**), iii) and 8 h ((**A**), iv) (Scale bar: 50 μm). Reproduced with permission from [129], copyright Elsevier 2020. ((**B**), i) SEM images and diameter distributions of electrospun nanofiber (30/70 weight ratio of chitosan to Pullulan) ((**B**), ii) Presentation of solubility behavior of C/P 0/100 nanofiber film and C/P 30/70 nanofiber film. The photos were taken before the film was in contact with water (a, a’) and after the film was in contact with water for 5 s (b, b’), 30 s (c, c’), and 60 s (d, d’). Reproduced with permission from [132], copyright Elsevier 2019.

#### 6.1.2. Ophthalmic Delivery

Ocular drug delivery presents a significant challenge in pharmaceutical research, mainly due to the unique pharmacokinetic and pharmacodynamic properties of the eye. The eye’s distinctive structure restricts the entry of drug molecules, making it hard to maintain an effective drug concentration at the required site of action. To achieve therapeutic concentrations of active agents in oral treatment for ocular disorders, substantial doses of active agents are required which may cause severe side. In addition, systemic access is impeded by the blood–retinal, blood–aqueous, and blood–vitreous barriers [133]. Therefore, topical administration of ophthalmic dosage forms remains the only practical option. Formulations like drops, gels, and creams are available to treat surface disorders (such as conjunctivitis, blepharitis, and keratitis sicca) as well as to deliver intraocular therapy through the cornea (for diseases such as glaucoma or uveitis) [134]. However, poor pre-corneal retention, poor penetration across the cornea, and efficient drug elimination mechanisms, such as tear drainage, protein binding, and induced tear production, limit the ocular bioavailability of topical preparations to less than 5% [135]. Attempts have been made to address the issue of poor bioavailability and the consequential repeated drug administration by focusing on enhancing corneal residence time by employing viscosity enhancers, mucoadhesive polymers, and in situ gel-forming systems. Chitosan inherently possesses all these attributes. Its biocompatibility, anti-bacterial nature, pH neutrality, and ability to undergo degradation in response to lysozyme (also known as muramidase which is an enzyme found in tears that is part of the innate immune system) make it a potential candidate to develop controlled release ocular platforms [136].

Nanoparticles are the most extensively studied among chitosan-based delivery systems for the eyes. They are typically produced through ionic gelation, in which tripolyphosphate (TPP) is employed as a cross-linking agent. During this process, the positively charged amino groups are utilized as they react with the negatively charged TPP groups, resulting in a decrease in mucoadhesive capability [137]. Dubashynskaya et al. reported mucoadhesive cholesterol–chitosan self-assembled particles for topical ocular delivery of dexamethasone (Dex). To synthesize these particles, the researchers utilized a carbodiimide-mediated coupling reaction with a succinyl linker to create cholesterol–chitosan conjugates. They then optimized the synthesis by adjusting the molar ratios of reactants to produce cholesterol–chitosan conjugates with varying degrees of substitution (ranging from 1.2 to 5.8%). The resulting submicron particles had hydrodynamic diameters of 700–900 nm and a ζ-potential greater than 30 mV, which provided enough repulsive force to ensure optimal physical colloidal stability and superior mucoadhesive properties. The Dex-loaded particles were determined to be non-toxic, had membrane-stabilizing properties, and demonstrated sustained anti-inflammatory activity [138].

Rather than preparing entire particles out of chitosan, its solution can be added to other nanoparticles that have actually been made or are in the process of being developed to export some of its advantages [139]. Eid et al. developed chitosan-coated niosomes for enhancing the precorneal residence period, eye permeation, and bioavailability of azithromycin (for the treatment of bacterial conjunctivitis). Drug-loaded niosomes were prepared through modified thin-film hydration strategy, followed by coating with chitosan solution. The optimized particles had a mean diameter of 376 nm, entrapment of 74.2%, the surface charge of 32.1 mV, and a muco-adhesion force of 3114 dyne/cm^2^. Compared to commercial drops, the platform displayed a three-fold increase in azithromycin concentration in the rabbit eyes, attributed to the enhanced permeability coefficient of chitosan-coated niosomes and sustained release of entrapped drug [140]. In a similar study, Li et al. investigated the transport mechanism of three surface-modified chitosan derivatives nanostructured lipid carriers (NLCs) to the anterior chamber via the cornea. Chitosan-N-acetyl-L-cysteine, chitosan oligosaccharides, and carboxymethyl chitosan were the derivatives employed for NLC surface decorating. In vivo precorneal retention studies on rabbits demonstrated that all coated formulations were more resistant than solution and uncoated NLCs. Upon comparison, the carboxymethyl chitosan decorated NLCs, chitosan-N-acetyl-L-cysteine, and chitosan oligosaccharides surface decorated NLCs demonstrated greater penetration across the whole corneal epithelial barrier, lower conjunctival to corneal permeability ratio, and higher bioavailability. The poor impact of carboxymethyl chitosan was attributed by the authors to its time limited hydrogel effect. Permeability of the particles into the whole cornea and their transport towards the anterior segment was visualized and confirmed by fluorescence microscopy ex vivo fluorescence imaging, respectively (Figure 5A). Chitosan-N-acetyl-L-cysteine surface decoration exhibited a significantly slower drug release than the other two coatings, likely because of the creation of disulfide bonds within the coating, which impeded the drug’s diffusion [141].

Superior therapeutic outcomes are observed with in situ hydrogels and film/membrane-based ocular delivery systems. When instilled into the eye, a typical in situ gelling system is initially in a liquid state. However, in response to various stimuli, it rapidly transforms into a viscoelastic gel within the conjunctiva sac of the eye. This results in a significantly prolonged residence time of the gel, which gradually releases the encapsulated drugs over an extended period of time. As a result, this sustained release provides improved bioavailability of the drug and reduces the frequency of administration. Various types of in situ gelling systems have been proposed as novel ophthalmic formulations, including those that are temperature-, pH-, and ionic-activated [142]. Shi et al. reported a system for enhancing ocular bioavailability using hexanoyl glycol chitosan (H-GCS) that gels in situ and is sensitive to temperature. The aqueous H-GCS solution undergoes a typical sol-gel transition at 32 °C, and the incorporation of levofloxacin does not seem to affect its gelation behavior. At concentrations below 0.8 mg/mL, H-GCS shows minimal cytotoxicity to L-929 and HCEC cells after 24 h incubation. Furthermore, topical application of 2 wt% H-GCS hydrogel does not cause eye irritation in rabbits after a single instillation. Upon comparison with an aqueous solution, 2 wt% H-GCS hydrogel remarkably prolongs the retention of drugs in the precorneal region after topical application, leading to increased ocular bioavailability [143].

Li et al. reported micro-sized drug loaded chitosan films. For enhancing the intractable aqueous solubility of chitosan, the authors dissolved chitosan in ionic liquid, followed by overnight freezing at −20 °C and subsequent solvent exchange with plain water at room temperature. This generates a plain water-based chitosan solution with a concentration of about 2.4 mg/mL. After the addition of brimonidine tartrate (an antiglaucoma drug) at a concentration of 1 mg/mL, the chitosan solution was cast into a rounded polytetrafluoroethylene mold (with a diameter of 32 mm) and air-dried to generate drug-loaded chitosan film. The obtained film was structurally stable, was transparent, and had mucoadhesive properties. The loaded drug formed micro-sized crystals that were evenly distributed throughout the chitosan film. To evaluate the corneal permeability of brimonidine tartrate in the chitosan film, the researchers used a Franz diffusion cell system. The results showed that more than 75% of the drug was released within 6 min, and about 90% was released in less than half an hour. The corneal permeability of the drug was determined to be 1.62 × 10^−5^ cm/s, which is much higher than that of many previous systems [144]. Bao et al. developed a hydrogel film using a mold casting method composed of Dex-glycol chitosan and oxidized hyaluronic acid. Levofloxacin was encapsulated into the hydrogel films to achieve dual ophthalmic delivery. As the degree of oxidation of the hyaluronic acid increased, the swelling ratio of the hydrogel film decreased. The optimized hydrogel films exhibited a dense cross-section and smooth surface. The platform facilitated a stepwise release of Lev and Dex, with Lev released rapidly, followed by a sustained release of Dex. In vitro studies showed that the hydrogel films were noncytotoxic to human corneal endothelial cells and had a potent capacity to inhibit bacterial growth in *E. coli* and *S. aureus*. Additionally, the fabricated blank hydrogel films demonstrated noteworthy anti-inflammatory activity through the downregulation of several inflammatory cytokines in lipopolysaccharide-activated RAW264.7 macrophages. Following topical administration in rabbit eyes, the hydrogel films displayed biocompatibility and good ocular tolerance [145].

#### 6.1.3. Transdermal Delivery

The skin, being the body’s largest and most expansive organ, is a highly advantageous avenue for administering numerous medications. Compared to intravenous and oral delivery methods, transdermal administration offers a variety of benefits, including avoidance of first-pass metabolism, consistent blood levels, and both localized and systemic delivery. It is also suitable for treating a wide range of conditions, such as chronic pain, hormonal imbalances, and cardiovascular diseases [146]. However, the effectiveness of transdermal delivery is often impeded by the stratum corneum. It is composed of keratin filaments enclosed in a cornified envelope and covered by a multilamellar lipid bilayer, which acts as a physical barrier to drug molecules. Despite the commercial availability of several transdermal treatments (such as patches, ointments, spreads, and sprays), drawbacks like poor bioavailability, low drug penetration, undesirable side effects (such as skin irritation), and poor patient compliance remain a concern [147]. To address these issues, researchers are investigating the use of synthetic high-molecular-weight polymers such as PEG and PLA due to their excellent stability and adhesion, but their degradation inertness, cytotoxicity, and resulting immunogenicity limit their applications. In such cases, the ability of chitosan (and its derivatives) to promote drug penetration can be leveraged [148]. The underlying mechanism involves the alteration of the secondary structure of stratum corneum proteins, opening of tight junction in the granular layer by redistribution of transmembrane proteins, and enhancing the stratum corneum water content (to facilitate delivery of hydrophilic drugs) [149].

With the aim to promote effective transdermal delivery, Shekh et al. developed a novel scaffold carrier by functionalizing polyacrylonitrile nanofibers with oxidized chitosan followed by grafting of antiviral drug acyclovir onto the nanofibers to achieve a sustained and controllable release profile. The modification of oxidized chitosan was verified by SEM, where the average diameter of nanofibers increased from 218 to 354 nm, with increased surface roughness. In vitro drug release study showed that the acyclovir nanofibers modified by oxidized chitosan exhibited a more sustained release curve (following the Korsmeyer-Peppas model with a Fickian diffusion mechanism) than acyclovir nanofibers without modification. This behavior may be explained by the formation of Schiff bases between the drug molecules on the nanofibers and the oxidized chitosan. As a means of managing the toxicity of long-acting antiviral drugs like acyclovir, oxidized chitosan-modified nanofibers can help bring about the desired decrease in the drug’s immediate high local concentration. The examination of blood combability and human adipose-derived stem cells revealed that the produced scaffold had good hemocompatibility and cell biocompatibility (Figure 5B) [150].

Anirudhan et al. fabricated a chitosan/hyaluronic acid-based transdermal film device for the controlled release of lidocaine. The chitosan was modified with glycidyl methacrylate and butyl methacrylate to enhance skin adhesion and mechanical properties, while hyaluronic acid was covalently linked with 3-(dimethylamino)-1-propylamine to encapsulate the poorly water-soluble lidocaine. The resulting film was transparent, colorless, and aesthetically pleasing. The researchers investigated the effect of storage time on the mechanical properties of the device and discovered that the tensile strength increased from 0.5896 to 0.5942 MPa, while the elongation of break reduced from 626 to 439% after 60 days of storage. The platform demonstrated excellent control over the release of lidocaine due to the improved hydrophobicity of the film matrix. The device also exhibited high storage stability, as the release profile after 30 and 60 days of storage was almost identical to that of a fresh patch. In an in vivo skin adhesion test, the device showed a peeling force of 0.960 N/cm^2^ and left no greasy residue upon removal from the skin. Additionally, the film did not cause any skin irritation even after 24 h of application on human skin [151].

Indulekha et al. developed a transdermal drug delivery system that responds to heat and pH to enable on-demand drug delivery. This system involved grafting pH-responsive chitosan onto thermos-responsive poly (N-vinyl caprolactam) to create a gel-like substance. The aim was to create a patient-friendly platform that allows patients to self-administer drugs by applying a heating pad over the transdermal drug delivery system. To test the system’s effectiveness, the researchers loaded two model drugs, acetaminophen and etoricoxib, and conducted in vitro drug release experiments at three different temperatures (25, 32, and 39 °C) and two different pH levels (5.5 and 7). The results showed that drug release was highest for both drugs at 39 °C and pH 5.5. Moreover, in vitro skin permeation studies using a Franz diffusion cell revealed that both drugs exhibited increased drug release when the skin was exposed to a higher temperature [152]. In a comparable investigation, Radwan-Pragłowska et al. reported on the development of crosslinked chitosan-based patches that were functionalized with ZnO nanoparticles, with the aim of facilitating controlled delivery of cannabidiol. The resulting materials demonstrated exceptional porosity, as demonstrated by SEM and fluorescent microscopy analyses, as well as excellent swelling properties. The incorporation of zinc oxide nanoparticles into the patches led to improvements in their physicochemical properties, including mechanical durability, drug loading capacity, and prolonged drug release in accordance with a first-order kinetics mechanism. Moreover, the nanocomposites exhibited conductive properties, which render them suitable for use in iontophoresis. The prepared materials were evaluated for cytotoxicity using both indirect and direct methods (XTT assay and morphology study, respectively), both of which confirmed the good biocompatibility of the transdermal systems with skin cells (L929 mouse fibroblasts) [153].

Chitosan can be fabricated into nanoparticles having highly tunable size and surface charge, which enables them to have extraordinary specific surface area and great permeability in the skin. Several nanoparticle-based systems have been looked into for the treatment of skin-related ailments. Riezk et al. conducted an investigation into the potential application of chitosan nanoparticles as a drug delivery system for the management of cutaneous leishmaniasis. The researchers prepared two different types of chitosan nanoparticles: one that was positively charged using tripolyphosphate sodium and another that was negatively charged with dextran sulphate. When Amphotericin B (AmB) was loaded onto both types of nanoparticles, they demonstrated in vitro activity against Leishmania major intracellular amastigotes that was comparable to that of unencapsulated AmB. However, both types of nanoparticles exhibited significantly lower toxicity to KB-cells and red blood cells. In murine models of CL caused by L. major, intravenous delivery of chitosan-tripolyphosphate sodium nanoparticles (with a size of 69 ± 8 nm and zeta potential of 25.5 ± 1 mV, administered at a dose of 5 mg/kg on alternate days for 10 days) showed significantly higher efficacy than AmBisome^®^ (administered at a dose of 10 mg/kg on alternate days for 10 days) in terms of reducing lesion size and parasite load, as measured by both bioluminescence and qPCR [154].

Fereig et al. reported tacrolimus-loaded chitosan nanoparticles for the treatment of psoriasis. The authors reported a scalable, toxic solvent-free, and modified ion gelation technology for the incorporation of hydrophobic drugs in chitosan. Upon drug loading, the particles had a size of 140.8 nm ± 50.0 with an entrapment efficiency of 65.5% ± 1.3. In the course of an in vitro study on skin permeation, the researchers found that after 24 h, the permeability of tacrolimus in chitosan nanoparticles was 24%, while the permeability of tacrolimus cream was 61%. This indicates that the chitosan nanoparticles could significantly delay the release of tacrolimus, potentially reducing the systemic toxicity of the drug. Moreover, the 24 h skin deposition rate of tacrolimus cream was 11.4%, whereas that of the tacrolimus chitosan nanoparticles reached 75%, thereby demonstrating the potential of the nanoparticles for delivering tacrolimus to the skin. Psoriasis is a condition that causes skin sclerosis at the lesion site, which greatly impedes the transdermal delivery of drugs. The chitosan nanoparticles were found to improve the transdermal permeability of tacrolimus, delay its release from the cortex, prolong its action time at target sites, and reduce the impact of the drug on the entire body [155].

To treat inflammatory skin conditions, Barbosa et al. created fucoidan/chitosan nanoparticles for the topical administration of methotrexate. The researchers screened different ratios of fucoidan:chitosan (F/C) and found that the drug loading was relatively high (>80%). The methotrexate-loaded nanoparticles had a size in the 300–500 nm range, and the 1F/1C ratio nanoparticles had a positive zeta potential of +60 mV, while the 3F/1C and 5F/1C nanoparticles had negative surface charges of −40 and −45 mV, respectively. The loading of methotrexate in the 3F/1C and 5F/1C nanoparticles did not affect cell viability and showed lower cytotoxicity than free methotrexate in fibroblasts and human keratinocytes. Furthermore, the fucoidan/chitosan nanoparticles led to a significant reduction in proinflammatory cytokines produced by activated human monocytes. Skin permeation studies demonstrated that the methotrexate-loaded nanoparticles permeated the pig ear skin barrier, with a 2.7- and 3.3-fold increase for 3F/1C and 5F/1C nanoparticles, respectively, after 6 h compared to free drug [156].

**Figure 5 pharmaceutics-15-01313-f005:**
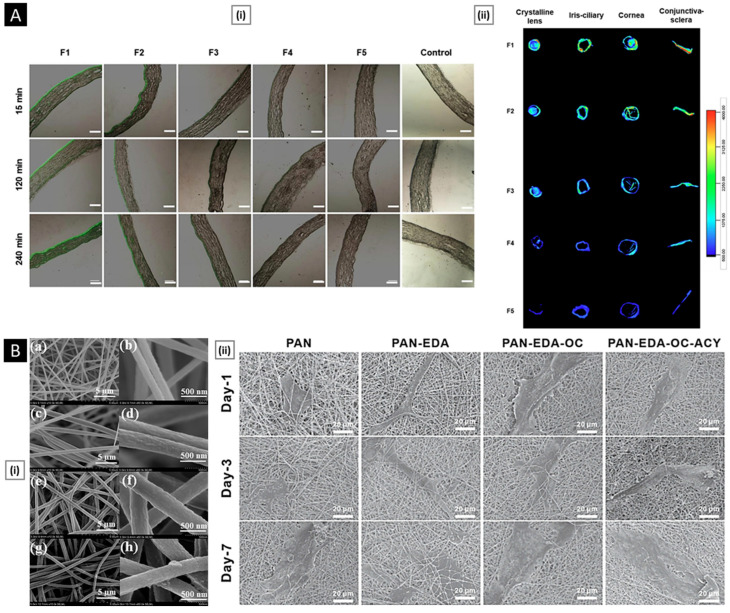
((**A**), i) Inverted fluorescence microscope micrographs after time-coursed in vivo corneal permeation of the preparations, the eye treated with normal saline was used as control. (Scale bar: 150 μm). ((**A**), ii) Ex vivo fluorescence imaging of rabbit ocular tissues from rabbits treated with different Coumarin 6-loaded formulations. Here, F1 = chitosan-N-acetyl-L-cysteine-coated NLC, F2 = chitosan oligosaccharide-coated NLC, F3 = carboxymethyl chitosan-coated NLC, F4 = Uncoated NLC, F5 = Coumarin 6 eyedrops. Reproduced with permission from [141], copyright Elsevier 2017. ((**B**), i) SEM images of polyacrylonitrile modified nanofibers at different magnifications: PAN (a) and (b); PANEDA (c) and (d); PAN-EDA-OC (e) and (f); PAN-EDA-OC-ACY (g) and (h). ((**B**), ii) Cell attachment of HASCs on modified fibrous scaffolds. Reproduced with permission from [150], copyright Elsevier 2020.

#### 6.1.4. Nasal Delivery

Nasal delivery of drugs refers to the administration of medication through the nasal cavity. This method takes advantage of the rich blood supply present in the mucous membranes of the nasal cavity to facilitate the rapid absorption and distribution of drugs throughout the body. Historically, the nasal route has been utilized to administer topical formulations for the management of local diseases in the upper respiratory tract, like nasal allergies, infections, and congestion. More recently, the nasal route has been increasingly used for the systemic delivery of small molecular weight drugs to treat chronic diseases such as diabetes and obesity. In particular, researchers are exploring the potential of the nasal route for the management of chronic neurodegenerative conditions of the central nervous system (CNS), including Parkinson’s disease and Alzheimer’s disease, as well as for faster treatment of disorders such as migraine and convulsions [157]. The nasal route offers advantages such as avoidance of first-pass metabolism, elimination of needle phobia associated with the parenteral route, and possible self-administration, leading to improved patient convenience. However, the nasal route is not without its challenges. The high nasal secretion rate can limit the residence time of drugs in the nasal cavity, and the volume of an administered formulation is often restricted. Furthermore, long-term localized treatment can result in damage to the nasal cavity. Additionally, only highly lipophilic molecules have optimal bioavailability when administered nasally [158]. To address these issues, researchers are developing strategies to improve the residence time of drugs in the nose, improve nasal adsorption, and modify the physicochemical properties of drugs. One promising approach involves the use of chitosan, which has positively charged amino groups that interact with anionic counterparts present in the mucous layers, mainly sialic acid. This interaction can affect the permeability of the epithelial membrane, resulting in chitosan’s ability to retain a formulation for longer periods in the nasal cavity and enable the transport of hydrophilic molecules across the membrane [159]. In addition, chitosan’s muco-adhesion and modulatory effect on mucociliary clearance have led to the development of several interesting platforms for nasal delivery.

Gholizadeh et al. developed chitosan-based thermosensitive hydrogel using β-glycerophosphate as a gelling agent for nose-to-brain delivery of ibuprofen (IBU). IBU is a commonly used non-steroidal anti-inflammatory drug that has been shown to extend the onset of Alzheimer’s disease. Though, the amount of IBU that reaches the brain when taken orally or intravenously is limited due to the blood–brain barrier and its low solubility. To overcome these limitations, the researchers developed a nasal spray that can be easily applied and undergoes a phase change from a liquid to a semi-solid with mucoadhesive properties when exposed to physiological temperatures. The spray rapidly gels in 4–7 min at 30–35 °C, which is about 2.9 times faster than the average mucociliary clearance rate of the human nasal cavity (around 20 min). At room temperature, the liquid formulation exhibits shear-thinning behavior, which helps generate fine droplets when sprayed with a nasal spray pump. The use of low molecular weight chitosan (110–150 kDa) produced an optimal spray with a smaller droplet size (Dv50), wider spray area, and higher surface area coverage (evaluated using the laser diffraction method). The formulation also enhances the solubility of ibuprofen by approximately 100 times its intrinsic aqueous solubility without using any organic solvent. In vitro experiments demonstrated that the formulation can accelerate the transport of IBU across nasal epithelial cells by reversibly modulating tight junctions [160]. In a similar study, Qi et al. reported curcumin-loaded guanidine–chitosan thermo-sensitive hydrogel in order to bypass the blood–brain barrier and promote the absorption of curcumin for the management of depression via nasal delivery. Hydroxypropyl-β-CD was employed as an absorptive accelerator for hydrogel formulation, and it also acted as the host molecular to form a complex with curcumin [161].

Trapani et al. developed nanoparticles based on an N-O-carboxymethyl chitosan-dopamine amide conjugate for nose-to-brain dopamine delivery as a treatment for Parkinson’s disease. The nanoparticles were prepared via the nanoprecipitation method and resulted in spherical particles with a bimodal size distribution and a particle size of 289.0 ± 50.0 nm, with a zeta potential of −32.4 ± 1.60 mV. The study demonstrated an encapsulation efficiency of dopamine of 89.0 ± 2.0%. Solid-state analysis using FT-IR, DSC, and thermogravimetric methods revealed that the particles were amorphous and thermally stable, making them potentially suitable for use in pharmaceutical preparations. X-ray photoelectron spectroscopy analysis confirmed the amide bond between dopamine and carboxymethyl chitosan and demonstrated that there was no free dopamine on the surface of the nanoparticle. In addition, cell-viability tests on olfactory ensheathing cells demonstrated only slight cytotoxicity at the highest dopamine concentration tested (75 μM). The epifluorescence microscopy results indicated that the presence of mucin was crucial for nanoparticle uptake, possibly due to adsorption on the surface, which led to a hard corona structure that could specifically bind to cell surface mucin sites (Figure 6A) [162].

Abbas et al. developed a novel luteolin-loaded nanoparticles coated with chitosan (chitosomes) for the treatment of cognitive impairment in Alzheimer’s disease. The lipids phosphatidylcholine, cholesterol, and phosphatidylserine were dissolved in 1.5 mL of absolute ethanol at a ratio of 7.5:1.5:6.7. Drop by drop, the ethanol-dissolved phospholipids were added to 5 mL of 0.1 M acetate buffer (pH 4.4) while the mixture was stirred magnetically at 800 rpm for 1 h at 25 °C. Overnight, the empty liposomal suspension (F1) was stabilized by being kept at 4 °C. For the luteolin loading, different luteolin concentrations were evaluated (20 mg% *w*/*v*, F1; 40 mg% *w*/*v*, F2) by dissolving in the ethanolic solution of the phospholipids before the addition to the aqueous medium. The varying concentration of chitosan coating was prepared by titration method (F4-F6). Loading of 20 and 40 mg% luteolin resulted in entrapment efficiency of 80.6% ± 1.28% and 77.9% ± 1.33%, respectively. The best quality attributes were achieved with a coating of 4 mg% chitosan solution, resulting in chitosomes with a particle size of 412.8 nm, a PDI of 0.378, and a zeta potential of 37.4. The chitosomes released over 90% of the entrapped drug in 24 h at a pH of 6.4, which mimics the nasal environment. When mixed with mucin solution, the chitosan coat on the chitosomes interacted with the sialic groups of the mucin, leading to the formation of a gel-like structure that allowed the nanoparticles to remain in the nasal cavity for a longer time. Behavioral tests in mice showed an improvement in short-term and long-term spatial memory after treatment with the chitosomes. Histological evaluation revealed increased neuronal survival and a reduction in the number of amyloid plaques, while biochemical results showed improved antioxidant effects and reduced pro-inflammatory mediators’ levels. The chitosomes also suppressed Aβ aggregation and hyperphosphorylated-tau protein levels by half, demonstrating their ability to attenuate the pathological changes of Alzheimer’s disease (Figure 6B) [163].

In a recent study, Kiss et al. synthesized a novel thiomer chitosan-cysteine for use as a mucoadhesive excipient for nasal powders. Thiolation of chitosan can significantly enhance muco-adhesion by promoting the formation of disulfide bonds between the polymer and mucus surface. To create thiomer, L-cysteine was conjugated with chitosan using 1-(3-Dimethylaminopropyl)-3-ethylcarbodiimide hydrochloride and N-hydroxysuccinimide in an aqueous solution. The researchers used a unique quantification method to confirm that there was no L-cysteine remaining in the final polymer matrix, indicating complete thiolation. The thiomer was found to be safe for nasal use based on cytotoxicity studies. The researchers then prepared nasal powder formulations using levodopa methyl ester hydrochloride, an anti-parkinsonian drug, by freeze-drying to investigate their nasal applicability. In vivo studies demonstrated that the powder is effective in reducing the off periods associated with Parkinson’s disease due to its high muco-adhesion [164].

#### 6.1.5. Vaginal Delivery

Apart from its role as a genital organ involved in reproduction, the vagina has also been identified as a potential route for drug administration. The vaginal lumen is susceptible to various pathological conditions such as bacterial, fungal, protozoal, and viral infections, which have traditionally been treated with localized administration of antimicrobial/antiviral drugs within the cervicovaginal region [165]. This route of administration has several advantages over oral administration, including avoiding first-pass metabolism, reducing gastrointestinal side effects, and ease of application. The development of any vaginal drug delivery system must take into account the unique microenvironment of the vagina, including its pH range (typically between 4.0 and 5.0), microbiota, and cyclical changes [166]. While vaginal administration of therapeutic agents for local infections has demonstrated comparable efficacy to oral administration, commercial products for topical vulvovaginal treatment have low patient acceptability as they are associated with discomfort and short drug retention time [167]. Therefore, the development of mucoadhesive systems is necessary to optimize the acceptability and therapeutic efficacy of such therapies [168]. Chitosan is a promising candidate for the development of vaginal drug-delivery systems due to its antimicrobial properties and muco-adhesion/mucopenetration feature. Its use in such systems may improve the retention time of drugs and increase patient compliance [169]. Moreover, chitosan-based formulations have shown to be safe, efficient, and well-tolerated, highlighting its potential as a favorable drug-delivery system within the vagina.

Campos et al. developed and characterized a chitosan-based gynecological gel formulation containing *Mitracarpus frigidus* methanolic extract (MFM) as a potential treatment for *vulvovaginal candidiasis* (VVC), which is an opportunistic infection caused by an overgrowth of Candida species, mainly Candida albicans. The standard treatment for VVC, which involves localized administration of antifungal drugs, is often associated with fungal drug resistance and hypersensitivity reactions. To overcome these limitations, the authors investigated the use of MFM, which has excellent antioxidant and antifungal properties, to develop a new therapeutic option. The authors incorporated MFM into a chitosan-gel and evaluated its effectiveness in an experimental VVC model. The rheological tests demonstrated that the gel exhibited a pseudoplastic behavior, becoming more viscous and elastic with an increase in extract concentration, indicating intermolecular interactions. In the murine model for C. albicans-VVC, treatment with MFM chitosan-gel showed a concentration-dependent effect. Animals treated with 2.5% and 5.0% MFM chitosan-gel exhibited an 80.69% and 89.43% reduction in vaginal fungal load, respectively, while those treated with 10.0% MFM chitosan-gel showed complete clearance of the fungal burden on the last day of treatment. The vaginal mucosa of infected animals treated with MFM chitosan-gel showed a tissue organization similar to that of the vaginal mucosa from non-infected animals, with a marked reduction in inflammatory cell infiltrates (which remained mostly in deep layers of the lamina propria) (Figure 7A). Furthermore, epithelial lesions in the vulvovaginal mucosa, such as cell death and vacuolization, were less prominent in animals treated with MFM chitosan-gel [170].

Jalalvandi et al. designed and evaluated vaginal pH-responsive chitosan hydrogels. The hydrogels were loaded with a non-hormonal spermicide to be used for contraceptive purposes. The platform was developed by modifying chitosan using succinic anhydride to create a water-soluble, biocompatible, biodegradable, and highly pH-sensitive hydrogel. The degradation rate of the hydrogels was found to be faster at lower pH values. The crosslinking density of the gel was observed to play a crucial role in determining the gelation time and degradation rate of the hydrogel, with lower crosslinking (3%) resulting in a faster degradation rate and longer gelation time compared to higher crosslinking density (5%). The in vitro cytotoxicity evaluation demonstrated that the hydrogels were nontoxic towards mesenchymal stem cells over 48 h [171].

Mucoadhesive matrix tablets, which create an in situ gelling effect due to their interaction with the vaginal fluid, are among the most preferred vaginal formulations due to their ease of preparation, low cost, and high stability. Fitaihi et al. conducted a study to investigate the use of chitosan, in combination with other bioadhesive polymers such as hydroxypropyl methylcellulose, guar gum, sodium carboxymethyl cellulose, and polyvinylpyrrolidone, to develop vaginal tablets with sustained-release properties of fluconazole. The tablets were designed to release the drug in acidic vaginal fluid (pH 4.8) for prolonged periods. The team evaluated 17 different formulations with varying ratios of chitosan and the other polymers and selected two formulations for further study. The first formulation contained chitosan, guar gum, and polyvinylpyrrolidone in a ratio of 1:2:1, while the second was formulated with chitosan and sodium carboxymethyl cellulose in a ratio of 1:2. Both formulations demonstrated acceptable powder flowability and tablet characteristics, including desirable hardness and friability. The drug release profiles at pH 4.8 showed sustained-release characteristics for both formulations, and the tablets exhibited good bioadhesion to the mouse peritoneum membrane [172]. In a similar study, Paczkowska et al. prepared mucoadhesive chitosan-based tablets with lyophilized extracts from *Chelidonium majus* for the treatment of vaginitis [173].

Moreno et al. fabricated electrosprayed chitosan microcapsules loaded with hydroalcoholic extracts of various Argentinean plants for treating vaginal infections. A 5% chitosan solution was prepared by dissolving the polysaccharide in 80% acetic acid (*v*/*v*) at room temperature under magnetic stirring. Dry plant extracts (10% *w*/*w* of the total solids content) from *Larrea divaricata*, *L. cuneifolia*, *L. nitida*, *Zuccagnia punctata*, and *Tetraglochin andina* were added to the chitosan solutions, and the mixtures were stirred until complete dissolution. The samples were electro-sprayed by using a home-made electrospinning/electro-spraying apparatus. Evaluation of water uptake capacity in simulated vaginal fluid showed that all chitosan-based encapsulation structures underwent up to 90% swelling. The solubility of the selected extracts was favored by microcapsule formation, thus increasing the bioavailability of the active compounds in the vaginal environment [174].

Maestrelli et al. developed chitosan-alginate microspheres as a potential vaginal delivery system for cefixime to address the problems of poor bioavailability and frequency of side effects appearance related to oral drug administration. They utilized calcium chloride as a cross-linking agent for ionotropic gelation to form the microparticles. The study found that the entrapment efficiency of the drug increased with drug loading concentration in the starting solution and reached an optimal drug-to-polymer ratio at 30 mg/mL. The swelling properties of the microspheres were found to increase with the entrapped drug amount, and the water uptake reached its maximum value at the same drug loading concentration of 30 mg/mL. The muco-adhesion studies on excised porcine vaginal mucosa demonstrated that all formulations ensured in situ permanence for longer than 2 h. Microbiological studies highlighted a direct relation between in vitro drug release rate from microspheres and the reduction of metabolic activity/viability of *E. coli* [175].

Darwesh et al. developed a chitosan/anion polyelectrolyte complex (PEC) for the formulation of fluconazole vaginal inserts with controlled release and considerable muco-adhesion. The vaginal inserts were prepared by lyophilization using mannitol. A factorial design was applied to investigate the effect of the anion type and Chitosan/anion ratio on the insert’s muco-adhesion and drug release properties. The release profile of the optimized insert (based on 5:5 chitosan: sodium alginate) was modulated by adding Compritol^®^ 888 (a release retardant). Fluconazole inserts showed satisfactory drug content, acceptable friability percentages, and the highest swelling indices at six hours. Statistical analysis showed a significant effect of the studied factors on detachment force and release properties. The platform showed enhanced antifungal activity against *Candida albicans* both in vitro and in vivo with reduced inflammatory cells (confirmed by a histological evaluation) (Figure 7B) [176].

**Figure 7 pharmaceutics-15-01313-f007:**
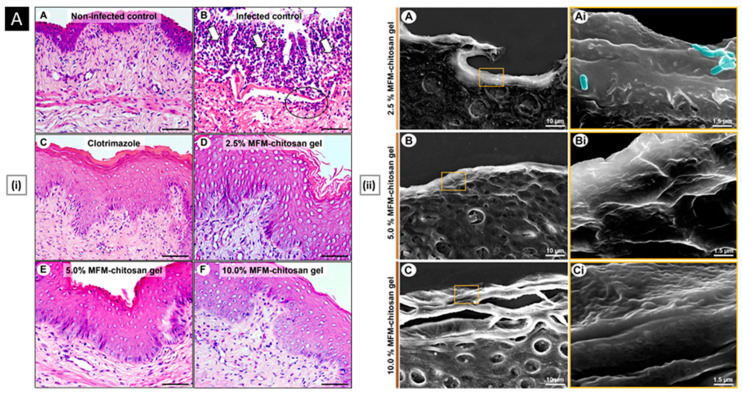
((**A**), i) Vulvovaginal histological sections from the experimental groups of animals. Female rats were intravaginally infected or not with *C. albicans* and treated or not with MFM-chitosan gel or clotrimazole. H&E-stained vulvovaginal tissue sections were analyzed by light microscopy. ((**A**), ii) SEM views of vulvovaginal epithelium in vulvovaginal candidiasis rat models after vaginal topical treatment with MFM-chitosan gel. (A–C) shows the well-preserved ultrastructure of vaginal epithelium treated with chitosan-gel containing 2.5% (A), 5.0% (B), and 10.0% (C) MFM [fungal cells are highlighted in green]. Reproduced with permission from [170], copyright Elsevier 2020. ((**B**), i) SEM images of (A) Unloaded 5:5 chitosan/sodium alginate PEC-based insert, (B) Fluconazole 5:5 chitosan/sodium alginate PEC, (C) Unloaded 5:5 chitosan/xanthan gum PEC-based insert, (D) Fluconazole 5:5 chitosan/xanthan gum PEC, (E) Unloaded 5:5 chitosan/carpobol PEC-based insert, (F) Fluconazole 5:5 chitosan/carpobol PEC ((**B**), ii) Histological examination of Candida infected vaginal tissue treated by unloaded vaginal insert, fluconazole PEC based vaginal insert and fluconazole solution. (A) Control normal vaginal tissue; (B) Control Candida infected, non-treated vaginal tissue; (C) Candida infected vaginal tissue treated by unloaded vaginal insert; (D) Candida infected vaginal tissue treated by fluconazole solution; (E) Candida infected vaginal tissue treated by fluconazole vaginal insert. Stars represent inflammatory cells; Black arrows represent normal epithelium; Dotted arrows represent hyperplastic or damaged epithelium. Reproduced with permission from [176], copyright MDPI 2018.

### 6.2. Biomedical Applications

#### 6.2.1. Bone Regeneration

Bone regenerative therapy offers the potential to treat various skeletal disorders/injuries, such as fractures, osteoporosis, and congenital defects, which can cause chronic pain, disability, and reduced quality of life. Conventional treatments for these conditions, such as surgery and drugs, have limitations and often result in incomplete recovery. In this context, chitosan-based platforms can be employed to promote bone regeneration by direct delivery of growth factors/drugs to the site of injury or by providing a biomimetic template for the growth of new bone tissue. Chitosan promotes the production of extracellular matrix (ECM) components, such as collagen, which are essential for bone growth and repair. Additionally, its antimicrobial properties reduce the risk of infection, and its biodegradable nature makes it an attractive option for use in long-term applications [177]. While chitosan is a promising material for various medical applications, it is important to acknowledge its limitations when it comes to bone tissue engineering. Specifically, chitosan is not strong enough to support the weight-bearing demands of bone implants, and it lacks the osteoconductive properties that natural bones possess. To overcome these limitations, researchers have created bio-composites by blending chitosan with other biopolymers, such as chitin, silk, and polycaprolactone, and bioactive nanoceramics, such as hydroxyapatite and zirconia [178,179]. These bio-composites offer improved mechanical strength and structural integrity, making them more suitable for bone tissue engineering applications. Osteoconductivity can be enhanced by delivering cytokines or by doping bioactive trace elements (like Sr^2+^, Zn^2+^, Mg^2+^, Cu^2+^, and Si^4+^) [180]. Some cutting-edge chitosan-based platforms for bone tissue regeneration are discussed below.

The key reasons for the failure of bone-defect healing are a delayed inflammatory response and poor osteogenesis. To address this, Wan et al. developed a dual-responsive hydrogel-microsphere composite composed of near-infrared (NIR)-responsive polydopamine-coated microspheres that were incorporated into a thermo-responsive hydroxy butyl chitosan hydrogel. When used in a critical-size calvaria-defect rat model, the platform provided a rapid release of aspirin (encapsulated in the hydrogel matrix) that mitigates the acute inflammatory reaction in the early stage (day 1–3) of bone healing, whereas bone morphogenetic protein-2 (encapsulated within NIR-responsive microspheres) released in a sustained manner (day 3–14) (Figure 8A). Such sequential release behavior conjoins well with the cascaded processes of bone healing, thereby achieving enhanced bone regeneration [181].

On account of their osteoconductivity, resorbability, and similarity to natural bone minerals, calcium phosphate (and its derivatives) is widely used in combination with other biomaterials for bone regeneration. Chen et al. synthesized three different types of calcium phosphates, namely calcium-deficient hydroxyapatite (CDHA), beta-tricalcium phosphate (TCP), and biphasic calcium phosphate (BCP) using a reverse emulsion method followed by calcination. They then compared the efficacy of these materials in promoting bone regeneration by blending them with chitosan to produce porous hybrid membranes. All membranes possessed interconnected pores with porosity and pore size ranging from 91 to 95% and from 102 to 147 µm, respectively. The findings suggest that CDHA promotes cell adhesion and differentiation by adsorbing cell-adhesive proteins, while TCP enhances cell proliferation by providing a calcium/phosphate-rich environment through its high solubility. BCP combines the beneficial effects of the individual phases, resulting in the most effective bone regeneration, as evident from the in vivo study using critical-sized calvarial bone defects in rats where BCP-loaded membranes showed the highest bone regeneration efficacy (roughly 57% newly formed bone in the original defect area after three weeks) (Figure 8B) [182].

ECM-mimicking materials are useful in bone regeneration because they mimic the native ECM of bones, which provides a template for cells to organize and form new tissue. These materials are designed to have similar chemical composition and microarchitecture as the natural ECM. By doing so, they can create a supportive environment that promotes cell adhesion, proliferation, and differentiation, leading to enhanced bone regeneration [183]. Additionally, they also are designed to have specific mechanical properties and degradation rates that match the desired time course of bone healing. This can help to optimize the delivery of growth factors/drugs and reduce the risk of implant failure. Yang et al. reported ECM-mimicking self-assembled nanofibrous chitosan microspheres developed through thermal induction of chitosan molecular chain from alkaline/urea aqueous solution. Modulation of the initial solution concentration and the reaction temperature provided precise control over the morphology of microspheres. Along with the required biocompatibility and biodegradability, the microspheres possessed excellent cell adhesion capability that facilitated the formation of large-sized 3D geometric constructs as observed during in vitro studies when MC3T3-E1 cells were co-cultured with microspheres for 3 days. The platform promoted osteogenic differentiation of rat bone marrow stem cells (quantified via alkaline phosphatase staining). After 14 days of co-culture, the expression levels of genes related to osteogenic differentiation, like alkaline phosphatase and osteocalcin, were found to be 6.0-fold and 5.6-fold higher (vs. the control group), respectively. Upon implantation of pre-differentiated rat bone marrow stem cells-loaded microspheres into a rat calvarial defect model, the defects were almost bridged by new bone, indicating better bone regeneration performance (Figure 9). The bone mineral density treatment group was significantly higher than that of the cell-free microspheres and the control group (399.15 mg/cm^3^ vs. 224.09 mg/cm^3^ vs. 74.50 mg/cm^3^, respectively) [184].

A study performed by Yong et al. aimed to examine the impact of different Strontium (Sr) percentages in Sr-hydroxyapatite nanocrystals on osteoinductivity [185]. They fabricated chitosan, strontium, and hydroxyapatite nanohybrid scaffolds to achieve this goal. The findings showed that the addition of Sr in hydroxyapatite enhanced the cell volume and axial length while reducing the particle size. The scaffolds were found to have uniformly distributed Sr-hydroxyapatite micropores of 100–400 μm, which facilitated cell proliferation and osteogenic differentiation due to the release of Sr^2+^ ions. This led to improved alkaline phosphatase activity, ECM mineralization, and increased levels of COL-1 and ALP expression related to the osteogenic activity. The Sr-hydroxyapatite chitosan scaffolds demonstrated remarkable osteoinductivity due to the synergistic effect of Ca^2+^ and Sr^2+^ ions, making it a promising candidate for bone engineering applications.

#### 6.2.2. Cartilage Tissue Regeneration

Cartilage regeneration is a challenging area due to the cumulative existence of several factors. Firstly, cartilage is a complex structure that consists of chondrocytes, ECM components, and signaling molecules. This complexity makes it difficult to recreate the natural environment of cartilage in vitro, which is necessary for successful regeneration. Secondly, it has a limited blood supply, making it difficult for cells to access the nutrients and oxygen needed to survive and function. Avascularity also hinders the effective delivery of therapeutic agents to the site of tissue damage. Lastly, cartilage is subjected to high loads and repetitive motions, so it is tricky to find materials that have the appropriate mechanical properties while also being biologically compatible [186]. The therapeutic success of chondrocyte transplantation and/or the quality of neocartilage production is highly dependent on the cell-carrier material used. By tailoring the properties of chitosan and its derivatives, it is possible to address some of the challenges associated with cartilage regeneration. Several platforms have been reported in recent years which focus on addressing the inherently low regenerative capacity of cartilages by utilizing the ECM-mimicking approach or by functionalization with signaling molecules that enhance the chondrocyte’s proliferation and differentiation [187]. Building upon the success of traditional chitosan-based approaches, new innovations are emerging that offer even greater potential for cartilage regeneration.

Osteochondral tissue (commonly found in weight-bearing joints, such as the knee and ankle) combines articular cartilage and subchondral bone within a single structure. In case of damage, its repair is extremely complex owing to a biphasic structure. The use of multizonal scaffolds that mimic the gradient transitions from the cartilage surface to the subchondral bone can open new avenues for osteochondral regeneration. To address this, Pitrolino et al. combined freeze-drying and porogen-leach-out methods to produce a bioresorbable multi-layered chitosan scaffold with controlled porosity in distinct but integrated layers. The mean pore size for the cartilage phase, interphase, and subchondral phase were 160 ± 12.3 μm, 248 ± 33.2 μm, and 275 ± 31.7 μm, respectively. The authors reported the incorporation of 70% (by weight) nano-hydroxyapatite to provide additional strength to the bone-like layer. Under compressive loading, the scaffold demonstrated a prompt mechanical recovery and remained intact under tensile loading. Human mesenchymal stem cells showed successful attachment and growth on the scaffold, with cells adopting a typical adherent morphology on the bone layer and a rounded shape on the chondrogenic layer. In vitro studies revealed that mesenchymal stem cells differentiate into both osteogenic and chondrogenic cells in specific layers of the scaffold, influenced by the unique pore gradient and composition of the material (Figure 10A) [188]. In a similar study, Samie et al. developed a porous subchondral bone substitute using chitosan, nano-hydroxyapatite, and either hydroxypropyl methylcellulose or silk fibroin. The scaffold was loaded with triamcinolone acetonide (to counter the elevated levels of tumor necrosis factor-α and interleukins at the defect site that downregulates ECM production) or transforming growth factor-β1 (for pro-osteogenic activity). The drug-loaded scaffold served dual functionality in terms of local delivery within the extracellular environment, along with facilitating support for the cells to organize [189].

Articular cartilage is a type of soft tissue that covers the ends of bones in synovial joints and provides a smooth, low-friction surface for movement. This type of cartilage plays a crucial role in absorbing shock and reducing friction between the bones in the joint. Approaches to regenerate it have been widely explored as the articular cartilage is prone to get damaged due to osteoarthritis or trauma (direct injuries to the joint, such as a fall or a sports-related injury). Acar et al. reported 3D coacervate scaffolds prepared through complex coacervation between different chitosan salts and sodium hyaluronate for the regeneration of articular cartilage. Complex coacervation is a microencapsulation technique based on the separation of two immiscible liquids in an aqueous environment induced by the interaction of macroions with opposite charges. It can be used to encapsulate cells within scaffold structures. SEM imaging revealed that the 3D architecture of coacervates encapsulating bone marrow mesenchymal stem cells had an interconnected porous network of a gel-like scaffold that facilitated cell–cell and cell–matrix interactions. Chondrogenic induction of encapsulated stem cells within the coacervates demonstrated remarkable cellular viability (>84%) with elevated expression levels of chondrogenic markers (Figure 10B) [190].

In a different study, Liu et al. produced a dual network hydrogel for articular cartilage regeneration comprising thiolated chitosan and silk fibroin. The system had excellent injectability and underwent spontaneous gelling in response to physiological pH and temperature. The chitosan component provided enhanced strength and stiffness, whereas silk fibroin majorly contributed to the gel’s elasticity. The platform provided support for chondrocyte growth and promoted chondrocyte-specified matrix deposition, which aided in the effective maintenance of their phenotype [191].

#### 6.2.3. Cardiac Tissue Regeneration

The global burden of cardiovascular disease is a major health challenge, with heart disease being the leading cause of death worldwide. As the demand for heart transplants far outweighs the supply of suitable donor organs, there is a growing need for alternative approaches to treat or replace damaged heart tissue. Cardiac tissue engineering offers a promising solution by combining the principles of biology, materials science, and engineering to develop functional cardiac tissue. There are several strategies for cardiac tissue engineering, including cell injection, implantation of three-dimensional tissue constructs or patches, acellular material injection, and valve replacement [192]. To achieve physiological function, the materials used in these approaches must possess mechanical and electrical properties similar to those of native cardiac tissue, which is composed of cardiomyocytes, smooth muscle cells, and endothelial cells. Additionally, the materials must be designed in a way that promotes tissue integration and functional connectivity with the surrounding cells and blood vessels. For example, surface or structural features that guide tissue alignment and dictate cell shape can enhance the regenerative potential of the heart [193]. In recent years, chitosan-based hybrid platforms have shown promising results as a material for cardiac tissue engineering [194]. Some interesting applications are discussed below.

Lv et al. constructed a 3D scaffold using human cardiac ECM, chitosan, and gelatin for use as a tissue-engineered heart patch. The authors collected specimens of discarded human right atrial appendages during heart surgery. Subsequently, the human cardiac-derived ECM was blended with chitosan and gelatin. Lyophilization of the final mixture generates a porous (pore size of 40–100 µm) composite scaffold. Upon incorporation of CD34+ endothelial progenitor cells, the scaffold not only promoted their survival and proliferation but also induced their differentiation into endothelial cells. The authors attributed these results to the presence of key ECM components like collagen, fibronectin, and laminin. In addition to mimicking the ECM environment, the composite scaffold possessed advantageous physical properties of porosity and efficient water absorption, making it an ideal candidate for use in congenital heart defects [195].

Ke at al. [196] reported an injectable thermosensitive hydrogel of chitosan/dextran/β–glycerophosphate (CS/DEX/β-GP) loaded with umbilical cord mesenchymal stem cells (HUVECs) for treatment of myocardial infarction. The system was prepared by the cumulative effect of electrostatic attractions (between the amino group of chitosan and the phosphate groups of β-GP) and hydrogen bond interactions (chitosan-dextran). Gelation time and favorable mechanical strengths (for cell growth) were controlled by varying the concentration of dextran between 0.5–2.0% (*w*/*v*). A dextran concentration of 1% *w*/*v* was used for all subsequent studies. The hydrogels showed good viability and proliferation of 3T3 cells and HUVECs in vitro. HUVECs encapsulated within hydrogel showed a linear-like profile, and the platform facilitated a four-fold increase in cell number after 3 days of incubation. The hydrogel activated p-Akt and p-ERK1/2 signaling pathways related to cell survival/proliferation. After 14 days of cultivation under differentiation media, the hydrogel remarkably upregulated the expression of key cardiac markers (cTnI and Cx43) (Figure 11A).

Zarei et al. have reported on the development of a nanofiber scaffold designed to enhance the healing process of damaged electroconductive tissues. This was achieved by combining a chitosan-collagen blend with varying weight percentages of conductive polypyrrole (5%, 10%, 15%, 20%, 25%). The polypyrrole particles were evenly dispersed within the nanofibers, resulting in a decrease in fiber diameter as the polypyrrole content increased. The inclusion of polypyrrole in the fibers improved their conductivity to 164.274 × 10^−3^ s/m, which falls within the range of semi-conductive and conductive polymers. Analysis using MTT and SEM showed that the nanofiber containing 10% polypyrrole had superior properties for cell adhesion, growth, and proliferation compared to other compositions. The tensile strength and Young’s modulus of the conductive nanofiber scaffold were also found to match the mechanical properties of desired tissues. In vitro degradation studies showed that the conductive scaffolds displayed a suitable degradation behavior in an aqueous environment, with a typical degradation rate. The results of a cell toxicity study demonstrated that all of the produced scaffolds were fully biocompatible, and the presence of bioactive components such as chitosan and collagen stimulated cellular morphology and enhanced cell proliferation. Additionally, the low electrical resistance of the scaffolds, derived from the presence of polypyrrole, enabled the effective transmission of electrical signals (Figure 11B) [197].

Mombini et al. used polyvinyl alcohol, chitosan, and different concentration of carbon nanotubes (CNTs, 1, 3, and 5 wt%) to produce electrospun cardiac conductive scaffolds. Nanofiber containing 1 wt% CNT had optimal physiochemical properties (elastic modulus 130 ± 3.605 MPa, electrical conductivity 3.4 × 10^−6^ S/cm, cell viability >80%). Using the optimal scaffold, the authors performed differentiation of rat mesenchymal stem cells to cardiomyocytes by electrical stimulation in the presence of 5-azacytidine, TGF-β, and ascorbic acid. Gene expression profiling using real-time qPCR showed that the expression of Nkx2.5, Troponin I, and β–MHC cardiac marker was increased by three-fold (in comparison to the control group) [198].

#### 6.2.4. Corneal Regeneration

The cornea, a critical component of the eye, relies on corneal epithelial cells to maintain its transparency. Unfortunately, these cells are unable to regenerate in humans, which can result in vision loss due to factors such as aging, trauma, or disease. In extreme cases, a significant reduction of corneal epithelial cells can lead to blindness. According to the World Health Organization, the corneal disease is a leading cause of blindness globally, second only to cataracts [199]. Given the high demand for corneal tissue and the limited availability of donors, the field of corneal tissue engineering has become increasingly important for the treatment of ocular cells [200]. The cornea plays a crucial role in ensuring the clarity of vision but is also susceptible to injury. Currently, treatment for corneal injuries often involves a transplant of an amniotic membrane, but this approach carries the risk of infection and rejection. To address this issue, the use of biomaterial-based scaffolds or membranes has been widely explored [201]. Chitosan is a popular choice due to its biocompatibility and anti-inflammatory properties, but it has limitations, such as poor mechanical strength. To address this, chitosan can be blended with other polymers to enhance its scaffold characteristics. An ideal cornea tissue scaffold must have similar mechanical and optical properties to the natural cornea and must be able to support cells and maintain high adhesion. In addition, optical transparency is a critical requirement for corneal implants, so extra care must be taken to ensure that the materials and processing methods used to preserve this important characteristic [202]. The scaffold or membrane should also be flexible and resilient enough to withstand surgical manipulation. The subsequent part highlights some of the exceptional research work centered on chitosan and corneal regeneration.

With the aim to develop robust substrates for the transplantation of corneal endothelial cells into the damaged cornea, Ozcelik et al. prepared ultrathin chitosan-poly (ethylene glycol) hydrogel films (CPHFs) for corneal tissue engineering applications. By adjusting the PEG content (1.5–5.9 wt%), the CPHFs were able to mimic the tensile strain and ultimate stress of human corneal tissue while maintaining similar tensile moduli. Optical transparency measurements showed that the films were over 95% transparent in the visible spectrum (400–700 nm), surpassing the maximum 75% transparency of the human cornea. The CPHFs also maintained their biodegradable properties when subjected to in vitro degradation with lysozyme. Cell culture experiments confirmed the ability of the CPHFs to support the attachment and growth of sheep corneal epithelial cells. Ex vivo surgical trials on sheep eyes showed that the ultrathin CPHFs have excellent physical properties for implantation and display favorable mechanical, optical, and degradation properties, making them a promising option for the regeneration and transplantation of corneal epithelial cells [203]. Tayebi et al. developed a biodegradable scaffold for growing corneal cells by incorporating chitosan nanoparticles (CSNPs) into a polycaprolactone (PCL)/chitosan membrane. They formulated different ratios of PCL and CSNP in a fixed chitosan concentration and created the films using the solvent casting method. In a dry state, the scaffolds were fragile, which became completely flexible and easy to handle after brief immersion in phosphate-buffered saline. Measurement of transparency in the wavelength of 450 nm and 600 nm showed that between all composites, scaffolds with CSNP/PCL composition of 50/25 had the highest degree of transparency at about 74% (statistically similar to the transparency of acellular stroma, 80%). The scaffold was found to be non-toxic and promoted the growth of corneal epithelial cells, as demonstrated by an MTT assay. In vitro tests such as H&E staining and flow cytometry showed that corneal epithelial cells attached well to the scaffold, forming a compact monolayer. This was also confirmed by visualization of the scaffold under SEM (Figure 12A) [204].

The current treatment methods for corneal stromal defects often result in inadequate clinical outcomes due to the irregular shapes of the defects. To address this problem, Feng et al. created a new type of biomimetic scaffold within situ-forming properties. This was achieved through dendronizing chitosan with oligoethylene glycol-based dendrons, resulting in a novel type of dendronized chitosan (DC) with thermo-gelling properties (Figure 12B). The DCs have specific radial amphiphilicity, which allows them to self-assemble into long fibers and form highly transparent hydrogels through thermal aggregation and gelation. The authors found that they could easily modulate the gelling points of the hydrogel by varying the grafting ratios of dendrons. When tested in physiological conditions (PBS at 37 °C), the DC hydrogel displayed enhanced mechanical strengths. The biocompatibility of the platform was demonstrated using rabbit corneal stromal cells, which showed that it could promote keratinocyte proliferation and migration. The DC hydrogel’s potential for corneal tissue engineering was further suggested by the migration of cells into the microporous material. In a corneal stromal defect model in rabbits, the in vivo performance of the platform was evaluated. Two weeks after implantation, the results of H&E showed that there was hardly any observed inflammatory response in the stromal layer of the cornea in the DC hydrogel group. This indicates that the hydrogel can fill the matrix while also reducing the inflammatory response. Four weeks after implantation, migration of resident keratocytes to the gel matrix was observed, suggesting that the platform is suitable for the growth of corneal stromal cells [205].

Xu et al. developed a novel in situ alginate-chitosan hydrogel (ACH) for the transplantation of limbal stem cells (LSCs) to induce corneal reconstruction after corneal alkali burns. The hydrogel was formed in situ using sodium alginate dialdehyde (SAD), which was made by oxidizing sodium alginate with periodate, and carboxymethyl chitosan, which was rapidly crosslinked with SAD through Schiff’s base formation. The hydrogel formed quickly on the wound surface without any chemical crosslinker and was tested for its transparency, gelation time, microscopic structure, swelling, biocompatibility, cytotoxicity, and degradability. Primary LSCs were then encapsulated in the hydrogel and transplanted to corneal alkali burns in a rabbit model. Visual inspection, slit lamp examinations, histological analysis, and immunofluorescence staining were used to assess reconstruction. The results showed that the hydrogel was highly transparent, gelated quickly, and was biocompatible with low cytotoxicity. In vitro, the LSCs expressed the stem cell marker p63 but lacked markers for differentiated epithelium. Furthermore, the hydrogel containing LSCs formed quickly on the alkali burn wound and significantly improved epithelial reconstruction [206].

#### 6.2.5. Periodontal Tissue Regeneration

The tissues that support teeth, known as the periodontium, are made up of the gingiva, cementum, periodontal ligament, and alveolar bone. An inflammatory disease called periodontitis leads to the deterioration of these tissues and eventually results in tooth lost. Currently, treatments for periodontitis center around removing plaque and controlling local inflammation through scaling and root planning and other surgical procedures. Although these treatments alleviate symptoms and slow the progression of the disease, they are unable to restore the connection between the periodontal tissues and teeth to their original state [207]. This means that even after treatment, the functionality of teeth and dentition remains impacted. Efforts have been made to regrow these tissues through methods such as bone grafts. However, the results of these approaches are inconsistent. Therefore, it is crucial to find alternative strategies for restoring the structure and function of periodontal tissues in periodontitis patients [208].

The process of tooth development involves niche-residing dental follicle cells transforming into cementoblasts, fibroblasts, and osteoblasts, which create cementum, periodontal ligament, and alveolar bone, respectively. However, once tooth development is complete, the niche that plays a crucial role in forming the supporting tissues is no longer present, making it challenging to repair damaged or lost periodontium. This is where biomaterial-assisted periodontal tissue engineering comes in, offering the possibility of recreating the microenvironment and regenerating functional tissues [209]. Chitosan is a favorable material for dental applications due to its bioactivity, natural antimicrobial properties, and biocompatibility. While it works well on its own, combining chitosan with compatible organic or inorganic materials can lead to improved results in periodontal regeneration [210]. This is because the periodontal microenvironment is complex and requires a multi-component platform to meet the varied compositional and mechanical needs for successful regeneration. The platform must also be able to regulate the formation of periodontal tissues with the proper structural features [211]. Below are a few recent examples of chitosan-based platforms utilized for periodontal regeneration.

Varoni et al. developed a trilayer scaffold made from chitosan to promote simultaneous multi-tissue healing for periodontal regeneration. They created two porous compartments for bone and gingiva regeneration using Genipin to cross-link either medium molecular weight (MMW) or low molecular weight (LMW) chitosan and then freeze-drying the obtained scaffolds. A third compartment for periodontal ligament regeneration was created through chitosan electrochemical deposition, resulting in highly oriented microchannels of around 450-µm in diameter to support periodontal ligament fiber growth towards the dental root. In vitro tests revealed a rapid equilibrium water content for both MMW- and LMW-chitosan compartments, with the MMW-chitosan compartment showing slower degradation and higher resistance to compression (28% ± 1% of weight loss at 4 weeks; compression modulus of 18 ± 6 kPa). More than 90% of human primary periodontal cell populations were found to be viable and showed active cell metabolism during cytocompatibility tests. In vivo tests demonstrated high biocompatibility in wild-type mice, with tissue ingrowth and vascularization within the scaffold. In the periodontal ectopic model in nude mice, the authors seeded the scaffold compartments with human gingival fibroblasts, osteoblasts, and periodontal ligament fibroblasts, resulting in a dense mineralized matrix within the MMW-chitosan region and weakly mineralized deposits at the dentin interface. Overall, the findings suggest that this resorbable trilayer scaffold holds great potential for periodontal regeneration [212].

Lee et al. stated that the tri-layer functional chitosan membrane can be used in guided periodontal tissue regeneration with epigallocatechin-3-gallate (EGCG) grafted on the outer layer for bactericidal activity and lovastatin entrapped in the middle layer for controlled release. EGCG inhibits lipopolysaccharide-induced osteoclastic bone resorption in vitro, and its localized delivery has been in use as an efficient adjunctive therapy treating chronic periodontitis, while lovastatin can induce angiogenesis and stimulate bone formation by enhancing expression of the bone morphogenetic protein-2 gene in bone cells. Even after the incorporation of drug molecules, the trilayer membrane possessed significantly higher tensile strength than commercial collagen membranes (BioMend^®^ and BioGide^®^; 1.67 ± 0.14 vs. 0.91 ± 0.00 vs. 0.87 ± 0.07). By playing with the membrane concentration, the authors achieved sustained drug release for over 70 days. The trilayer membrane displayed excellent anti-bacterial and ALPase activity. In vivo periodontal regeneration ability was evaluated in 1.5-year-old beagle dogs. The maxillary second premolars were extracted, and one-walled defects were formed at the mesial and distal sides. As per the micro-CT analysis, the chitosan-based platform outperformed the commercial membrane by inducing 62.03% of new bone formation (vs. 46.07%) (Figure 13A) [213]. In a study by Zhou et al., the ability of a biomimetic electrospun composite nanofiber membrane consisting of fish collagen, bioactive glass, and chitosan (Col/BG/CS) to induce periodontal tissue regeneration was evaluated using a dog class II furcation defect model. The findings showed that the biomimetic structure of the composite membrane has high hydrophilicity (contact angle of 12.83 ± 3°) and a tensile strength of 13.1 ± 0.43 Mpa. In comparison to a pure fish collagen membrane, the Col/BG/CS composite membrane demonstrated improved antibacterial activity against Streptococcus mutans. Additionally, the composite membrane was found to enhance cell viability and osteogenic gene expression in human periodontal ligament cells, as well as promote the expression of RUNX-2 and OPN protein [214].

Park et al. investigated the clinical benefits of adding ε-aminocaproic acid (ACA) loaded chitosan-tripolyphosphate nanoparticles to fibrin to enhance cementoblast differentiation and cementum regeneration. The formation of cementum on the exposed root surface of a tooth is a key factor in periodontal regeneration. The researchers found that cementoblasts (OCCM-30 cells) grown on a fibrin matrix showed high levels of matrix proteinases, which led to the degradation of fibrin and apoptosis of the OCCM-30 cells. This issue was reversed when the cells were treated with a proteinase inhibitor, ACA. As per these findings, ACA-releasing chitosan-tripolyphosphate nanoparticles (ACP) were created and incorporated into fibrin to create fibrin-ACP. This composite showed enhanced differentiation of cementoblasts in vitro, as indicated by bio-mineralization and expressions of mineralization-related molecules. In a periodontal defect study on beagles, fibrin-ACP lead to substantial cementum formation on the exposed root dentin in vivo, outperforming fibrin alone and the clinically used enamel matrix derivative (EMD) in terms of cementum regeneration. The fibrin-ACP also enabled the formation of structural connections between the cementum, periodontal ligament, and bone, as demonstrated by Sharpey’s fiber insertion. Furthermore, fibrin-ACP improved alveolar bone regeneration by increasing the bone volume in tooth furcation defects and improving root coverage (Figure 13B) [215].

#### 6.2.6. Wound Healing

Wounds are disruptions to the integrity of the skin resulted due to external physical/thermal injury or internal pathological conditions. They can be categorized into two types: acute and chronic. Acute wounds heal entirely in 8–12 weeks with little scarring; however, chronic wounds may reoccur and have a recovery duration that exceeds 12 weeks [216]. The healing process can be hindered by underlying physiological conditions, making it necessary to properly manage these wounds. Examples of chronic wounds comprise of venous leg ulcers, pressure ulcers (bed sores), diabetic foot ulcers, and non-healing surgical wounds. Skin layers and affected areas serve as the basis for the gradation of wounds. The epidermal skin surface is only involved in superficial wounds. Partial thickness wounds are injuries involving the epidermis, deeper dermal layers, blood vessels, hair follicles, and sweat glands. When the subcutaneous fat or deeper tissue, as well as the epidermal and dermal layers, are wounded, full-thickness wounds result [217].

Wound healing is a complex and multifaceted process that requires a conducive environment to support its acceleration. The body’s normal reaction to damage consists of a sequence of interconnected stages such as hemostasis, inflammation, proliferation, and remodeling that aim to restore the skin’s integrity. This process involves numerous cell types, enzymes, cytokines, proteins, and hormones working in concert. Following injury, hemostasis is triggered to form blood clots and constrict blood vessels, followed by the secretion of proinflammatory cytokines and growth factors that promote inflammation. Macrophages, neutrophils, and lymphocytes are then recruited to facilitate reepithelization and angiogenesis, leading to the proliferation of fibroblasts and keratinocytes and eventual differentiation of fibroblasts into myofibroblasts, which deposit extracellular matrix [218]. To aid the healing process, a proper wound dressing must be applied to protect the wound site from external mechanical and microbial stress. Traditional dressings like cotton, bandages, and gauze often fail to provide a moist and supportive environment and can cause pain upon removal due to wound drainage [219].

An ideal wound dressing must have several key properties to facilitate the healing process. Firstly, it should maintain a moist environment to allow for proper gas exchange and prevent dryness of the wound. Secondly, it should be able to absorb any exudates produced by the wound to maintain a clean and hygienic environment. Thirdly, it should support cell proliferation and inhibit bacterial growth by preventing infections and promoting healing [220]. Studies have shown that chitosan is a promising option for use in wound care. Its unique set of biological features, such as antibacterial, hemostatic, and mucoadhesive characteristics, make it an attractive option for wound dressings. Additionally, chitosan has been shown to accelerate the wound-healing process. Several chitosan-based platforms have emerged and used in wound healing applications, either alone or in combination with several natural or synthetic biomaterials. These platforms offer state-of-the-art solutions for wound dressings and are being actively researched.

Deng et al. fabricated porous sponge-like dressings by freeze-drying a novel water-soluble adenine-modified chitosan derivative (CS-A), synthesized using a simple amide reaction between chitosan and carboxylated adenine [221]. The aim was to tackle the issue of bacterial infections that can impede the healing process and result in severe tissue damage during the initial phase of wound healing. The choice to incorporate adenine into the wound dressing was driven by its role as a critical component of nucleotides, which are necessary for the growth and division of biological cells. Adenine has been shown to boost the proliferation of IEC-6 cells, fibroblasts, astrocytes, and human endothelial cells. Additionally, it can stimulate the production of angiogenesis factors and encourage the development of newer blood vessels. Thus, the addition of adenine to the wound dressing offers potent growth factor capabilities, promoting both cell proliferation and angiogenesis. The CS-A derivatives displayed outstanding antimicrobial properties and cell compatibility. Even at high concentrations of up to 2 mg/mL, the rate of hemolysis was less than 1.5%, indicating excellent blood compatibility. In a full-thickness skin defect model, the wounds given treatment with CS-A sponges showed quicker wound healing compared to those of gauze and chitosan sponges. As anticipated, the CS-A derivatives demonstrated antibacterial properties during the inflammation stage of wound healing. During the proliferation stage, they facilitated the advancement of cell proliferation by decreasing the duration of the G1 phase, thereby encouraging fibroblasts to move to the next stage. Parallelly, the CS-A derivatives reduced the infiltration of inflammatory cells, stimulated the formation of new blood vessels, accelerated the buildup of collagen tissue, and stimulated the regeneration of epithelial tissue.

Efficient and prompt control of bleeding is crucial to address hypotension and multi-organ failure that can result from massive blood loss, often leading to high death rates in both civilian and military populations. The human body’s natural blood clotting process can be aided by shape-memory hemostats to quickly stop severe bleeding from deep and non-compressible wounds. An ideal hemostat should have a porous structure that allows for rapid shape recovery, which helps to hold it in place at the wound site by draining excess water and absorbing blood to regain its original shape. This shape recovery can also apply pressure on the wound, effectively controlling the bleeding. With these prerequisites in mind, Du et al. developed a hemostatic chitosan sponge (micro-channeled alkylated chitosan sponge, MACS) that has highly interconnected microchannels by using a combination of 3D printed microfiber leaching, freeze-drying, and surface modification [222]. The fabrication process for MACS started with the creation of a sacrificial PLA microfiber template using a 3D printer. This template was then filled with a 4% chitosan solution and lyophilized, resulting in a chitosan sponge with a microchannel structure. To enhance its pro-coagulant and anti-infective characteristics, the sponge was then grafted with hydrophobic alkyl chains through a very effective Schiff-base reaction between the amine group of chitosan and the aldehyde group of dodecyl aldehyde. SEM analysis revealed that the resulting platform had a hierarchical porous structure, taking into consideration microchannels measuring 136.5 ± 17.8 μm and micropores measuring 8.3 ± 0.8 μm. The MACS had a compressive stress value of 23.0 ± 1.5 kPa, with increased structural porosity leading to increased mechanical strength. The platform can be compressed and shape-fixed after squeezing out the free water. Upon absorbing the water or blood, it recovered its original shape in under 4 s, giving a 100% recovery ratio. The MACS showed excellent pro-coagulant and hemostatic capabilities in lethally normal and heparinized rat and pig liver perforation wound models, where it substantially outperformed commercial hemostatic agents. Additionally, the MACS had robust anti-infective properties against *S. aureus* and *E. coli*, as determined by a contact-killing assay (Figure 14A).

Li et al. created a core-shell chitosan microsphere using high-voltage electrostatic drop technology. The core of the microsphere was filled with the antimicrobial polypeptide PonG1, which suppresses bacterial growth at wound sites, and bFGF, which stimulates cell migration, adhesion, and blood vessel formation. The chitosan microspheres acted as a controlled drug delivery system, releasing the drugs over a sustained period during the wound healing process. The animal study demonstrated that the microspheres loaded with PonG1 (3 mg/mL) and bFGF (300 μg/mL) reduced inflammation, enhanced blood vessel growth, and increased collagen deposition. The group treated with PonG1 (3 mg/mL) specifically decreased tissue damage and calcification resulting from the action of bacteria. The results of full thickness wound experiments revealed that the combination of bFGF and PonG1 accelerated wound healing by decreasing the inflammatory response and enhancing new blood vessels and deposition of collagen [223].

Bioglass can stimulate angiogenesis by encouraging fibroblasts and endothelial cells to release angiogenic growth factors such as VEGF and bFGF, which are important during the proliferation stage of skin wound healing [224]. Chen et al. reported a nanobioglass (nBG)-incorporated tri-layered chitosan/polyvinyl alcohol (PVA) nanofibrous membrane (nBG-TFM), fabricated via sequential electrospinning. The trilayer design aimed to offer different functions, such as hemostasis and antibacterial properties in the chitosan sublayer, moisture retention and exudate absorption in the chitosan-PVA mid-layer, and tissue generation promotion in the top-layer made of PVA-nBG (Figure 14B). The mechanical strength of the scaffold was improved by using 5% to 10% nBG, but using a higher percentage caused clumping. In vivo testing on rats with full-thickness skin wounds showed that the trilayer scaffold promoted faster wound healing, more extensive epithelium growth, increased capillary vessels, and greater extracellular matrix formation compared to the use of nBG alone. The platform also demonstrated good skin tissue regeneration effects in a diabetic mouse model of full-thickness skin injury by elevating growth factors like TGF-β and VEGF and reducing inflammatory cytokines like IL-1β and TNF-α [225].

**Figure 14 pharmaceutics-15-01313-f014:**
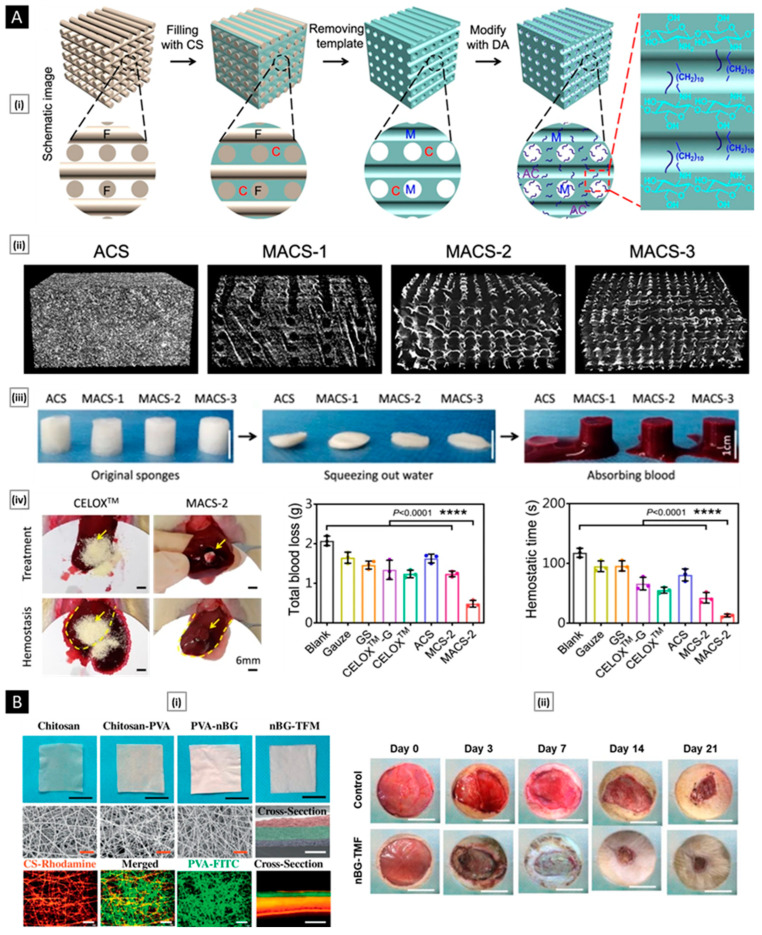
((**A**), i) Schematic illustration of the fabrication process of the MACSs. (Here, F: PLA microfiber; C: Chitosan; M: Microchannel; AC: Alkylated chitosan). ((**A**), ii) Micro-CT images depicting the macro and microstructure of the alkylated chitosan sponge (without microchannel structure) ACS and different preparations of MACS. ((**A**), iii) Macro photographs of the blood-triggered shape recovery of MACS. ((**A**), iv) Photographs of the hemostatic effect of CELOXTM and MACS in the normal rat liver perforation wound model (yellow arrow and dotted line represented the bleeding site and liver boundary, respectively) along with quantitative data of total blood loss and hemostatic time for different treatment groups. Reproduced with permission from [222], copyright Springer Nature 2021. ((**B**), i) Photographs (upper), SEM images (middle), and fluorescent images (bottom) of electrospinning membranes (scale bar: 1 cm, black; 10 μm, Re; 100 μm, white). ((**B**), ii) Representative images showing full-thickness skin defects treated with the petrolatum gauze (Control) and nBG-TFM at a predetermined time post-surgery (scale bar: 10 mm). Reproduced with permission from [225], copyright Elsevier 2019.

Peng et al. purposed the development of a material for achieving rapid and effective hemostasis in irregularly shaped, noncompressible, and high-pressure arterial bleeding wounds. To accomplish this, they created an ultrafast self-gelling and wet adhesive polyethyleneimine/polyacrylic acid/quaternized chitosan (PEI/PAA/QCS) powder. This was achieved by mixing aqueous solutions of PEI, PAA, and QCS in their optimized ratios, followed by direct plunging into liquid nitrogen for 10 min and immediate freeze-drying for 24 h. The dried solid was then grinded to obtain the final powder. When deposited on bleeding wounds, the PEI/PAA/QCS powder absorbs huge amount of blood and concentrates the coagulation factors. It transforms into a pressure-resistant hydrogel within 4 s upon hydration and forms an “in situ” adhesive physical barrier. The hydrogel further enhances hemostasis by aggregating blood cells and platelets at the injury site. The PEI/PAA/QCS powder was tested on several bleeding wounds in rats, including the liver and heart, high-pressure femoral artery, and tail vein and observed arrested bleeding in 10 s with no rebleeding after ten minutes. The excellent hemostasis of the PEI/PAA/QCS powder was further demonstrated against massive hemorrhage in the porcine spleen and liver in vivo, that are non-compressible organs having abundant supply of blood. The simplicity and ease of use of this platform make it a promising material for wound dressing (Figure 15) [226].

The use of hydrogel-based platforms in bioengineering has the potential to significantly enhance wound healing due to their key attributes. Hydrogels have the ability to retain moisture, reduce pain by cushioning the wound site and reducing friction during dressing changes, and minimize scarring by providing a moist and protective environment that supports the growth of healthy cells. Additionally, hydrogels are user-friendly and can easily be applied directly to the wound, making them a practical option for wound care. Several chitosan-based hydrogels having wound-healing potential have surfaced. Sun et al. described a green synthesis method for producing mussel-inspired adhesive chitosan hydrogels. In physiological conditions, these hydrogels showed exceptional mechanical strength and long-term dynamic adhesion to biological interfaces, making them promising candidates for wound healing dressings. Gallic acid was chemically attached to chitosan chains using a liquid-phase glow discharge plasma method. The properties of the hydrogel, including its wound healing and hemostatic capabilities, could be regulated by adjusting the chitosan-gallic acid composition and the self-crosslinking activity induced by oxygen. The hydrogel also demonstrated excellent biocompatibility and superior anti-bacterial activity against *E. coli* and *S. aureus* [227].

Diabetic wounds pose a major clinical challenge due to reduced migration and proliferation of fibroblasts, decreased angiogenesis, and the risk of microbial infection. To address these issues, Hao et al. developed a bio-multifunctional hydrogel consisting of benzaldehyde-terminated 4-arm polyethylene glycol (4-arm-PEG-CHO), carboxymethyl chitosan (CMCS), and basic fibroblast growth factor (bFGF). Diabetic wound healing was sped up attributable to the injectable, self-healing, and adhesive capabilities of the porous hydrogels, which also inhibited bacterial infection and promoted fibroblast responses. According to the rheological tests, the hydrogels had impressive structural stability, and the addition of bFGF did not affect their mechanical properties. The self-mending ability of the hydrogel was demonstrated by the negligible change in G′ after four cycles of step strain, making it suitable for various applications and preventing local stress damage. The carboxymethyl chitosan-based hydrogel had significant antibacterial activity, as shown in the colony-forming unit test. In vivo, the hydrogels demonstrated rapid hemostatic ability due to their strong wet-tissue adhesion. Furthermore, the hydrogels combined with bFGF showed biocompatibility comparable to a commercial BFX™ gel formulation and were beneficial for cell migration and proliferation, as demonstrated by their promotion of full thickness wound repair in diabetic mice. The hydrogels upregulated the production of Ki67, greatly facilitated the production of collagen fibers and the growth of epithelialization, induced angiogenesis by increasing the expression of CD34 and CD31, and promoted the formation of hair follicles (Figure 16) [228]. In a recent study, Toeh et al. created a personalized hydrogel wound dressing that can be manufactured and tailored using 3D printing. The chitosan methacrylate-based dressing was able to form a gel in a short time due to its ability to crosslink when exposed to UV light in the presence of a photo-initiator. The use of 3D printing allowed for different combinations of chitosan methacrylate and drugs to be utilized to produce wound dressings with various designs and drug dosages. The inclusion of an antibacterial agent did not impact the 3D printing performance of the material while significantly improving its antibacterial properties [229].

### 6.3. Miscellaneous Applications

#### 6.3.1. Gene Delivery

Gene delivery is the process of introducing genetic material, such as DNA or RNA, into cells to modify their function or behavior. Gene delivery has become a crucial tool in modern biotechnology and medicine as it allows for the targeted modification of cells and tissues, opening up new possibilities for disease treatment, gene therapy, and genetic engineering. It involves the use of gene delivery techniques to treat diseases caused by genetic mutations [230]. These mutations can be corrected by the introduction of normal genes into the patient’s cells, either to replace a defective gene or to supplement its function. Gene delivery therapy has shown great promise in the treatment of a wide range of genetic disorders, including cystic fibrosis, sickle cell anemia, muscular dystrophy, and various cancers. Challenges like susceptibility of gene degradation by nucleases in the bloodstream, lack of specificity towards targeted cells, and the inability of negatively charged genes to enter negatively charged cellular membranes make it impractical to directly deliver therapeutic genes (neither systemically nor locally) in the absence of a delivery vector [231].

Gene delivery vectors can be either viral or non-viral, each with its own advantages and limitations. Viral gene delivery vectors use modified viruses that have evolved to efficiently infect cells to deliver therapeutic genetic material. The have high transfection efficiency and provide stable gene expression, but their utilization is not preferred due to their significant safety concerns (unwanted inflammatory and immune responses, toxicity, immunogenicity, and substantial cost) [232]. Over the past few years, several non-viral gene delivery vectors have been developed owing to their ease of production and scale-up, biological safety, and high loading efficiency [233]. While their ability to enter cells and gene transfection efficiencies are lower than their viral counterparts, their flexibility makes them ideal vectors, as any limitations can be addressed by adequate physical or chemical modification [234].

The unique physiochemical and biological properties of modified chitosan derivatives have made them a popular option as non-viral vectors for gene delivery. Chitosan’s polycationic nature enables the formation of complexes with negatively charged nucleic acids via electrostatic interactions, providing protection from nucleases [235]. As cellular and nuclear membranes are also negatively charged, they interact with positively charged chitosan, allowing the complex to be taken up and relocated to the nucleus. Additionally, chitosan’s amino groups contribute to endosome escape through the “proton sponge effect”, which facilitates the intracellular release of the nucleic acid complex [11]. To enable efficient and effective gene delivery, critical chitosan characteristics must be tuned. First, for appropriate stabilization/protection and intracellular gene release after uptake, the optimal molecular range of chitosan should be between 65 and 170 kDa [236]. Secondly, the degree of deacetylation should be high as it translates to more primary amines, thereby increasing positive charge and eventually promoting transfection [237]. Lastly, the chitosan-based gene delivery system should have a high N/P ratio (fraction of chitosan nitrogen per gene phosphate). A higher N/P ratio improves the stability of the chitosan-gene complex and promotes better interaction with cells resulting in better transfection efficiency [238]. Keeping these considerations in mind, the subsequent section discussed some cutting-edge gene delivery platforms based on chitosan.

Ziminska et al. developed a “smart” gene delivery system by grafting chitosan on Poly(N-isopropylacrylamide) (PNIPAAm) which generates a biodegradable and thermos-responsive hydrogel (Cs-g- PNIPAAm). Different hydrogel batches with 10, 20, and 30 wt% of chitosan in relation to PNIPAAm were synthesized through a novel free radical polymerization method. RALA/pEGFP-N1 nanoparticles, composed of RALA peptide (a 30-amino acid arginine-rich peptide that can condense nucleic acids into nanoparticles) and a plasmid encoding the green fluorescent protein (GFP), were developed at an N/P ratio of 10 and subsequently incorporated into the hydrogel. Results from ^1^H-NMR, FTIR, and TGA confirmed the successful grafting of the copolymers. SEM analysis confirmed that the porosity of hydrogel remained unaffected by the incorporation of nanoparticles. Based on the results from rheological analysis and swelling studies, the authors reported that the hydrogel had a viscoelastic behavior that could be tuned by changing the dissolution media. The injectability of Cs-g-PNIPAAm at room temperature indicated the possibility of minimally invasive delivery to the target location. The hydrogel degraded gradually over a period of three-weeks, thereby facilitating the sustained release of plasmid-loaded nanoparticles. The released RALA/pEGFP-N1 was capable of transfecting NCTC-929 cells, demonstrating that the hydrogel had no effect on the stability of the delivered nucleic acid. Such a platform can be utilized to enable sustained delivery of growth factor-encoding genes in tissue regeneration therapy [239].

In a similar study, Akbari et al. incorporated histidine-conjugated trimethyl chitosan (HTMC) nanocomplex into injectable thermosensitive hydrogels for localized gene delivery. The authors screened pDNA/HTMC polyplex at different N/P ratios to optimize the nanoparticles. Histidine conjugation of trimethyl chitosan enhances the positive charge density at the surface of the chitosan backbone, improving the polymers’ capacity to condensate with nucleic acids. pDNA/HTMC complexes at an N/P ratio of 15 demonstrated the highest transfection efficiency. In addition, the complexation protected pDNA from interaction with plasma components and facilitated the efficient endosomal escape. Subsequently, pDNA/HTMC were loaded into two different hydrogel systems, hyaluronic acid/Pluronic F127 (HA/PF127) and Chitosan/PF127. Both systems showed optimal sol-gel phase transition behavior. Owing to the highly negative charge of HA, strong electrostatic interaction with positively charged chitosan resulted in nanoparticle aggregation in the HA/PF127 system. On the other hand, the nanoparticles were uniformly distributed in the chitosan/PF127 system. The platform facilitated the delivery of DNA to HEK293T cells, but a lower transfection efficiency was observed (compared to just nanoparticles), possibly owing to the additional barrier brought by the hydrogel for the diffusion of nanoparticles before entering the targeted cells [240].

Gene therapy has emerged as a promising approach for promoting the expression of neurotrophic proteins in neuronal cells, which can facilitate the restoration of damaged neurological function. This method has several advantages over the use of exogenous growth factors, including their limited half-life, immunogenicity, and pleiotropic effects. In the context of nerve injury, scaffolds can further enhance tissue regeneration by regulating the extracellular microenvironment. The combination of neural scaffolds with gene carriers can provide an effective means of guiding the growth of neurons, promoting functional recovery and achieving localized delivery of genes at the injury site, thereby increasing transfection efficiency. Wang et al. prepared Arginine-Glycine-Aspartate (RGD)-functionalized chitosan-graft-polyethyleneimine (RCP) gene vectors through the maleic anhydride and the carbodiimide methods that were electrostatically bound with c-Jun plasmids (pJUN). Subsequently, the vectors were loaded on poly-L-lactic acid/silk fibroin parallel fiber films to fabricate nerve scaffold (RCP/pJUN-PSPF@PGA), which could locally deliver c-Jun plasmids into Schwann cells via the mediation of RGD peptides and upregulate the localized expression of nerve growth factor and brain-derived neurotrophic factor. After the scaffold was bridged in sciatic nerve defect, the delivery of c-Jun plasmids from RCP/pJUN-PSPF@PGA facilitated Schwann cells to sustain the expressions of neuronal and vascular endothelial growth factor in the injury field, promoting myelination, axonal growth and microvascular generation and nerve regeneration, muscle reinnervation and functional recovery (Figure 17) [241].

Gene-activated matrix technology (GAM) is a breakthrough tissue engineering technique for wound healing that may be described as a local gene delivery system that can not only maintain a moist environment but also enhance the concentration of active components. After gene transfer, the recombinant cytokines are produced in situ by endogenous wound-healing cells in minimal levels but for an extended length of time, resulting in consistent tissue regeneration [242]. Based on this concept, Chang et al. fabricated the mVEGF165/TGF-β1 gene-loaded N-carboxymethyl chitosan/sodium alginate hydrogel (NS-GAM) and studied its effect on promoting deep second-degree burn wound repair. The hydrogel pores had an average diameter of 100 μm, and the porosity was determined to be 50%. The hydrogel was appropriate for cell attachment and proliferation, as evidenced by SEM and CLSM images. In the in vitro studies on NIH3T3 cells, the NS-GAM could maintain continuous expression for at least 9 days, demonstrating long-term gene release and expression effects. To evaluate the wound healing potential, a deep second-degree burn wound model was created on the backs of Wistar rats. The results indicated that the NS-GAM demonstrated improved wound healing with prolonged high expression of VEGF and TGF- β1 proteins. Moreover, a higher degree of neovascularization and expression of CD34 was observed in the NS-GAM group within 21 days. The histological analysis revealed that the platform is safe for the surrounding tissue and effectively promotes epithelialization and collagen regeneration. (Figure 18) [243].

One of the most researched domains in gene delivery is its application for the treatment of cancer. Cancer is caused by genetic abnormalities in any gene that encodes cell cycle proteins, or by somatic mutations in upstream cell signaling pathways [244]. Cancer gene-therapy involves the transfer of nucleic acids into tumor cells to eliminate or reduce tumor burden by direct cell-killing, immunomodulation, or correcting genetic errors to reverse the malignant state. Gene therapy has the potential to revolutionize cancer therapy as its selectivity and cell-specificity make it a lucrative option over traditional treatment modalities like chemotherapy and radiotherapy [245]. Several chitosan-based platforms have been employed for cancer gene-therapy. Zhang et al. developed hyaluronic acid (HA)-modified chitosan nanoparticles (sCS NPs-HA) for targeted delivery of cyanine 3 (Cy3)-labeled siRNA in non-small cell lung cancer. CD44 is a complex transmembrane glycoprotein that has been discovered as a HA receptor as well as a human cell homing receptor. It is a HA-binding glycoprotein on the cell surface that has a role in cancer invasion and metastasis. Many kinds of cancer cells are known to overexpress CD44 in comparison to normal cells, indicating its relevance as a potential therapeutic target in cancer. To exploit this, the authors modified chitosan with HA and loaded siRNA to silence B-cell lymphoma/leukemia 2 (BCL2) oncogene (which can inhibit cancer apoptosis and cell death caused by various cytotoxic factors). Briefly, chitosan nanoparticles were developed by ion gelation using sodium tripolyphosphate, followed by conjugation with HA via charge adsorption. Cy3-labeled siRNA targeting BCL2 was incorporated by mixing it with sodium tripolyphosphate during preparation. The prepared particles had a size of around 130 nm with up to 94% encapsulation efficiency of siRNA. The particles were biocompatible and showed reasonable stability in vitro. sCS NPs-HA displayed a high transfection efficiency of 36.4% at 48 h into CD44 expressing A549 cells. In a mouse xenograft tumor model, the authors observed significant downregulation of the BCL2 gene that coincided with outstanding anti-tumor effects in treated mice [246].

In a similar study, self-cross-linkable chitosan-hyaluronic acid dialdehyde nanoparticles were developed by Liang et al. for CD44-targeted siRNA delivery for the treatment of bladder cancer. Bladder cancer therapy is significant since it does not have as much blood supply as other organs such as the liver, pancreas, lung, etc., so an active targeted delivery system is necessary to inhibit it. High CD44 expression as a HA receptor is associated with a higher clinical stage, lower treatment response rates, and shorter survival rates in bladder cancer. The synthesized chitosan-HA-siRNA nano-complexes were 120 nm in size and had a siRNA loading of >95% to successfully target bladder cancer and decrease BCL2 expression at the post-translational level. The endosomal escape test and blood biocompatibility testing revealed that siRNA was released from the lysosome at 6 h without distraction and with < 5% hemolysis, demonstrating that nano-complexes are extremely biocompatible. Furthermore, blood clotting revealed that CS-HA-siRNA had no detrimental effect on the activities of plasma proteins and platelets, confirming the efficiency of nanocarriers. The nano-complex reduced tumor volume and weight by more than threefold and twofold, respectively, as compared to control and free siRNA, indicating that siRNA was successfully delivered to the cancer location and that BCL2 was suppressed. The low level of BCL2 in nude mice was also measured using a western blot analysis to further validate the nanocarrier’s therapeutic effectiveness. The kidney, tumor, and liver had the largest accumulation, metabolism, and biodistribution of CS-HA-siRNA nano-complexes, and the occurrence of many CS receptors in the kidney might be the major cause for the highest concentration of targeted nanoparticles. When compared to the control, the tumor contained more than three times the amount of CS-HA-siRNA.

Bladder cancer treatment is challenging due to its low blood supply, which necessitates an active targeting delivery system to suppress it. The high expression of CD44 in bladder cancer, as a receptor for HA, is directly related to a higher clinical stage, lower treatment response rates, and lower survival rates. The chitosan-HA-siRNA nano-complexes synthesized had a size of 120 nm, with siRNA loading greater than 95%, making them effective in suppressing BCL2 expression at the post-translational level. The endosomal escape test and blood biocompatibility tests depicted that the siRNA was released from the lysosome at 6 h without any disturbance and had less than 5% hemolysis, indicating excellent biocompatibility of the nano-complexes. Furthermore, blood clotting assays demonstrated that the platform did not negatively impact the functions of plasma proteins and platelets. The nano-complexes showed more than three-fold and two-fold decreases in tumor volume and weight, respectively, compared to control and free siRNA, indicating successful targeted delivery of siRNA to the cancer site and suppression of BCL2. The western blot assay in nude mice further confirmed the therapeutic potency of the nanocarrier, demonstrating low levels of BCL2. A biodistribution study shows high accumulation in tumor and metabolic/excretory organs (kidney, liver). Tumor accumulation was roughly three-fold higher for chitosan-HA-siRNA compared with control (Figure 19A) [247].

Despite the promising advantages, the poor buffering capacity of the chitosan-based system remains a significant intracellular barrier. The crucial step for effective gene delivery is the escape of nucleic acid-chitosan complexes from the endosome into the cytoplasm. The proton sponge mechanism explains that upon the entry of nucleic acid-cationic polymer complexes into the cell, they get enclosed by the endosomal membrane. As the endosomal pH changes, the cationic polymer becomes protonated, causing water to diffuse into the endosome. This leads to an increase in osmotic pressure, which eventually disrupts the membrane, allowing the nucleic acid to escape into the cytoplasm. However, chitosan’s weak endo-lysosomalytic proton sponge effect, owing to its low buffering capacity, results in poor gene delivery efficiency. To address this, Huang et al. developed a series of modified chitosan polycations with alkylamines (AA-CS) to enhance the buffering capacity of chitosan. Unlike conventional amino group modifications in chitosan, the authors used a cyclo-opening synthetic route of glucosamine units with periodate oxidized reaction, followed by reductive amination with propylamine (PA), diethylaminopropylamine (DEAPA), and N, N-dimethyldipropylenetriamine (DMAPAPA). This modification rearranged the primary, secondary, and tertiary amines of the chitosan matrix. The AA-CS polycations inherited excellent biocompatibility from chitosan and showed an improved ability to bind to nucleic acids. The authors analyzed the buffering capacity of AA-CS under endosomal conditions using acid-base titration. AA-CS showed higher buffering capacities than chitosan as a result of its multiple amino groups, with DMAPAPA-chitosan showing the highest buffering capacity. The DMAPAPA-chitosan complex demonstrated superior cell internalization and endosomal escape, resulting in enhanced transfection efficiency (Figure 19B). Using A549 tumor cells, the authors showed that the DMAPAPA-chitosan complex effectively transported the therapeutic plasmid p53 into the cells, inducing apoptosis at high levels when functionalized with the plasmid. Treatment of tumor-bearing mice with the DMAPAPA-chitosan/p53 system resulted in significant inhibition of tumor growth (inhibitory rate of 59.0% vs. control group) and a decrease in tumor mass, indicating the platform’s potential for solid tumor therapy due to its increased tumor penetration and efficient in vivo transfection [248].

#### 6.3.2. Bioimaging

Molecular imaging provides the capacity to assess biological and metabolic processes in live organisms without intrusive methods. Many non-invasive procedures have been developed for uses ranging from clinical diagnostics to cellular biology research and drug development since the advent of X-ray technology. Throughout clinical and preclinical drug research, these cutting-edge technologies have the potential to enhance our understanding of illnesses and therapeutic action. In contrast to traditional readouts such as immunohistochemistry, molecular imaging techniques may be done in the intact organism with an excellent spatial and temporal resolution, making them ideal for in vivo studies of biological processes [249]. In addition, molecular imaging permits repeated, noninvasive examinations of the same living subject at multiple time periods utilizing identical or alternative biological imaging tests. This statistical strength of longitudinal investigations not only decreases the number of test animals needed but also saves expenses [250]. Molecular imaging employs probes and intrinsic tissue properties as the source of image contrast and offers prospective advantages such as comprehending integrated biology, early disease identification and characterization, and therapy evaluation [251]. The flexibility of chitosan facilitates its combination with various organic and/or inorganic materials to develop novel nanocomposite platforms having ideal physicochemical and functional attributes for use in bioimaging [252].

Tan et al. reported a chitosan-based core-shell platform for the pH-triggered release of anticancer drugs and near-infrared (NIR) bioimaging. The platform comprised oleic acid capped NIR photoluminescent Ag_2_S quantum conjugated with chitosan through esterification. Doxorubicin, an anticancer agent with a propensity for hydrophobic oleoyl groups, was entrapped to develop Ag_2_S(DOX)@CS nanospheres. In vitro and in vivo testing revealed that the nanospheres showed high anticancer efficacy and preferentially released doxorubicin at the tumor’s lower pH. More importantly, the strong NIR signal resulting from the entrapped Ag_2_S quantum dots aided in monitoring the distribution of nanospheres in real-time. Ag distribution in tumor-bearing mice was studied after injection of Ag_2_S(DOX)@CS nanospheres at various time intervals. The nanospheres were found to get accumulated in the tumor due to the EPR effect, with higher levels in the liver and spleen compared to the stomach, heart, and kidney. The Ag levels in the tumor and organs started to decrease after 6 h upon metabolism, but the decrease in the tumor was less noticeable, indicating extended retention of the nanospheres. Non-invasive in vivo NIR imaging of nude mice was used for examining the nanospheres’ distribution in a living body. The fluorescence from the tumor was stronger compared to other areas in the mice bodies and, after 12 h post-injection, indicated accumulation of the nanospheres in the tumor through the EPR effect. Fluorescence signals were observed in the mice bodies even after 24 h, but with decreased intensities. However, the fluorescence intensity remained high at the tumor site, suggesting that the nanospheres could remain there for an extended time (Figure 20A) [253].

Aggregation-induced emission (AIE) is a photophysical effect in which non-luminescent molecules in solution can become brightly luminescent upon aggregation owing to the restricted intramolecular motions. This means that when the molecules come together, the nonradiative decay pathways that usually cause low fluorescence quantum yields are suppressed, leading to an increase in radiative decay and a higher quantum yield of fluorescence [254]. Several chitosan-based systems working on the AIE effect have been explored for biological imaging. Shi et al. developed a redox-responsive polymeric nanocarrier with AIE-mediated bioimaging ability. The system was fabricated from a biotinylated chitosan-modified amphiphilic polymer, wherein the hydrophilic chitosan was associated via a hydrophobic tetraphenylethylene (responsible for AIR) unit through a disulfide bond. The polymer self-assembles into a nanosphere (TPE-bi(SS-CS-Bio)). The presence of biotin enhanced the nanocarrier’s cellular uptake, and subsequent exposure to a high level of glutathione caused rapid disassembly and release of constituents. AIE feature allowed tracking upon distribution into cells with a time-dependent increase in observed fluorescence (Figure 20B) [255]. In a subsequent study, the group reported etyl 4-formylbenzoate alkyl and 4-(2-hydroxyethoxy) benzophenonesalicylaldazide modified biotinylated chitosan that self-assembled into nano-sized micelles with superior AIE-mediated bioimaging properties [256].

Zu et al. reported a facile alkali-assisted one-pot hydrothermal method for the development of polysaccharide-based nanoparticles with the use of starch or chitosan as raw materials. TEM analysis showed the average size of nanoparticles was around 14 and 75 nm for starch and chitosan, respectively. The surface of these particles contained hydroxyl or amino groups, resulting in high water solubility. When exposed to ultraviolet excitation, the particles demonstrated strong fluorescence with excitation-dependent emission behavior. Compared to traditional fluorescent compounds such as fluorescein and rhodamine B, the polysaccharide-based nanoparticles displayed high stability against photo-bleaching. The fluorescence intensity of the nanoparticles was quenched by certain oxidative metal ions, including Hg(II), Cu(II), and Fe(III). The nanoparticles showed maximum fluorescence intensity at physiological pH and excellent biocompatibility in B16-F10 cells at concentrations less than 10 mg/mL. Their ability to serve as fluorescent probes was evaluated in B16-F10 melanoma cells and guppy fish [257].

**Figure 20 pharmaceutics-15-01313-f020:**
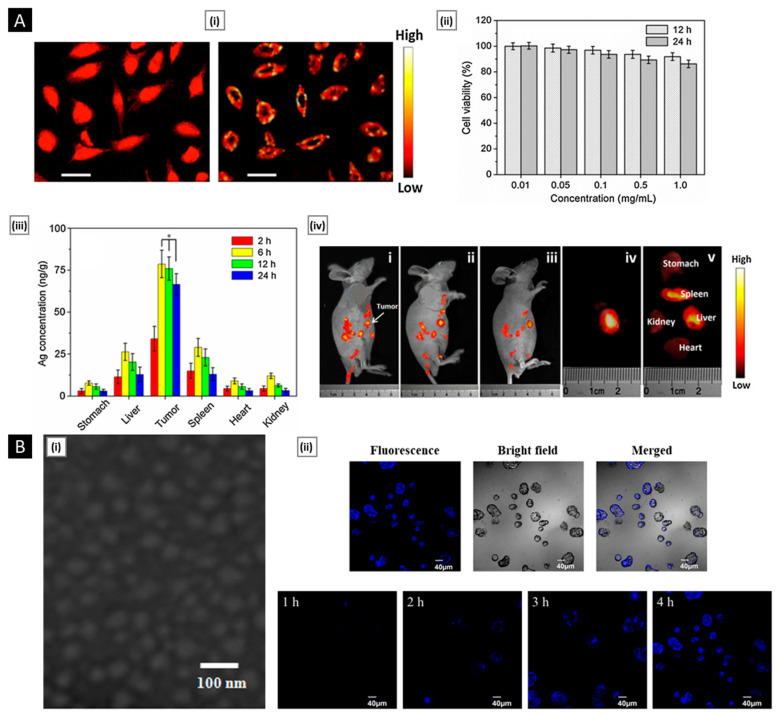
((**A**), i) Fluorescence image excited at 488 nm and NIR image excited at 808 nm of HeLa cells incubated with Ag_2_S(DOX)@CS nanospheres for 12 h. The fluorescence image was acquired in a wavelength window between 560 and 600 nm (Scale bar: 25 μm) ((**A**), ii) Viability of HeLa cells incubated with different concentrations of Ag_2_S@CS nanospheres ((**A**), iii) ICP-MS analysis of tumor and five major organs of the mice sacrificed at different time points (statistical significance: * *p* < 0.05). ((**A**), iv) In vivo NIR images of a nude mouse at 6 h (i), 12 h (ii), and 24 h (iii) after injection of the Ag_2_S(DOX)@CS nanospheres; ex vivo NIR image of the tumor (iv) and the organs (v) harvested from the sacrificed nude mouse. Reproduced with permission from [140], copyright Elsevier 2017. ((**B**), i) TEM images of blank TPE-bi(SS-CS-Bio) micelles (Scale bar: 100 nm). ((**B**), ii) Confocal laser scanning microscopy images of MCF-7 cells after incubation with TPE-bi(SS-CS-Bio) for 4 h (Scale bar: 10 μm); and after incubation at different time points (1, 2, 3, and 4 h). Reproduced with permission from [255], copyright Elsevier 2021.

### 6.4. Other Applications

#### 6.4.1. Vaccination

Chitosan derivatives, specifically trimethyl chitosan, are widely hypothesized to function as vaccine adjuvants. It can activate the innate immune system through the stimulation of Toll-like receptors and other pattern-recognition receptors. This results in the production of cytokines and chemokines that promote the recruitment of immune cells to the site of antigen exposure. Additionally, it can modulate the immune system by promoting a Th1 response and inhibiting a Th2 response. This leads to a stronger cellular immune response and a reduced risk of allergic reactions [258].

In one study, Wang et al. explored the use of chitosan hydrochloride salt stabilized emulsion as a vaccine adjuvant. Since conventional emulsion adjuvants are stabilized by non-ionic surfactants, their electroneutrality limits the loading efficiency of negatively charged antigens. The chitosan salt facilitated the loading of negatively charged Ovalbumin (adsorption rate of up to 97.99 ± 0.33%) and significantly enhanced humoral immunity by boosting recognition/uptake by APCs (via charged interaction) [259]. Chitosan-based systems have also been widely investigated as vectors for intranasal vaccine delivery. Mosafer et al. developed alginate-coated chitosan or trimethyl chitosan nanoparticles loaded with inactivated PR8 influenza virus (via direct coating) for nasal immunization. When tested in BALB/c mice, PR8-trimethylchitosan-alginate formulation elicited a higher ratio of IgG2a/IgG1 antibody titer (which promotes Th1 immune response) for immunization against the influenza virus [260]. Using the ionic crosslinking method, Lin et al. developed a chitosan nanoparticle system with high antigen loading and mucosal absorption capacity for sustained immunization effect. Using bovine serum albumin as a model antigen, the authors demonstrated that the particles with excellent biostability and mucosal absorption could efficiently stimulate the proliferation of lymphocytes and the release of associated pro-inflammatory substances, therefore stimulating particular mucosal and systemic immune responses when administered nasally [261].

In an animal model, Zare et al. assessed the immunological effectiveness of multi-subunit vaccines incorporating chitosan- or trimethyl chitosan-coated PLGA nanospheres to activate cell-mediated and mucosal responses against Mycobacterium tuberculosis. The PLGA vaccines having tri-fusion protein from three Mtb antigens were developed and subcutaneously or nasally administered. The vaccines induced a shift of Th1/Th2 balance toward Th1-dominant response, and all designed PLGA vaccines were able to elicit mucosal IgA, IgG1, and IgG2a production and secretion of IL-4, IFN-γ, IL-17, and TGF-β cytokines. The modified PLGA NPs using trimethyl chitosan cationic polymer was more efficient in elevating Th1 and mucosal responses than the normal chitosan coated PLGA nanospheres [262].

#### 6.4.2. Cosmeceuticals

Cosmetics refer to a range of products and substances that are applied to any part of the body for the purpose of enhancing, maintaining, or altering one’s appearance. Although synthetic compounds are often used as an active ingredient in cosmetic products, their long-term use can lead to skin irritation, itching, phototoxicity, and photo allergy. Due to current international regulations and increased demand for eco-friendly products, the cosmetic industry has shifted towards “green cosmetics” and has been conducting extensive research to find natural ingredients to replace traditional petrochemical-derived ones [263]. Chitosan, owing to its natural source and biocompatibility, has attracted significant attention in the cosmetic industry due to its unique qualities such as a natural humectant and moisturizer, a rheology modifier, and a formulation stability enhancer. Additionally, chitosan reduces the requirement of preservatives. Of late, some interesting cosmetic applications of the chitosan-based system have been reported [264].

Libio et al. employed chitosan films neutralized in citrate buffer (without glycerol as plasticizer) as a platform for stratum corneum exfoliation. The films made use of chitosan’s bioadhesive property, which helped to reduce cell cohesion and promote cell detachment. This, in turn, aids in preventing skin aging by stimulating cell proliferation and regenerating the corneum layer. Moreover, the platform facilitated the thickening of the epidermis and dermis by accumulating dermal glycosaminoglycans, leading to an increase in collagen density and a reduction of lines and wrinkles. When applied to pig skin, the film showed a significant increase in hydration within 10 min compared to untreated skin [265]. Chen et al. developed a new ingredient for cosmetic creams that consists of a combination of quaternized carboxymethyl chitosan (QCOM) and organic montmorillonite, which has potential anti-aging properties. The study found that the ideal QCOM composite showed superior moisture absorption and retention abilities compared to hyaluronic acid, and the QCOM solution demonstrated effective UV protection. Additionally, the QCOM-infused cream met the standards for hygiene in cosmetic products, causing minimal skin irritation while effectively retaining moisture in the outermost layer of human skin [266]. Petrick et al. reported the creation of a chitosan/TiO_2_ nanocomposite through the wet impregnation method, which was intended for use as an antibacterial sunscreen. Their FT-IR analysis revealed that the amino group of chitosan formed a bond with the oxygen in TiO_2_, leading to a decrease in the band gap energy. This caused TiO_2_ to become more active when exposed to UV rays. Additionally, the resulting sunscreen exhibited a remarkable ability to eliminate up to 99.7% of bacteria within a 2 h period [267].

## 7. Conclusions and Future Perspective

In conclusion, the diverse properties and applications of chitosan make it a promising material in the fields of drug delivery and biomedical research. Chitosan can be extracted from chitin through various techniques, and the resulting chitosan can be modified to enhance its bioactivities and properties. The modification of chitosan has enabled researchers to tailor its properties to suit specific applications, allowing for the development of novel chitosan-based materials. Chitosan-based drug delivery systems have been developed for oral, ophthalmic, transdermal, nasal, and vaginal routes of administration, providing targeted and sustained release of drugs. Additionally, chitosan has been used as a wound dressing material due to its antibacterial and hemostatic properties, promoting wound healing by accelerating the formation of new tissues and preventing infections. Chitosan has also been used as a scaffold material in tissue engineering due to its biocompatibility and ability to support cell growth and differentiation. Thus, the review provides details about the biomedical applications of chitosan, including bone regeneration, cartilage tissue regeneration, cardiac tissue regeneration, corneal regeneration, periodontal tissue regeneration, and wound healing. The use of chitosan in gene delivery, bioimaging, and cosmeceuticals has shown remarkable potential, highlighting the versatility of this material. Their potential in drug delivery and biomedical research is vast, and their use in numerous applications shows great promise for the future.

Despite the significant progress made in chitosan-based materials, challenges such as stability and reproducibility still need to be addressed. Further research is required to investigate the full potential of chitosan-based materials in various biomedical applications, as well as to explore the potential of chitosan in combination with other materials. Overall, the versatility of chitosan, along with its biocompatibility, biodegradability, and unique properties, make it a promising material for the development of innovative drug delivery and biomedical applications that have the potential to revolutionize the field of medicine.

## Figures and Tables

**Figure 1 pharmaceutics-15-01313-f001:**
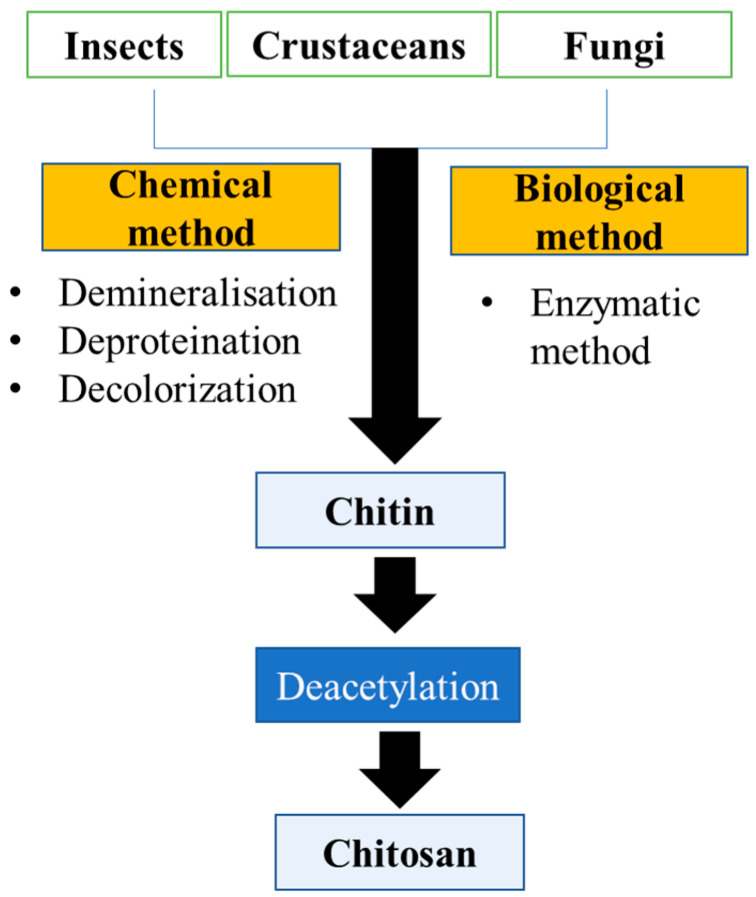
Extraction of chitin and chitosan.

**Figure 2 pharmaceutics-15-01313-f002:**
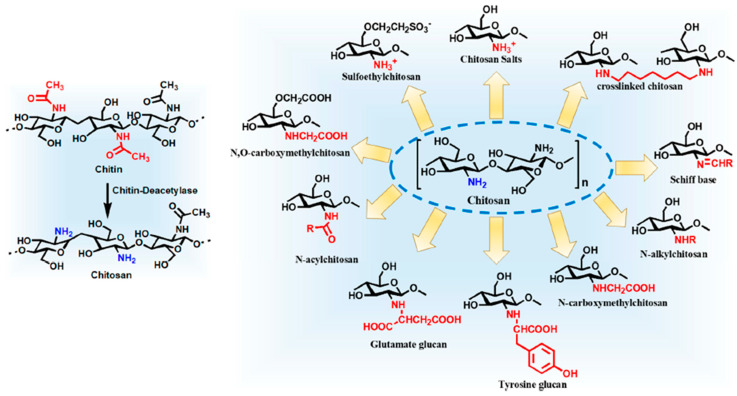
Functionalized chitosan derivatives. Adapted from reference [88].

**Figure 3 pharmaceutics-15-01313-f003:**
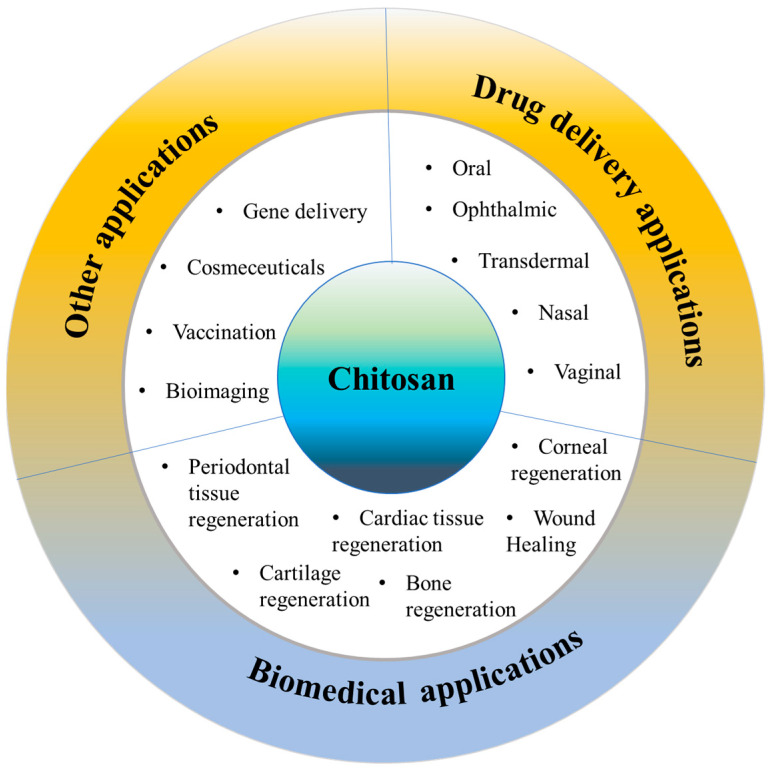
Various applications of chitosan.

**Figure 6 pharmaceutics-15-01313-f006:**
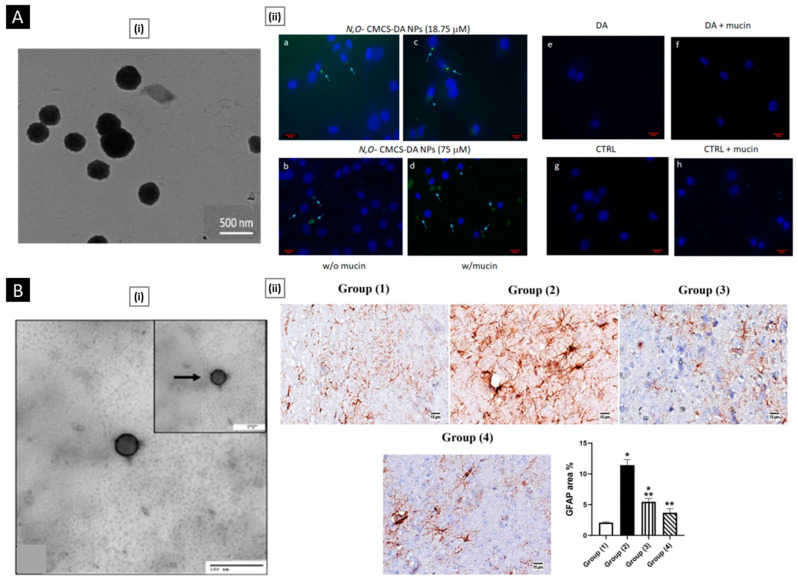
((**A**), i) Cryo-TEM images of N-O-carboxymethyl chitosan-dopamine amide conjugate nanoparticles (Scale bar: 500 nm). ((**A**), ii) Epifluorescence microscopy of olfactory ensheathing cells incubated with FITC-loaded N-O-carboxymethyl chitosan-dopamine amide conjugate nanoparticles at dopamine concentrations of 18.75 (a,c) and 75 μM (b,d), and FITC-loaded N-O-carboxymethyl chitosan-dopamine amide conjugate nanoparticles 75 μM (e,f), incubated with olfactory ensheathing cells in the presence or absence of mucin for 2 h and then evaluated by epifluorescence microscopy. Controls (CTRL) were cells incubated with medium only in the presence or absence of mucin (g,h). Arrows indicate nanoparticles in close vicinity of nuclei as dots, while arrowheads point to more diffuse perinuclear staining (Scale bar: 10 μm). Reproduced with permission from [162], copyright MDPI 2019. ((**B**), i) TEM image of luteolin-loaded chitosomes (150,000× magnification). Arrows point to the chitosan coating layer. ((**B**), ii) Photomicrograph immunohistochemistry of GFAP expression in brain tissue. Group 2 (disease control) shows a marked expression of GFAP; however, the morphological difference between the two treated groups was not observed. Here, significant difference was considered at *p* < 0.05, *. Statistically significant difference from the normal group at *p* < 0.05, **. Reproduced with permission from [163], copyright MDPI 2022.

**Figure 8 pharmaceutics-15-01313-f008:**
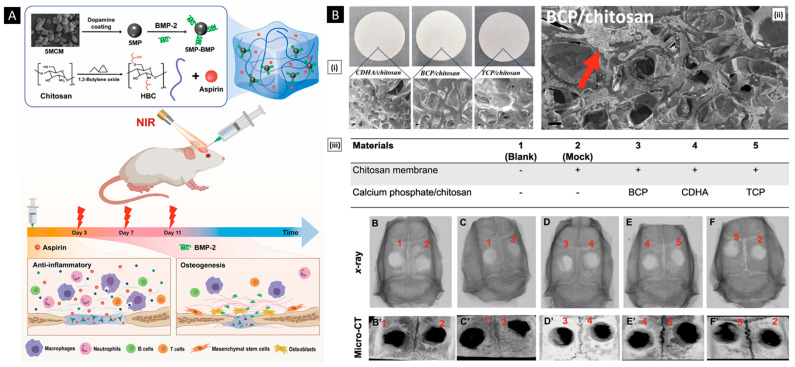
(**A**) Synthesis of the dual-responsive hydrogel composite by incorporating NIR-responsive polydopamine-coated magnesium–calcium carbonate microspheres into a thermo-responsive hydroxy butyl chitosan hydrogel and its application for sequential Aspirin/bone morphogenetic protein-2 delivery. Reproduced with permission from [181], copyright Elsevier 2022. ((**B**), i) Photographic and SEM images of porous hybrid calcium phosphate/chitosan membranes, scale bar: 10 μm. ((**B**), ii) SEM image showing the osteoblast cell growth and formation of mineral-surrounded clusters, indicated by arrows. Scale bar: 10 μm. ((**B**), iii) X-ray and micro-CT imaging in Sprague-Dawley rats (21 days post-surgery). All rat skulls were punched with two holes having 4 mm diameter and then covered with different membranes listed in the top panel. B–F panel are X-ray images while B’–F’ are micro-CT images as explained in figure. B–F and B’–F’ correspond to materials 1–5 as shown in panel above X-ray images. Reproduced with permission from [182], copyright Elsevier 2019.

**Figure 9 pharmaceutics-15-01313-f009:**
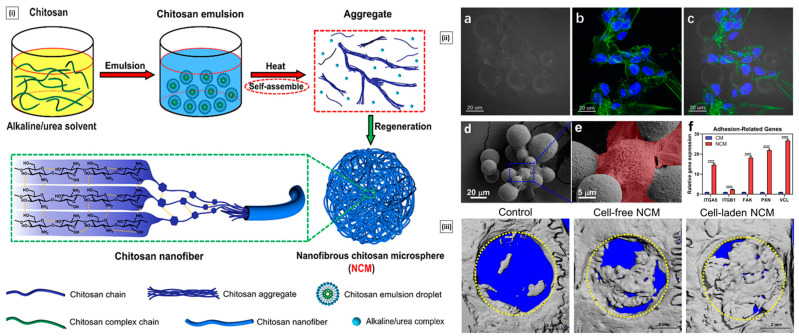
(**i**) Graphical illustration for the mechanism of the self-assembled microspheres. (**ii**) Bright-field [a], florescent [b], mix CLSM [c], and SEM images [d, e] of MC3T3-E1 cells co-cultured with microspheres after 3 days. The graph shows the adhesion-related gene expression of MC3T3-E1 cells on chitosan microspheres without nanofibers and nanofibrous chitosan microspheres (Here, **** *p* < 0.0001) [f]. (**iii**) Micro CT reconstruction images of the bone defect. Reproduced with permission from [184], copyright Elsevier 2022.

**Figure 10 pharmaceutics-15-01313-f010:**
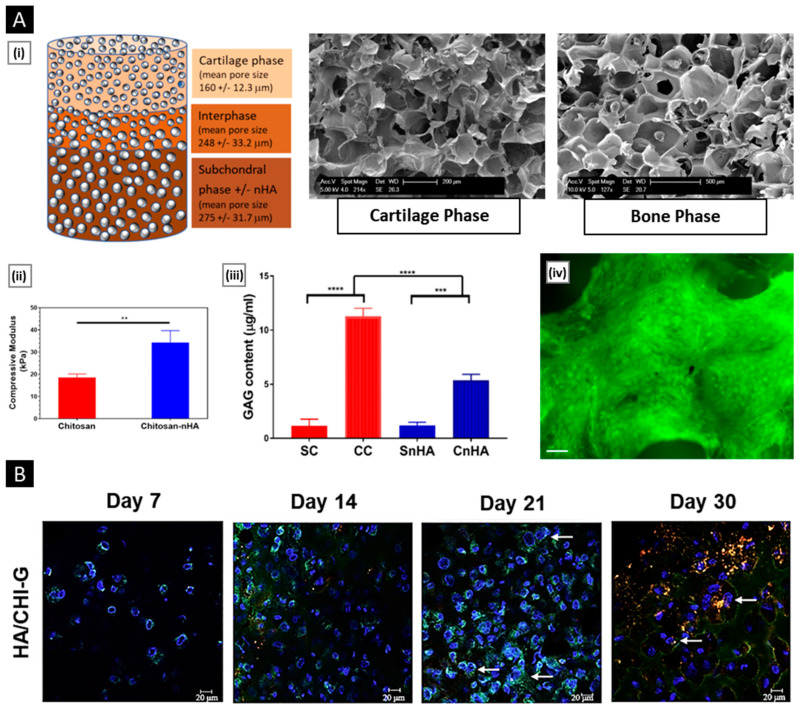
((**A**), i) Schematic image of the scaffold showing mean pore size in each phase with corresponding SEM images showing internal pore structure of the composite scaffolds in the cartilage and bone phases. ((**A**), ii) Comparison of compressive modulus of chitosan-only and chitosan-nano-hydroxyapatite composite scaffolds. ((**A**), iii) Sulphated glycosaminoglycans measured in mesenchymal stem cells-seeded chitosan scaffolds exposed to chondrogenic culture conditions. (Here, SC = chitosan scaffold in standard culture, CC = chitosan scaffold in chondrogenic medium, SnHA = chitosan-nHA scaffold in standard culture, CnHA = chitosan-nHA scaffold in chondrogenic medium; *p* values ** ≤ 0.01, *** ≤ 0.001, **** ≤ 0.0001). ((**A**), iv) Fluorescent microscopy showing MSCs seeded onto chitosan-nHA composite scaffolds after 14 days in osteogenic medium. Reproduced with permission from [188], copyright Elsevier 2022. (**B**) Immunofluorescent staining of bone marrow mesenchymal stem cells encapsulated in hyaluronic acid/chitosan coacervate-based scaffolds at different days of chondrogenic differentiation. Blue represents cell nuclei, green represents COL2A1, red represents ACAN and Phalloidin. Reproduced with permission from [190], copyright Elsevier 2021.

**Figure 11 pharmaceutics-15-01313-f011:**
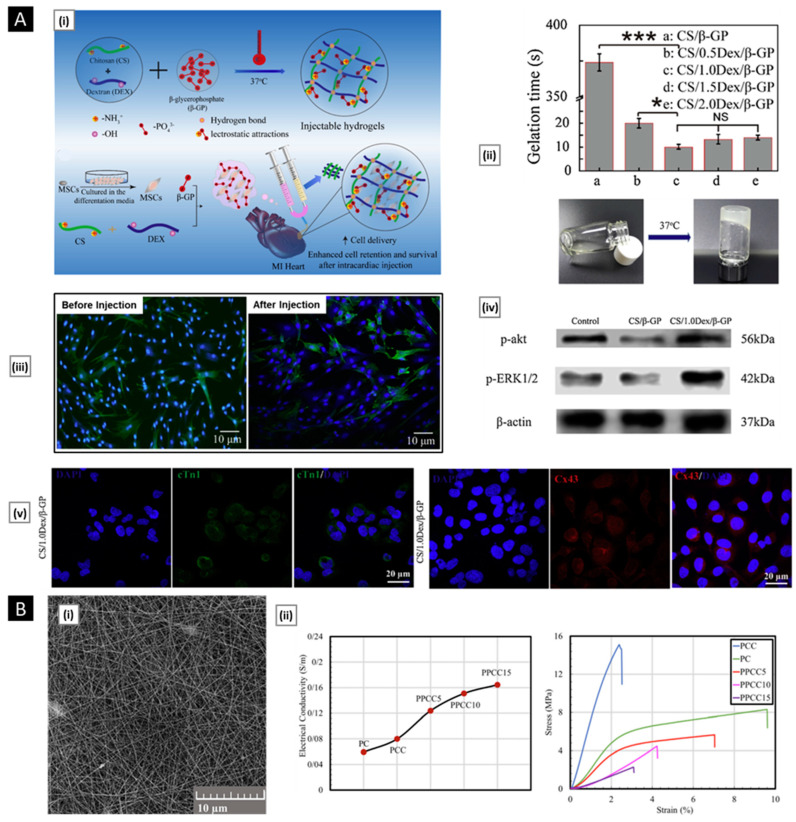
((**A**), i) Schematic depicting the synthesis route and UCMSCs encapsulated in CS/Dex/β-GP hydrogel for use in cardiac repair applications. ((**A**), ii) Gelation time of CS/Dex/β-GP hydrogel with varying concentrations of dextran, here significant differences were defined as *p* values * ≤ 0.05, *** ≤ 0.001. ((**A**), iii) Confocal microscope images showing the morphology of UCMSCs in hydrogels before and after injection. ((**A**), iv) Representative Western blot assay for detecting the levels of p-ERK and p-ERK1/2 of UCMSCs cultured in hydrogels for 2 days. ((**A**), v) cTnI (green) and Cx43 (red) expression of UCMSCs in cultured hydrogels, cell nuclei were stained by Hoechst (blue). Reproduced with permission from [196], copyright Elsevier 2020. ((**B**), i) SEM image of polypyrrole/chitosan/collagen electrospun nanofiber scaffold. ((**B**), ii) The electrical conductivity and stress–strain curve of different nanofibrous scaffolds. Reproduced with permission from [197], copyright Elsevier 2019.

**Figure 12 pharmaceutics-15-01313-f012:**
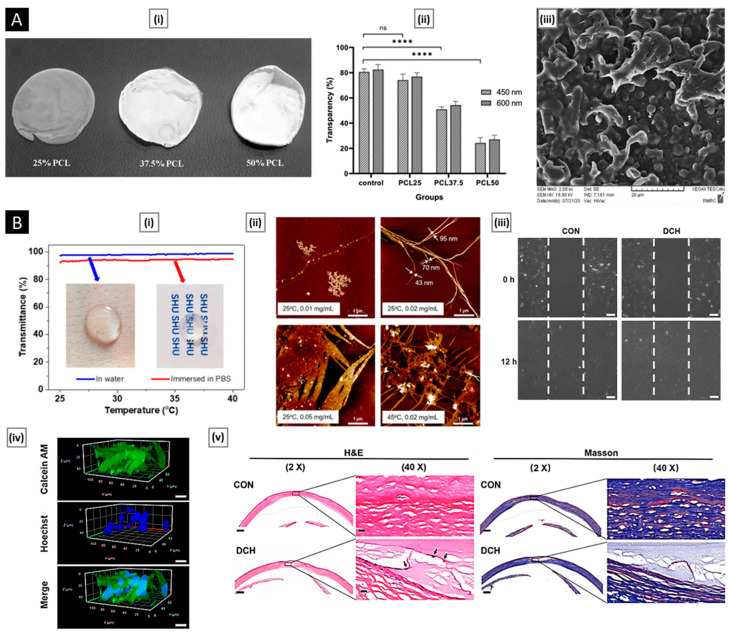
((**A**), i) Macroscopic photographs of dried composite membranes showing a decrease in visual transparency with increasing PCL content. ((**A**), ii) The effect of PCL content on light transmittance of the composite membranes (significant differences were defined as *p* values **** ≤ 0.0001). ((**A**), iii) Representative SEM images of corneal epithelial cells cultured on CSNP/PCL 50/25 for 5 days. Reproduced with permission from [204], copyright Springer Nature 2021. ((**B**), i) Transmittance of DC hydrogel in water and in PBS (2 wt%) between 25 and 40 °C (λ = 700 nm). ((**B**), ii) AFM images of self-assembled DC hydrogel at different temperatures. ((**B**), iii) Representative micrographs of DC hydrogel stimulating corneal stromal cell migration after 12 h in the scratching assay (100× magnification, Scale bar: 100 μm). ((**B**), iv) Confocal laser scanning microscopy graphs of rabbit corneal stromal cells cultured in hydrogel at day 7 (Scale bar: 20 μm). ((**B**), v) H&E and Masson staining images of corneal stroma defect with and without DC hydrogel at 4 weeks after surgery (Scale bar: 500 μm for 2×, 20 μm for 40×; Black arrows point to keratocytes). Reproduced with permission from [205], copyright American Chemical Society.

**Figure 13 pharmaceutics-15-01313-f013:**
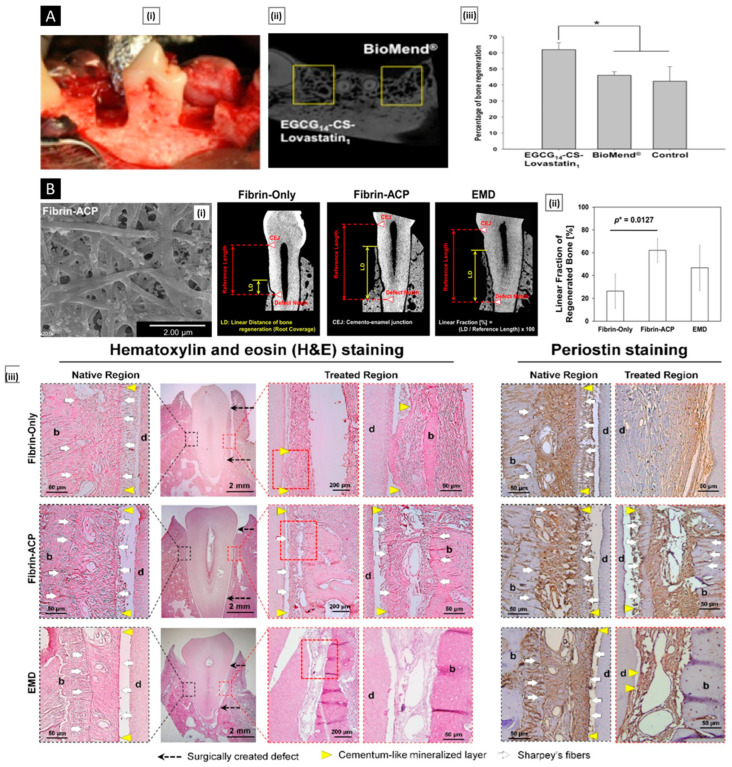
((**A**), i) Image showing the one-walled defects created surgically at mesial and distal sides of maxillary first premolar ((**A**), ii) New bone formation seen in defects of trilayer functional chitosan membrane (left)- and Biomend^®^ (right)-treated groups, the rectangular frame was chosen for bone density analysis. ((**A**), iii) Graph depicting the percentage of new bone formation (Here, significant difference was labeled as * *p* < 0.05). Reproduced with permission from [213], copyright Elsevier 2016. ((**B**), i) SEM image of fibrin with ε-aminocaproic acid loaded chitosan-tripolyphosphate nanoparticles (Scale bar: 2 μm). ((**B**), ii) Micro-computed topography-based measurement of the linear distance of vertical alveolar bone regeneration (compared with enamel matrix derivative, EMD). ((**B**), iii) Histological and immune-histological analyses of cementum formation and Sharpey’s fiber insertions to bone and newly formed cementum tissues. Upon comparison with fibrin-only (unmodified fibrin hydrogel) and EMD groups, the fibrin-ACP group facilitated the regeneration of periodontal tissues such as cementum on tooth-root surfaces, periodontal ligament, and the alveolar bone. More critically, the fibrin-ACP promoted Sharpey’s fiber formations and insertions into the cementum layers and alveolar bone surfaces, indicated by white arrows. Reproduced with permission from [215], copyright Elsevier 2017.

**Figure 15 pharmaceutics-15-01313-f015:**
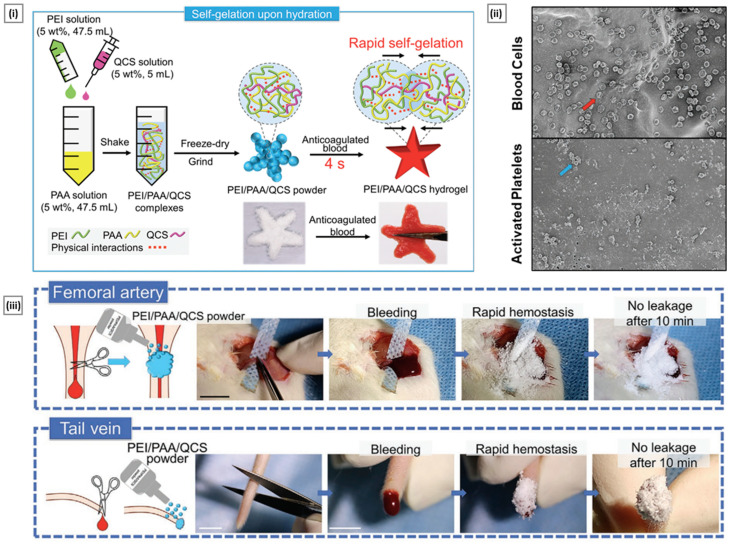
(**i**) Schematic illustrations for the preparation of PEI/PAA/QCS powder and the formation of PEI/PAA/QCS powder-derived hydrogel by adding anticoagulated blood. The photos are of the PEI/PAA/QCS powder and a pentagram PEI/PAA/QCS hydrogel formed by adding anticoagulated blood. (**ii**) SEM images of red blood cells (red arrow) and activated platelets (blue arrow) on the surface of PEI/PAA/QCS hydrogel. (**iii**) Schematic and photos of creating acute bleeding and stopping bleeding by applying PEI/PAA/QCS powder femoral artery and tail vein bleeding models. Reproduced with permission from [226], copyright Wiley-VCH 2021.

**Figure 16 pharmaceutics-15-01313-f016:**
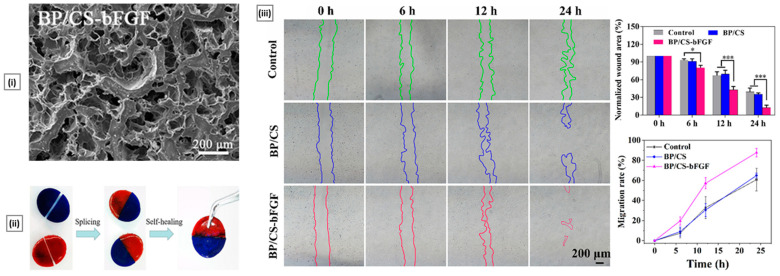
(**i**) SEM images of BP/CS-bFGF hydrogels. (Scale bar: 200 μm). (**ii**) Photographs of the self-healing performance of the BP/CS-bFGF hydrogel. (**iii**) Images of cell migration at different times with corresponding values of wound area closure treated with various samples and migration rate of hGFs cells upon the prepared hydrogels (Here, significant differences were defined as *p* values * ≤ 0.05, *** ≤ 0.001). Reproduced with permission from [228], copyright Elsevier 2022.

**Figure 17 pharmaceutics-15-01313-f017:**
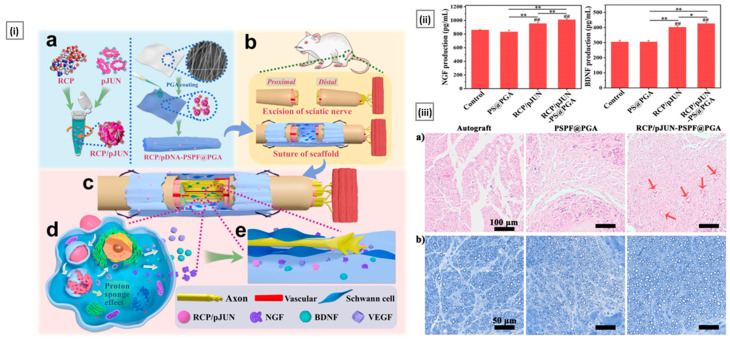
(**i**) Schematic mechanism of RCP/pJUN-PSPF@PGA scaffold on the nerve regeneration via the located gene transfection of c-JUN: (a) preparation of RCP/pJUN and RCP/pDNA-PSPF@PGA; (b) bridging surgery in sciatic nerve defect of rat; (c) located delivery of RCP/pJUN nanoparticles and nerve repair; (d) transfection of c-Jun via RCP/pJUN in cells and three factors secretion; (e) Bungner bands formation and axon regeneration. (**ii**) Nerve growth factor and brain-derived neurotrophic factor expression level in transfected RSC96s cell line (Here, ## *p* < 0.01, compared with control; * *p* < 0.05, ** *p* < 0.01). (**iii**) Evaluation of nerve regeneration: (a) H&E-stained tissue section images and (b) TB-stained tissue sections images at 12 weeks postoperatively. Reproduced with permission from [241], copyright Elsevier 2022.

**Figure 18 pharmaceutics-15-01313-f018:**
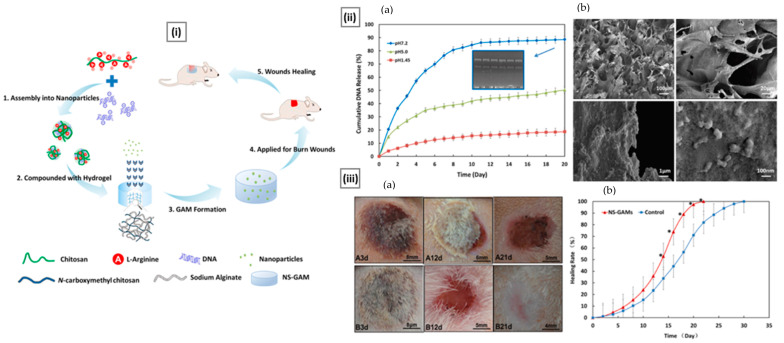
(**i**) Schematic illustration of NS-GAM as a gene delivery system for deep second-degree burn wound. (**ii**) In vitro release of pDNA from NS-GAM: (a) Cumulative amount of pDNA released in vitro from NS-GAM and the agarose gel electrophoresis of the plasmids; (b) SEM of the surface of NS-GAM. (**iii**) Gross examination and healing rate: (a) Observation of the deep second-degree burn wounds. A: Control group; B: NS-GAM group; (b) The calculated wound size reduction (Significant difference was considered at *p* < 0.05 *). Reproduced with permission from [243], copyright Elsevier 2022.

**Figure 19 pharmaceutics-15-01313-f019:**
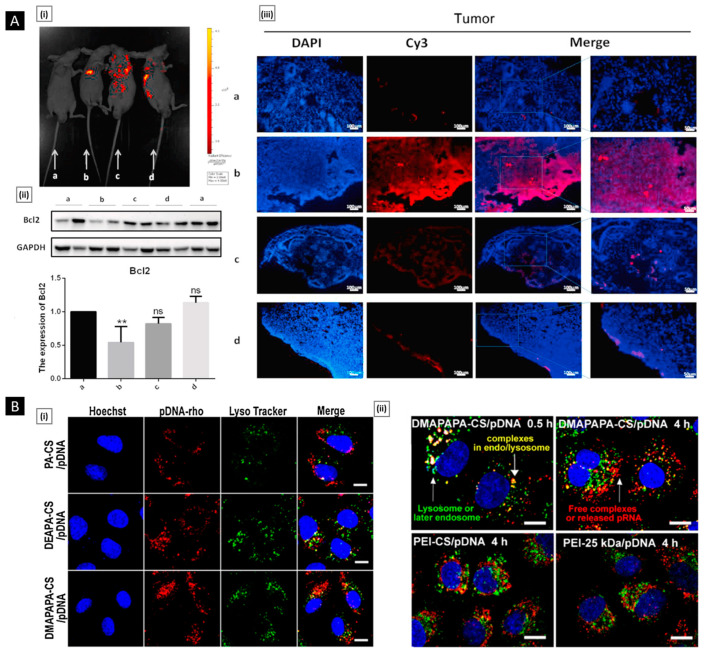
((**A**), i) The fluorescence intensities of tumor-bearing mice were monitored after 4 h of the last injection to detect the targeting effect. ((**A**), ii) The protein expression of CLl-2 was analyzed by western blotting (quantification of the protein level was normalized to GAPDH, significant differences were defined as *p* values * ≤ 0.05 and ns represents non-significance). ((**A**), iii) Immunofluorescence observation under the same exposure for each fluorescent channel after siRNA@chitosan-HAD nanoparticles (b), naked siRNA (c), and siRNA@chitosan nanoparticles (d) treatment compared with the control group (a). Reproduced with permission from [247], copyright Elsevier 2021. ((**B**), i) Confocal laser scanning microscopy images showing uptake in A549 cells after incubation with AA-CS/pDNA (2 μg/mL pDNA-rho) for 4 h. Hoechst 33342 (blue) and Lyso-Tracker Green were used to stain cell nuclei and lysosome, respectively, (Scale bar: 10 μm) ((**B**), ii) Confocal laser scanning microscopy images showing the endosomal escape of DMAPAPA-chitosan/pDNA-rho, PEI-chitosan/pDNA-rho, or PEI-25 kDa/pDNA-rho in A549 cells. The cells were stained with Lyso-Tracker Green and Hoechst 33342 (Scale bar: 10 μm). Reproduced with permission from [248], copyright Elsevier 2020.

**Table 1 pharmaceutics-15-01313-t001:** Chitosan-based Systems for Pharmaceutical and Biomedical Applications.

Type of System	Overview	Method of Preparation	Key Attributes/Features	Ref.
**Microspheres**	They are spherical particles with diameters of 10 μm to 1000 μm. Variants like hollow, core-shell, and fibrous microspheres allow modulation of the release profile.	Emulsion or thermal cross-linking Ionotropic gelation Coacervation/precipitation Spray drying	High drug loading and entrapment efficiency Sustained drug release Flexibility in the route of administration Physically stabilizes entrapped biomolecules	[118]
**Tablets**	It is used as a matrix material in tablet formation to control drug release, improve the stability/shelf life, and enhance the mechanical properties of the tablets	Direct compression Wet granulation	Can yield oral mucoadhesive tablets Provides extended drug release profile Improves gastric stability of orally delivered drugs	[119]
**Nanoparticles**	They are particulate systems employed for their tunable size (1 to 100 nm) and ability to undergo surface modification, making them versatile platforms for the targeted delivery of drugs, proteins, and genes.	Emulsion-Solvent Evaporation Reverse micellization Modified ionic gelation Polyelectrolyte complexation Desolvation technique Emulsification and cross-linking	High site-specific drug localization by using targeting ligands or by enhanced permeability and retention effect (in cancer) Can be modified to yield stimuli triggered (pH, temperature, redox) drug release Can facilitate co-delivery of drug molecules	[120]
**Nanofibers**	They are a novel platform where the drug is encapsulated within or attached to fibers with diameters in the nanometer range. The high surface area to volume ratio of nanofibers makes them suitable for controlled drug release and regenerative applications	Electrospinning Melt/solution blowing Templating	Large surface area relative to their volume allows for high drug loading and efficient release Modified drug release can be achieved by varying polymeric composition and concentration Multiple drugs can be incorporated together by means of preparing bilayered or trilayered nanofibers	[121]
**Hydrogels**	They are cross-linked polymer chains that form a 3D network capable of retaining large quantities of water. The gelation chemistry involved can be controlled at a molecular level, facilitating the creation of hydrogels with tailored physicochemical properties for various biomedical applications	Physical crosslinking Chemical crosslinking Enzymatic crosslinking Photo-crosslinking	Excellent injectability and biocompatibility Through molecular-level modifications, superior control over the platform’s degradation and, by extension, drug release rates can be controlled In situ forming properties can be incorporated Provides moisturizing effect when used tropically	[122]
**Membranes**	They are thin, flexible sheets that act as a dosage form and can be made to specific dimensions. They facilitate the direct release of drugs into biological environments.	Solvent casting Hot pressing	Chitosan improves the transport of polar drugs across epithelial surfaces Due to its cationic polyelectrolyte structure, the polymer has cell-binding ability and attracts negatively charged cell surfaces	[123]
**Powder/microgranules**	They are subcategories of solid dosage forms that consist of non-uniform micron-sized aggregates of drugs with polymeric chitosan.	Spray drying Gelation Salt-/Organic solvent-induced precipitation	Ease of preparation, handling, and use Improves chemical stability of incorporated drugs Small particle size of powder/granules facilitates rapid dissolution in body, thereby increasing bioavailability Useful for bulky drugs with a large dose	[124]

## Data Availability

Not applicable.

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
