# Peer review of "Chitosan: A Potential Biopolymer in Drug Delivery and Biomedical Applications"

_pharmaceutics, 2023, doi:10.3390/pharmaceutics15041313_

Round 1

Reviewer 1 Report

Very complete review paper with a scientifically sound distribution of the exposed material.Probably some references to nanotechnology could have given an uodated vision of the theme

Author Response

Reviewer 1

Very complete review paper with a scientifically sound distribution of the exposed material.

Probably some references to nanotechnology could have given an updated vision of the theme.

Ans: We appreciate the reviewer’s positive response for our current work. As suggested by reviewer, some references have been included in revised manuscript and highlighted in yellow colour.   

Reviewer 2 Report

This review includes many studies on chitosan, it also introduced the application of chitosan in various fields, I think this work can provide some new perspectives for the follow-up research on chitosan. However, in this recent manuscript, there are still some questions that need to be solved.

1. The author is not quite right about the arrangement of some groups of pictures. First, each picture needs to be unified, for example, The size of Figure 3 and Figure 5 is much larger than the other pictures. Second, the images inside the group should also be aligned, for example, in Figure 1 and Figure 12. 

2. some paragraphs on page 29 are not coherent and logical.

3. In lines 524 and line 633, the order of 5 and 6 should be reversed. This paper is mainly about the introduction of chitosan, it should expound on the application of chitosan first, and the application of chitosan derivatives should be introduced later.

4. In section 4.1, the introduction of antibacterial properties and chitosan oligosaccharide lactate is not detailed enough, so it is recommended to refer to: https://doi.org/10.1016/j.carbpol.2022.120485 ; https://doi.org/10.1016/j.pmatsci.2022.101045

5. In addition, there are some formatting problems in this article, for example, line 140, line 153, and line 170. (I gave up picking up the formatting errors after line 170, the author should check it carefully.)

Author Response

Reviewer 2

This review includes many studies on chitosan, it also introduced the application of chitosan in various fields, I think this work can provide some new perspectives for the follow-up research on chitosan. However, in this recent manuscript, there are still some questions that need to be solved.

  1. The author is not quite right about the arrangement of some groups of pictures. First, each picture needs to be unified, for example, The size of Figure 3 and Figure 5 is much larger than the other pictures. Second, the images inside the group should also be aligned, for example, in Figure 1 and Figure 12. 

Ans: As per the reviewer’s suggestion, required changes have been made and highlighted in yellow colour.

  1. Some paragraphs on page 29 are not coherent and logical.

Ans: As per the reviewer’s suggestion, suggested content has been verified and modified accordingly.

  1. In lines 524 and line 633, the order of 5 and 6 should be reversed. This paper is mainly about the introduction of chitosan, it should expound on the application of chitosan first, and the application of chitosan derivatives should be introduced later.

Ans: We appreciate reviewer’s keen observation and completely agree that if the contents of pristine chitosan and modified chitosan can be segregated then it would be better. However, we have attempted our best to organize the content in this way but found that it is not practically feasible. Hence, request reviewer to kindly waive this point and consider the organization of text in the current format.

  1. In section 4.1, the introduction of antibacterial properties and chitosan oligosaccharide lactate is not detailed enough, so it is recommended to refer to: https://doi.org/10.1016/j.carbpol.2022.120485 ; https://doi.org/10.1016/j.pmatsci.2022.101045

Ans: As per the reviewer’s recommendation, required information has been included in the revised manuscript and highlighted in yellow colour.

  1. In addition, there are some formatting problems in this article, for example, line 140, line 153, and line 170. (I gave up picking up the formatting errors after line 170, the author should check it carefully.)

Ans: We appreciate reviewers’ keen observation. However, the formatting has been done as per the journal’s guideline.

Reviewer 3 Report

This is a well-conducted research on the potential of chitosan. The authors thoroughly investigate the extraction and preparation methods of chitosan, the structural modifications and the multiple activities and applications shown by chitosan.

The manuscript is written in a scholarship style, avoiding verbosity, that facilitates reading. I strongly support publication in Pharmaceutics after few minor, merely formal corrections:

1.      The structure of chitosan should be shown earlier in the manuscript. I suggest adding it in the introduction.

2.      It would be better to use an acronym for chitosan instead of repeating the whole word in the manuscript.

3.      It seems right to me to suggest that you deepen the paragraph: " 5.10. Cyclodextrin linked chitosan", enriching it with these two manuscripts:

-        De Gaetano, F.; d’Avanzo, N.; Mancuso, A.; De Gaetano, A.; Paladini, G.; Caridi, F.; Venuti, V.; Paolino, D.; Ventura, C.A. Chitosan/Cyclodextrin Nanospheres for Potential Nose-to-Brain Targeting of Idebenone. Pharmaceuticals 2022, 15, 1206. https://doi.org/10.3390/ph15101206

-        De Gaetano, F.; Marino, A.; Marchetta, A.; Bongiorno, C.; Zagami, R.; Cristiano, M.C.; Paolino, D.; Pistarà, V.; Ventura, C.A. Development of Chitosan/Cyclodextrin Nanospheres for Levofloxacin Ocular Delivery. Pharmaceutics 2021, 13, 1293. https://doi.org/10.3390/pharmaceutics13081293

4.      Line 617: What does the acronym "CD" refer to? If it refers to free cyclodextrins, it must be placed after the word "cyclodextrin".

5.      The ionotropic gelation technique is used to form micro and nanoparticles. In paragraph 6.1 in table 1 it must be added that this method is used to prepare nanoparticles. you can insert this reference:

-        Cannavà, C.; De Gaetano, F.; Stancanelli, R.; Venuti, V.; Paladini, G.; Caridi, F.; Ghica, C.; Crupi, V.; Majolino, D.; Ferlazzo, G.; Tommasini, S.; Ventura, C.A. Chitosan-Hyaluronan Nanoparticles for Vinblastine Sulfate Delivery: Characterization and Internalization Studies on K-562 Cells. Pharmaceutics 2022, 14, 942. https://doi.org/10.3390/pharmaceutics14050942

Author Response

Reviewer 3

This is well-conducted research on the potential of chitosan. The authors thoroughly investigate the extraction and preparation methods of chitosan, the structural modifications and the multiple activities and applications shown by chitosan.

The manuscript is written in a scholarship style, avoiding verbosity, that facilitates reading. I strongly support publication in Pharmaceutics after few minors, merely formal corrections:

  1. The structure of chitosan should be shown earlier in the manuscript. I suggest adding it in the introduction.

Ans: As per the reviewer’s suggestion, structure of chitosan have been included in the revised manuscript.

  1. It would be better to use an acronym for chitosan instead of repeating the whole word in the manuscript.

Ans: We appreciate the valuable comment provided by the reviewer. However, inclusion of acronym for chitosan is difficult in the figures obtained from reported literature. Therefore, for the better understanding to reader we did not included acronym for chitosan. 

  1. It seems right to me to suggest that you deepen the paragraph: " 5.10. Cyclodextrin linked chitosan", enriching it with these two manuscripts:

-        De Gaetano, F.; d’Avanzo, N.; Mancuso, A.; De Gaetano, A.; Paladini, G.; Caridi, F.; Venuti, V.; Paolino, D.; Ventura, C.A. Chitosan/Cyclodextrin Nanospheres for Potential Nose-to-Brain Targeting of Idebenone. Pharmaceuticals 202215, 1206. https://doi.org/10.3390/ph15101206

-        De Gaetano, F.; Marino, A.; Marchetta, A.; Bongiorno, C.; Zagami, R.; Cristiano, M.C.; Paolino, D.; Pistarà, V.; Ventura, C.A. Development of Chitosan/Cyclodextrin Nanospheres for Levofloxacin Ocular Delivery. Pharmaceutics 202113, 1293. https://doi.org/10.3390/pharmaceutics13081293

Ans: As per the reviewer’s suggestion, suggested references have been included in the revised manuscript and highlighted in yellow colour.

  1. Line 617: What does the acronym "CD" refer to? If it refers to free cyclodextrins, it must be placed after the word "cyclodextrin".

Ans: The required changes have been incorporated in revised manuscript and highlighted in yellow colour.

  1. The ionotropic gelation technique is used to form micro and nanoparticles. In paragraph 6.1 in table 1 it must be added that this method is used to prepare nanoparticles. you can insert this reference:

-        Cannavà, C.; De Gaetano, F.; Stancanelli, R.; Venuti, V.; Paladini, G.; Caridi, F.; Ghica, C.; Crupi, V.; Majolino, D.; Ferlazzo, G.; Tommasini, S.; Ventura, C.A. Chitosan-Hyaluronan Nanoparticles for Vinblastine Sulfate Delivery: Characterization and Internalization Studies on K-562 Cells. Pharmaceutics 202214, 942. https://doi.org/10.3390/pharmaceutics14050942

Ans: As per the reviewer’s suggestion, suggested reference has been included in the revised manuscript and highlighted in yellow colour.

Reviewer 4 Report

In this manuscript (pharmaceutics-2308871) entitled “Chitosan: A Potential Biopolymer in Drug Delivery and Biomedical Applications”, the authors comprehensively introduced the source, extraction techniques, preparation, bioactivities, modification, and multifunctional applications of chitosan. Especially, the paper focuses on the application of chitosan in drug delivery and biomedicine. The manuscript is well-organized and the conclusion is supported by the analysis. Therefore, this reviewer would suggest an acceptance after addressing the following minor issues.

1.      The reason why authors chose chitosan as the topic should be further clarified with more detailed introduction on the structure, properties and applications of chitosan with necessary supporting articles: Sources, production and commercial applications of fungal chitosan: A review; Recent advancements in applications of chitosan-based biomaterials for skin tissue engineering; New Ulva lactuca Algae Based Chitosan Bio-composites for Bioremediation of Cd(II) Ions; Sandwich-like chitosan porous carbon Spheres/MXene composite with high specific capacitance and rate performance for supercapacitors; etc.

2.      Chitosan is obtained by deacetylation of chitin and consists of β-(1-4) N-acetyl glucosamine and D-glucosamine repeating units. Please supplement the structural formula of chitosan to better understand its chemical structure.

3.      It is recommended to supplement relevant charts between “Chitin and chitosan extraction techniques” of Part 2 and “Modified Chitosan Preparation/ Derivatives” of Part 5 to intuitively demonstrate the viewpoint.

4.      The application of chitosan is the focus of the paper. One picture to show the various applications of chitosan is suggested for better understanding of this work and highlight the paper.

5.      Figures 2A and 9B are not mentioned in the paper. The resolution of Figure 3B is too low. Please provide a clearer image.

6.      “spices” in page 3, line 98 should be “species”. Authors should carefully recheck the whole manuscript.

7.      Some of the figures are too big, which are suggested to be divided into two figures.

8.      Please carefully recheck the whole references to make sure full information is provided, such as volume and pages.

Author Response

Reviewer 4

In this manuscript (pharmaceutics-2308871) entitled “Chitosan: A Potential Biopolymer in Drug Delivery and Biomedical Applications”, the authors comprehensively introduced the source, extraction techniques, preparation, bioactivities, modification, and multifunctional applications of chitosan. Especially, the paper focuses on the application of chitosan in drug delivery and biomedicine. The manuscript is well-organized, and the conclusion is supported by the analysis. Therefore, this reviewer would suggest an acceptance after addressing the following minor issues.

Comment: We appreciate the reviewer’s positive response for our current work

  1. The reason why authors chose chitosan as the topic should be further clarified with more detailed introduction on the structure, properties and applications of chitosan with necessary supporting articles: Sources, production and commercial applications of fungal chitosan: A review; Recent advancements in applications of chitosan-based biomaterials for skin tissue engineering; New Ulva lactuca Algae Based Chitosan Bio-composites for Bioremediation of Cd(II) Ions; Sandwich-like chitosan porous carbon Spheres/MXene composite with high specific capacitance and rate performance for supercapacitors; etc.

Ans: As per the reviewer’s suggestion, required information has been included in the revised manuscript and highlighted in yellow colour.

  1. Chitosan is obtained by deacetylation of chitin and consists of β-(1-4) N-acetyl glucosamine and D-glucosamine repeating units. Please supplement the structural formula of chitosan to better understand its chemical structure.

Ans: As per the reviewer’s suggestion, suggested structure have been included in the revised manuscript and highlighted in yellow colour.

  1. It is recommended to supplement relevant charts between “Chitin and chitosan extraction techniques” of Part 2 and “Modified Chitosan Preparation/ Derivatives” of Part 5 to intuitively demonstrate the viewpoint.

Ans: As per the reviewer’s recommendation, relevant chart for chitin and chitosan extraction technique and suitable figure for modified chitosan preparation/ derivatives have been included in the revised manuscript and highlighted in yellow colour.

  1. The application of chitosan is the focus of the paper. One picture to show the various applications of chitosan is suggested for better understanding of this work and highlight the paper.

Ans: As per the reviewer’s suggestion, figure demonstrating various applications of chitosan have been included in the revised manuscript

  1. Figures 2A and 9B are not mentioned in the paper. The resolution of Figure 3B is too low. Please provide a clearer image.

Ans: As per the reviewer’s suggestion, required information has been included in the revised manuscript and highlighted in yellow colour.

  1. “spices” in page 3, line 98 should be “species”. Authors should carefully recheck the whole manuscript.

Ans: Required changes have been corrected in the revised manuscript. Further, whole manuscript have been verified for typographical errors.

  1. Some of the figures are too big, which are suggested to be divided into two figures.

Ans: As per the reviewer’s suggestion, required changes have been made in the revised manuscript and highlighted in yellow colour.

  1. Please carefully recheck the whole references to make sure full information is provided, such as volume and pages.

Ans: As suggested by reviewer, all references have been verified and corrected and as per the journal’s guideline.

Round 2

Reviewer 4 Report

Authors have made revisions according to previous comments. However, there are still some minor issues to be addressed.

1. There is only section 2.1, where is the section 2.2? Please revise.

2. When introducing the applications of chitosan, two more different applications in removal of heavy metal ions and supercapacitors should be included: New Ulva lactuca Algae Based Chitosan Bio-composites for Bioremediation of Cd(II) Ions; Sandwich-like chitosan porous carbon Spheres/MXene composite with high specific capacitance and rate performance for supercapacitors; etc.

3. Generally, it is not proper to include sub-section under introduction section. It is suggested to combine the section 1.1 and section 2 together.

Author Response

We appreciate the reviewer for the valuable suggestions to improve the quality of manuscript. 

  1. There is only section 2.1, where is the section 2.2? Please revise.

Ans: As per the reviewer’s suggestion, required changes have been made and highlighted in yellow colour.

  1. When introducing the applications of chitosan, two more different applications in removal of heavy metal ions and supercapacitors should be included: New Ulva lactuca Algae Based Chitosan Bio-composites for Bioremediation of Cd(II) Ions; Sandwich-like chitosan porous carbon Spheres/MXene composite with high specific capacitance and rate performance for supercapacitors; etc.

Ans: We would like to bring to the reviewer’s notice that the scope of the present work is drug delivery and biomedical applications of chitosan. Hence, we have not included its applications in areas such as water treatment and supercapacitors.

  1. Generally, it is not proper to include sub-section under introduction section. It is suggested to combine the section 1.1 and section 2 together.

Ans: As per the reviewer’s suggestion, required changes have been made in the revised manuscript and highlighted in yellow colour.